# A Unified Theory of Quantum Neural Network Loss Landscapes

**Eric R. Anschuetz**
Institute for Quantum Information and Matter & Walter Burke Institute for Theoretical Physics, Caltech
Pasadena, CA 91125, USA
eans@caltech.edu

## Abstract

Classical neural networks with random initialization famously behave as Gaussian processes in the limit of many neurons, which allows one to completely characterize their training and generalization behavior. No such general understanding exists for quantum neural networks (QNNs), which—outside of certain special cases—are known to not behave as Gaussian processes when randomly initialized. We here prove that QNNs and their first two derivatives instead generally form what we call *Wishart processes*, where certain algebraic properties of the network determine the hyperparameters of the process. This Wishart process description allows us to, for the first time: give necessary and sufficient conditions for a QNN architecture to have a Gaussian process limit; calculate the full gradient distribution, generalizing previously known barren plateau results; and calculate the local minima distribution of algebraically constrained QNNs. Our unified framework suggests a certain simple operational definition for the "trainability" of a given QNN model using a newly introduced, experimentally accessible quantity we call the *degrees of freedom* of the network architecture.

## 1 Introduction

### 1.1 Motivation

One of the miracles of machine learning on classical computers is that simple, gradient-based optimizers can efficiently find the minimum of extremely high-dimensional, nonconvex loss landscapes, allowing for the efficient training of deep neural networks. Over the past decade this has been understood in more and more detail via random matrix theory. In particular, it is now known that the loss landscapes of randomly initialized, wide neural networks are distributed as Gaussian processes with covariance given by the so-called *neural tangent kernel* (NTK) (Neal, 1996; Choromanska et al., 2015; Chaudhari, 2018; Lee et al., 2018). The NTK is completely determined by the neural structure of the network, linking the asymptotic behavior of the network to architectural choices made in its construction. This understanding of classical neural networks has been used to show that wide neural networks train efficiently via gradient descent (Choromanska et al., 2015; Chaudhari, 2018; Jacot et al., 2018; Allen-Zhu et al., 2019). It also explains other emergent phenomena in deep learning, such as the remarkably good generalization performance of neural networks beyond what learning theory predicts (Jacot et al., 2018; Wei et al., 2022).

One might hope that a similar, universal story would exist for *quantum neural networks* (QNNs). These are classes of neural networks where the associated loss function $\ell$ takes as input a *quantum state* $\boldsymbol{\rho} \succeq \mathbf{0} \in \mathbb{C}^{N \times N}$ with trace 1, conjugates $\boldsymbol{\rho}$ by a unitary operation $\boldsymbol{U_\theta} \in \mathrm{U}(N)$ parameterized by $\boldsymbol{\theta} \in \mathbb{R}^p$, and then takes the Frobenius inner product with the Hermitian $\boldsymbol{O} \in \mathbb{C}^{N \times N}$. That is,

$$\ell(\boldsymbol{\rho}; \boldsymbol{\theta}) = \mathrm{Tr}\left(\boldsymbol{\rho} \boldsymbol{U_\theta^\dagger} \boldsymbol{O} \boldsymbol{U_\theta}\right). \tag{1}$$

Such networks are defined by the choice of parameterization for $\boldsymbol{U_\theta}$ (typically called the *ansatz*) and the Hermitian $\boldsymbol{O}$ (here called the *objective observable*) (Peruzzo et al., 2014), and are known to efficiently (on a quantum computer) perform learning tasks provably difficult using traditional machine learning methods (Liu et al., 2021; Hastings & O'Donnell, 2022; Anschuetz et al., 2023b;

Huang et al., 2024; Anschuetz & Gao, 2024). Unfortunately, it is known that QNNs generally do not have a Gaussian process (or *quantum neural tangent kernel (QNTK)*) asymptotic limit (García-Martín et al., 2023; Girardi & Palma, 2024). Indeed, it is not even obvious whether an equivalently simple universal description exists due to QNN training dynamics differing completely in various parameter regimes. For instance, the loss landscapes of generic shallow QNNs are known to be described by so-called *Wishart hypertoroidal random fields* (WHRFs) and dominated by poor local minima (Anschuetz, 2022; Anschuetz & Kiani, 2022); this is a far cry from the effectively convex training landscapes of deep quantum networks which, in certain circumstances, *can* be shown to have a Gaussian process limit (Liu et al., 2022; 2023; You et al., 2022; García-Martín et al., 2023; Girardi & Palma, 2024; García-Martín et al., 2024). Unfortunately, these deep networks are still untrainable in practice due to an exponential decay in their gradients known as the *barren plateau* phenomenon (McClean et al., 2018), making estimating gradients on a quantum computer asymptotically intractable.[1]

The presence of barren plateaus or poor local minima in QNN loss landscapes paints a pessimistic picture for the practical utility of generic QNNs. However, these negative results can be circumvented by considering *structured* QNNs, further complicating any unifying theory of the asymptotic training behavior of QNNs. For instance, $U_\theta$ may be constrained to belong to some low-dimensional Lie subgroup $G$ of the full unitary group, and $O$ in the generating algebra of $G$. This is known as the *Lie algebra-supported ansatz* (LASA) setting, and has been shown to be capable of preventing the barren plateau phenomenon from occurring (Fontana et al., 2024; Ragone et al., 2024). Though many of the initial proposals for such structured networks have since been "dequantized" (Anschuetz et al., 2023a; Goh et al., 2023; Cerezo et al., 2023)—i.e., efficient classical algorithms have been found which simulate such networks—there still do exist unconditionally provable quantum advantages in using such networks for certain learning tasks (Anschuetz et al., 2023b; Anschuetz & Gao, 2024). Though this is perhaps the most promising direction in QNN research, nothing concretely is known about the loss landscapes of such networks beyond the variance of the loss function over parameter space.

## 1.2 Contributions

Fully understanding how the various phenomenologies of QNN loss landscapes are related is important if we ever hope to have as deep an understanding of QNNs as the neural tangent kernel has enabled for classical neural networks. Motivated by this, we here for the first time prove a concise asymptotic limit for the loss functions of effectively all QNNs with approximately uniformly random initialization, and show that it unifies much of our previous understanding of QNN training behavior. We then use this new asymptotic description to prove a variety of novel results on the asymptotic training behavior of QNNs.

We achieve this by demonstrating that all QNNs have a natural algebraic structure described by a *Jordan subalgebra* $\mathcal{A}$ of the complex Hermitian matrices $\mathcal{H}^N(\mathbb{C})$ generated by $\left\{U_\theta^\dagger O U_\theta\right\}_\theta$ under the anticommutator $\{A, B\} = AB + BA$. The Jordan subalgebras of $\mathcal{H}^N(\mathbb{C})$ have been completely classified in the sense that they are expressible as a direct sum of subalgebras:

$$\mathcal{A} \cong \bigoplus_\alpha \mathcal{A}_\alpha, \tag{2}$$

where for our purposes each $\mathcal{A}_\alpha$ is isomorphic to the algebra of $N_\alpha \times N_\alpha$ Hermitian matrices $\mathcal{H}^{N_\alpha}(\mathbb{F}_\alpha)$ over a field $\mathbb{F}_\alpha$ that is one of the reals $\mathbb{R}$, complex numbers $\mathbb{C}$, or quaternions $\mathbb{H}$. This allows us to show that the loss functions and first two derivatives of randomly initialized QNNs have a concise asymptotic description in terms of *Wishart-distributed random matrices* over $\mathbb{F}_\alpha$:

$$W_\alpha = X_\alpha X_\alpha^\dagger, \tag{3}$$

where $X_\alpha$ is $N_\alpha \times r_\alpha$ with i.i.d. standard Gaussian entries over $\mathbb{F}_\alpha$ and $r_\alpha$ is called the *degrees of freedom* of $W_\alpha$. We give an explicit expression for $r_\alpha$ based on the structure of the network, and we find that it dictates the asymptotic behavior of the loss landscape. We call this connection between the algebraic structure of a given QNN and its asymptotic Wishart process description a *Jordan*

---

[1]Girardi & Palma (2024) also study instances which do not suffer from barren plateaus, but which only achieve an asymptotically-vanishing improvement over random $\theta$ in the optimization of Eq. (1).

| **Barren Plateaus** | | **Gaussian Processes (QNTK)** | |
| --- | --- | --- | --- |
| Reference | Corollary 4 | Reference | Corollary 5 |
| McClean et al. (2018) | $\mathbb{F} = \mathbb{C}$ | García-Martín et al. (2023) | $\mathbb{F} = \mathbb{C}$, $\boldsymbol{O}$ Pauli |
| Cerezo et al. (2021) | $\{\mathbb{F}_\alpha = \mathbb{C}\}_\alpha$ | Girardi & Palma (2024) | $\mathbb{F} = \mathbb{C}$, $\boldsymbol{O}$ 1-local |
| Ragone et al. (2024) | $e^{i\boldsymbol{O}} \in \mathrm{Aut}\,(\mathcal{A})$ | García-Martín et al. (2024) | $\mathbb{F} = \mathbb{H}$, $\boldsymbol{O}$ Pauli |

| **Local Minima** | |
| --- | --- |
| Reference | Corollary 6 |
| Anschuetz (2022) | $\mathbb{F} = \mathbb{C}$ |
| Anschuetz & Kiani (2022) | $\{\mathbb{F}_\alpha = \mathbb{C}\}_\alpha$, $p < 2r_\alpha$ |
| You et al. (2022) | $p \gg r$ |

Table 1: **Previous quantum neural network loss landscape results.** We summarize (a representative subset of) previous results in quantum neural network loss landscape theory, as described in Sec. 1.1. We also state which limits of our Corollaries 4, 5, and 6 encompass these referenced results. Here, $\mathcal{A}$ is the Jordan algebra, $\mathbb{F}_\alpha$ are the fields, and $r_\alpha$ are the degrees of freedom parameters as defined in the associated corollary statements; $\boldsymbol{O}$ is the objective observable; and $p$ is the number of trained parameters.

*algebraic Wishart system* (JAWS). We summarize in Table 1 how this JAWS description generalizes previous models of QNN loss landscapes which only held in specific parameter regimes.

The JAWS framework allows us to prove many new, general properties of QNNs. First, we prove a generalized barren plateau result for the variance of the loss function $\ell\,(\boldsymbol{\rho}; \boldsymbol{\theta})$ of a given QNN over its initialization:

$$\mathrm{Var}_{\boldsymbol{\theta}}\,[\ell\,(\boldsymbol{\rho}; \boldsymbol{\theta})] = \sum_\alpha \frac{\mathrm{Tr}\left((\boldsymbol{O}^\alpha)^2\right)\mathrm{Tr}\left((\boldsymbol{\rho}^\alpha)^2\right)}{\dim_\mathbb{R}\left(\mathrm{Aut}\,(\mathcal{A}_\alpha)\right)}, \tag{4}$$

where $\mathcal{A} = \bigoplus_\alpha \mathcal{A}_\alpha$ is the Jordan algebra associated with the QNN, $\dim_\mathbb{R}\left(\mathrm{Aut}\,(\mathcal{A}_\alpha)\right)$ is the dimension of the automorphism group of $\mathcal{A}_\alpha$ as a real manifold, and $\cdot^\alpha$ denotes projection into $\mathcal{A}_\alpha$ (see Eq. (12) for a mathematical definition). This extends previously known barren plateau results proved in the LASA framework (Fontana et al., 2024; Ragone et al., 2024) to the setting where neither $\boldsymbol{\rho}$ nor $\boldsymbol{O}$ are elements of the algebra generating $\mathrm{Aut}\,(\mathcal{A}_\alpha)$. It also captures the gradient scaling in the setting of so-called matchgate networks (Diaz et al., 2023), unifying barren plateau results beyond just the LASA setting.

We are also able to calculate the asymptotic density of local minima at a loss value $\ell = z$ for a QNN with associated Jordan algebra $\mathcal{A} = \bigoplus_\alpha \mathcal{A}_\alpha$:

$$\kappa\,(z) = \underset{\alpha}{\text{\Large$\divideontimes$}} f_\Gamma\,(z; k_\alpha, s_\alpha) = \int_{\sum_\alpha z_\alpha = z} \prod_\alpha \frac{\mathrm{d}z_\alpha}{\Gamma\,(k_\alpha)\,s_\alpha^{k_\alpha}} z_\alpha^{k_\alpha - 1} \exp\left(-\frac{z_\alpha}{s_\alpha}\right); \tag{5}$$

i.e., we show that it is a convolution of gamma distributions $f_\Gamma$ with shape and scale parameters $k_\alpha$ and $s_\alpha$, respectively, which we calculate in Sec. 5.3. The $k_\alpha$ and $s_\alpha$ are such that $\kappa\,(z)$ experiences a phase transition: local minima of a network with $p$ parameters are concentrated near the global minimum if and only if the network is *overparameterized*, which occurs when (up to a constant):

$$p \gtrsim \max_\alpha \frac{\mathrm{Tr}\,(\boldsymbol{O}^\alpha)^2}{\mathrm{Tr}\left((\boldsymbol{O}^\alpha)^2\right)}. \tag{6}$$

This result generalizes previous studies of local minima in QNNs (Anschuetz, 2022; Anschuetz & Kiani, 2022) to a setting where the variational ansatz may have some sort of algebraic structure.[2]

Finally, we prove the necessary and sufficient conditions for a class of QNNs to asymptotically converge to a Gaussian process limit. Taken together, our results indicate that QNNs are efficiently trainable asymptotically using problem-independent optimization algorithms if and only if Eq. (4) is not exponentially vanishing with the problem size and Eq. (6) is satisfied.

---

[2]Anschuetz & Kiani (2022) consider local minima in local, shallow networks, where the Hilbert space can be decomposed into a direct sum of Hilbert spaces associated with the light cones of local observables. Here, we allow for general algebraic structures.

## 2 PRELIMINARIES

### 2.1 QUANTUM NEURAL NETWORKS

We first review *quantum neural networks* (QNNs). These are defined by a parameterized unitary:

$$\boldsymbol{U_\theta} = \boldsymbol{g}_{p+1} \prod_{i=p}^{1} \exp\left(-\mathrm{i}\theta_i \boldsymbol{A}_i\right) \boldsymbol{g}_i, \tag{7}$$

often called an *ansatz* in the physics literature. Here, the $\boldsymbol{A}_i$ are complex Hermitian $N \times N$ matrices and the $\boldsymbol{g}_i$ are $N \times N$ unitary matrices. Generally, the $\boldsymbol{g}_i$ and $\exp\left(-\mathrm{i}\theta_i \boldsymbol{A}_i\right)$ may be constrained to belong to a path-connected Lie subgroup $G$ of the unitary group $\mathrm{U}(N)$ to strengthen the inductive bias of the network (Meyer et al., 2023). Such a constraint has also been used as a theoretical model for shallow quantum networks, where $G$ is approximately the direct product $\times_\alpha \mathrm{U}(N_\alpha)$ of unitaries acting on local patches in the network (Anschuetz & Kiani, 2022).

Training such networks involves minimizing an empirical risk of the form:

$$f(\boldsymbol{\theta}) = \frac{1}{|\mathcal{R}|} \sum_{\boldsymbol{\rho} \in \mathcal{R}} \ell(\boldsymbol{\rho}; \boldsymbol{\theta}) = \frac{1}{|\mathcal{R}|} \sum_{\boldsymbol{\rho} \in \mathcal{R}} \mathrm{Tr}\left(\boldsymbol{U_\theta} \boldsymbol{\rho} \boldsymbol{U_\theta}^\dagger \boldsymbol{O}\right). \tag{8}$$

Here, $\mathcal{R}$ can be thought of a data set comprising multiple input *quantum states* $\boldsymbol{\rho}$—that is, $\boldsymbol{\rho} \succeq \boldsymbol{0} \in \mathbb{C}^{N \times N}$ with trace 1—and $\boldsymbol{O} \in \mathbb{C}^{N \times N}$ is Hermitian. When $\boldsymbol{O}$ and the algebra elements generating $\boldsymbol{U_\theta}$ can be expressed as a sum of $\mathrm{O}\left(\mathrm{poly}\log(N)\right)$ Pauli matrices, such a loss can be estimated in $\mathrm{O}\left(\mathrm{poly}\log(N)\right)$ time on a quantum computer (Nielsen & Chuang, 2010b).

Our first main result is that all losses of the form of $\ell(\boldsymbol{\rho}; \boldsymbol{\theta})$ can be interpreted in terms of automorphisms of Jordan algebras. In preparation for discussing this connection, we now review Jordan algebras.

### 2.2 JORDAN ALGEBRAS

A *Jordan algebra* over the reals is formally a real vector space $V$ with a commutative multiplication operation $\circ$ acting on $u, v \in V$ satisfying the *Jordan identity*:

$$u \circ ((u \circ u) \circ v) = (u \circ u) \circ (u \circ v), \tag{9}$$

which ensures the associativity of the power. A simple example is the Jordan algebra $\mathcal{H}^N(\mathbb{C})$ of $N \times N$ complex-valued Hermitian matrices with $\circ$ given by the anticommutator $\boldsymbol{A} \circ \boldsymbol{B} = \boldsymbol{AB} + \boldsymbol{BA}$.

The Jordan subalgebras $\mathcal{A}$ of $\mathcal{H}^N(\mathbb{C})$ have been completely classified (Koecher, 1999d) and are known to have a semisimple decomposition:

$$\mathcal{A} \cong \bigoplus_\alpha \mathcal{A}_\alpha; \tag{10}$$

that is, $\mathcal{A}$ is isomorphic to a direct sum of subalgebras $\mathcal{A}_\alpha$. For our purposes, each $\mathcal{A}_\alpha$ is isomorphic to the algebra $\mathcal{H}^{N_\alpha}(\mathbb{F}_\alpha)$ of $N_\alpha \times N_\alpha$ Hermitian matrices over a field $\mathbb{F}_\alpha$ that is one of the reals, complex numbers, or quaternions. We label these three cases with the integers $\beta_\alpha = 1, 2, 4$, respectively, and call the representation of each of these algebras as $N_\alpha \times N_\alpha$ matrices over $\mathbb{F}_\alpha$ the *defining representation*. We call the $\mathcal{A}_\alpha$ the *simple components* of $\mathcal{A}$ and say that $\mathcal{A}$ is *simple* when there is only a single $\mathcal{A}_\alpha$, i.e., when $\mathcal{A} \cong \mathcal{H}^N(\mathbb{F})$ for some $\mathbb{F}$.

The automorphism group $\mathrm{Aut}(\mathcal{A})$ of such a Jordan algebra $\mathcal{A}$ is also known: its path-connected components are isomorphic to direct products of the classical Lie groups $\mathrm{SO}(N_\alpha)$, $\mathrm{SU}(N_\alpha)$, and $\mathrm{Sp}(N_\alpha)$, each a subgroup of the corresponding $\mathrm{Aut}(\mathcal{A}_\alpha)$ (Koecher, 1999b; Orlitzky, 2024). As a Lie group, each $\mathrm{Aut}(\mathcal{A}_\alpha)$ has a well-defined dimension as a real manifold which we denote as $\dim_\mathbb{R}(\mathrm{Aut}(\mathcal{A}_\alpha))$.

It is also known that the Frobenius inner product $\mathrm{Tr}_R\left(\boldsymbol{A}^\dagger \boldsymbol{B}\right)$ between $A, B \in \mathcal{A}_\alpha$ in any representation $R$ is always proportional to the Frobenius inner product in the defining representation (Koecher, 1999a). In particular, for any $A, B \in \mathcal{A}_\alpha \subseteq \mathcal{A} \subseteq \mathcal{H}^N(\mathbb{C})$, using $\mathrm{Tr}_\alpha(\cdot)$ to denote the trace in the defining representation of $\mathcal{A}_\alpha$ and $\mathrm{Tr}(\cdot)$ to denote that of $\mathcal{H}^N(\mathbb{C})$, we have:

$$\mathrm{Tr}(\boldsymbol{AB}) = I_\alpha \, \mathrm{Tr}_\alpha(\boldsymbol{AB}) \tag{11}$$

for some $I_\alpha > 0$ that is independent of $A, B$. In what follows we will continue to use $\mathrm{Tr}\,(\cdot)$ to denote the trace in the defining representation of $\mathcal{H}^N\,(\mathbb{C})$ and $\mathrm{Tr}_\alpha\,(\cdot)$ to denote that of $\mathcal{A}_\alpha$. More details on Jordan algebras and some subtleties in their classification are provided in Appendix A.2.

Finally, we define the *projection* of an algebra element $A \in \mathcal{A} \cong \bigoplus_\alpha \mathcal{A}_\alpha$ into one of the $\mathcal{A}_\alpha$; to represent this, we use the notation $A^\alpha$. Mathematically, letting $\mathrm{Tr}_\alpha\,(\cdot)$ denote the trace in the defining representation of $\mathcal{A}_\alpha$ and considering an orthonormal[3] basis $\{B_{\alpha,i}\}_i$ of $\mathcal{A}_\alpha \subseteq \mathcal{A}$, we have:

$$A^\alpha \equiv \sum_i \mathrm{Tr}_\alpha\,(\boldsymbol{B}_{\alpha,i}\boldsymbol{A})\,B_{\alpha,i}. \tag{12}$$

We use the same notation for the equivalent projection of a Lie group element $g \in G = \bigtimes_\alpha G_\alpha$ into one of the $G_\alpha$. Using $\{B_{\alpha,i}\}_i$ to denote an orthonormal basis of the Lie algebra generating $G_\alpha$, and using the observation that $g = \exp\left(\sum_{\alpha,i} c_{\alpha,i} B_{\alpha,i}\right)$ for some $\{c_{\alpha,i} \in \mathbb{R}\}_{\alpha,i}$, we define:

$$g^\alpha \equiv \exp\left(\sum_i c_{\alpha,i} B_{\alpha,i}\right). \tag{13}$$

# 3 JORDAN ALGEBRAIC DESCRIPTIONS OF QUANTUM NEURAL NETWORKS

Recall the general form of a QNN loss function:

$$\ell\,(\boldsymbol{\rho}; \boldsymbol{\theta}) = \mathrm{Tr}\left(\boldsymbol{\rho}\boldsymbol{U}_{\boldsymbol{\theta}}^\dagger \boldsymbol{O}\boldsymbol{U}_{\boldsymbol{\theta}}\right), \tag{14}$$

where

$$\boldsymbol{U}_{\boldsymbol{\theta}} = \boldsymbol{g}_{p+1}\prod_{i=p}^{1}\exp\left(-\mathrm{i}\theta_i\boldsymbol{A}_i\right)\boldsymbol{g}_i \tag{15}$$

generally may be such that the $\boldsymbol{g}_i$ and $\exp\left(-\mathrm{i}\theta_i\boldsymbol{A}_i\right)$ belong to some path-connected Lie subgroup $G \subseteq \mathrm{U}\,(N)$.

The $\boldsymbol{U}_{\boldsymbol{\theta}}^\dagger\boldsymbol{O}\boldsymbol{U}_{\boldsymbol{\theta}}$ generate under the anticommutator a Jordan subalgebra $\mathcal{A}$ of the $N \times N$ Hermitian matrices $\mathcal{H}^N\,(\mathbb{C})$. Due to this fact, as well as the classification results discussed in Sec. 2.2, we may equivalently consider Eq. (14) in the following way:

1. $\boldsymbol{O} \in \mathcal{A} \cong \bigoplus_\alpha \mathcal{H}^{N_\alpha}\,(\mathbb{F}_\alpha)$.
2. $\boldsymbol{U}_{\boldsymbol{\theta}} \in G = \bigtimes_\alpha G_\alpha$, where each $G_\alpha$ is isomorphic to a subgroup of $\mathrm{Aut}\,(\mathcal{A}_\alpha)$ that is one of $\mathrm{SO}\,(N_\alpha)$, $\mathrm{SU}\,(N_\alpha)$, or $\mathrm{Sp}\,(N_\alpha)$.
3. For $I_\alpha$ the proportionality constant of Eq. (11),

$$\ell\,(\boldsymbol{\rho}; \boldsymbol{\theta}) = \sum_\alpha I_\alpha \mathrm{Tr}_\alpha\left(\boldsymbol{\rho}^\alpha \boldsymbol{U}_{\boldsymbol{\theta}}^{\alpha\dagger}\boldsymbol{O}^\alpha\boldsymbol{U}_{\boldsymbol{\theta}}^\alpha\right). \tag{16}$$

We discuss the universality of this Jordan algebraic description in more detail in Appendix B.1.

This description of QNN loss landscapes in terms of Jordan algebraic properties allows us to give a simple way to classify various QNN architectures through the $N_\alpha$, $\mathbb{F}_\alpha$, $I_\alpha$, and $\boldsymbol{O}^\alpha$; we call this collection of algebraic objects a *Jordan algebraic Wishart system* (JAWS). We will next tie this classification to the asymptotic properties of the loss landscape through the use of Wishart random matrices, justifying the use of "Wishart" in the nomenclature.

# 4 QUANTUM NEURAL NETWORKS ARE WISHART PROCESSES

Our main result is proving that the loss functions of wide QNNs—that is, as $\dim\,(\mathcal{A}) \to \infty$—form *Wishart processes*. Such processes can be written exactly in terms of the matrix elements of Wishart-distributed positive semidefinite random matrices $\boldsymbol{W}$, which are distributed as:

$$\boldsymbol{W} = \boldsymbol{X}\boldsymbol{X}^\dagger \tag{17}$$

---

[3] Orthonormal with respect to the Frobenius inner product in the defining representation of $\mathcal{A}_\alpha$.

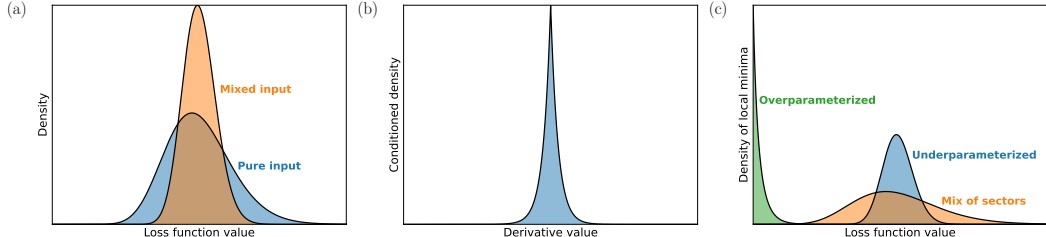

Figure 1: **Loss and derivative densities.** (a) The loss density when the quantum neural network has a *pure input* (i.e., rank-1) and when it has a *mixed input* (i.e., of rank greater than 1). The distributions are centered at the mean eigenvalue of the objective observable. The mixed input density also illustrates when the input is mixed when projected into any simple component of the Jordan algebra associated with the network. (b) The gradient density conditioned on a nonzero loss function value. The distribution is centered at zero. (c) The density of local minima when the quantum neural network is *underparameterized*, *overparameterized*, and when some simple components of the associated Jordan algebra are underparameterized and some are overparameterized (*mix of sectors*).

for $\boldsymbol{X}$ a rectangular matrix with i.i.d. standard normal entries over a field $\mathbb{F}$ (assumed $\mathbb{R}$ if not otherwise stated). The number of columns $r$ of $\boldsymbol{X}$ is called the *degrees of freedom* of $\boldsymbol{W}$ in analogy with the degrees of freedom of a $\chi^2$-distributed random variable; indeed, the diagonal entries of $\boldsymbol{W}$ are (up to a constant scaling) i.i.d. $\chi^2$-distributed with $\beta r$ degrees of freedom.

Our main result is as follows, stated informally here with a formal statement deferred to Appendix B.2:

**Theorem 1** (Quantum neural networks are Wishart processes, informal). *Consider a QNN with associated JAWS as in Sec. 3, initialized approximately uniformly at random. Let $\ell^*$ be the optimum of the loss and $\overline{o}_\alpha$ the mean eigenvalue of $\boldsymbol{O}^\alpha$. Then, as $\dim(\mathcal{A}) \to \infty$, there is a convergence in joint distribution over $\boldsymbol{\rho}$ at any $\boldsymbol{\theta}$:*

$$\ell(\boldsymbol{\rho}; \boldsymbol{\theta}) - \ell^* \leadsto \sum_\alpha \frac{I_\alpha \overline{o}_\alpha}{r_\alpha} \operatorname{Tr}_\alpha (\boldsymbol{\rho}^\alpha \boldsymbol{W}_\alpha). \tag{18}$$

*The $\boldsymbol{W}_\alpha$ are each independent Wishart-distributed random matrices in the defining representation of $\mathcal{A}_\alpha$ with*

$$r_\alpha = \left\lfloor \frac{\operatorname{Tr}_\alpha(\boldsymbol{O}^\alpha)^2}{\operatorname{Tr}_\alpha\left((\boldsymbol{O}^\alpha)^2\right)} \right\rceil \tag{19}$$

*degrees of freedom, where $\lfloor \cdot \rceil$ denotes rounding to the nearest integer.*

An illustration of this distribution is provided in Figure 1(a). The proof follows by using the fact that the marginal distributions of matrix elements of the ansatz are approximately Gaussian distributed; this is made quantitative by bounding the errors in the associated joint characteristic functions over the inputs $\boldsymbol{\rho}$. This is enough to demonstrate convergence of $\ell(\boldsymbol{\rho}; \boldsymbol{\theta}) - \ell^*$ to a convolution of Wishart processes. We then prove a Welch–Satterthwaite-like result to demonstrate an asymptotic convergence of this convolution to a Wishart process with degrees of freedom parameters given by Eq. (19). The full proof is given in Appendix D. We there also prove a strengthened result—convergence *pointwise* in the corresponding probability densities—under certain additional assumptions on the initialization of the network.

We not only prove the convergence of the QNN loss function $\ell$ to that of a Wishart process, but also $\ell$, its gradient, and Hessian jointly. In doing so we are able to show that the *loss landscape* of QNNs—not just the landscape at a single point in parameter space, or averaged over it—converges to that of a Wishart process. We here only give an informal statement of the result in the special case when each $\boldsymbol{\rho}^\alpha$ is rank-1, but give the full distribution and formal statement in Appendix B.2 with proof in Appendix D. This distribution when conditioned on a nonzero value for the loss is illustrated in Figure 1(b).

**Theorem 2** (Gradient distribution, informal)**.** *Consider the setting of Theorem 1. Assume each $\boldsymbol{\rho}^\alpha$ is rank-1 in its defining representation. Let $\sigma_\alpha$ be the standard deviation of the eigenvalues of $\boldsymbol{O}^\alpha$. Then, as $\dim(\mathcal{A}) \to \infty$, the joint distribution of $\partial_{\theta_i} \ell(\boldsymbol{\rho}; \boldsymbol{\theta})$ conditioned on all $\ell(\boldsymbol{\rho}^\alpha; \boldsymbol{\theta}) = z_\alpha$ asymptotically converges to the joint distribution of*

$$\hat{\ell}_i \equiv \sum_\alpha \frac{2 I_\alpha \sigma_\alpha \operatorname{Tr}_\alpha(\boldsymbol{\rho}^\alpha)}{N_\alpha} \sqrt{\frac{\beta_\alpha z_\alpha}{I_\alpha \overline{o}_\alpha}} G_{\alpha,i} \chi_{\alpha,i}. \tag{20}$$

*Here, the $\chi_{\alpha,i}$ are independent $\chi$-distributed random variables with $\max(2, \beta_\alpha)$ degrees of freedom and the $G_{\alpha,i}$ are i.i.d. standard normal random variables.*

We also give an expression for the Hessian distribution at critical points. We informally report the result when each $\boldsymbol{\rho}^\alpha$ is rank-1 and the spectrum of each $\boldsymbol{O}^\alpha$ is sufficiently concentrated around its mean; this latter condition is typical of sums of low-weight fermionic (Feng et al., 2019) and local spin operators (Erdős & Schröder, 2014). The full distribution and formal statement is once again given in Appendix B.2 with proof in Appendix D.

**Theorem 3** (Hessian distribution, informal)**.** *Consider the setting of Theorem 2. Assume as well that $\frac{\sigma_\alpha}{\overline{o}_\alpha} \to 0$ for all $\alpha$. Then, as $\dim(\mathcal{A}) \to \infty$, the joint distribution of $\partial_{\theta_i} \partial_{\theta_j} \ell(\boldsymbol{\rho}; \boldsymbol{\theta})$ $(i \geq j)$ conditioned on all $\ell(\boldsymbol{\rho}^\alpha; \boldsymbol{\theta}) = z_\alpha$ and all $\partial_{\theta_i} \ell(\boldsymbol{\rho}^\alpha; \boldsymbol{\theta}) = 0$ asymptotically converges to the joint distribution of*

$$\hat{\ell}_{i,j} \equiv \sum_\alpha \frac{2 I_\alpha \sigma_\alpha \operatorname{Tr}_\alpha(\boldsymbol{\rho}^\alpha)}{N_\alpha^2} \sqrt{\frac{z_\alpha}{I_\alpha \overline{o}_\alpha}} G_{\alpha,i} \chi_{\alpha,j} W_{\alpha,i,j}. \tag{21}$$

*Here, the $\chi_{\alpha,i}$ are independent $\chi$-distributed random variables with $\max(2, \beta_\alpha)$ degrees of freedom and the $G_{\alpha,i}$ are i.i.d. standard normal random variables. The $\boldsymbol{W}_\alpha$ are independent Wishart-distributed random matrices with $\beta_\alpha r_\alpha$ degrees of freedom.*

Both of these derivative results follow by carefully considering the joint distribution of elements of a Wishart-distributed random matrix via the use of the Bartlett decomposition of Wishart-distributed matrices. Further simplifications follow by bounding the error in probability between various products of these marginal elements to simpler distributions.

## 5 NEW RESULTS IN LANDSCAPE THEORY FROM THE JAWS FRAMEWORK

With the full asymptotic distribution of QNN loss landscapes in hand, we are now able to prove several novel results in QNN loss landscape theory.

### 5.1 BARREN PLATEAUS

The first implication of our results that we will discuss is the unification of barren plateau results (Fontana et al., 2024; Ragone et al., 2024) in the limit of large Jordan algebra dimension.

**Corollary 4** (General expression for the loss function variance, informal)**.** *Let $\ell$ be as in Theorem 1. The variance of the loss over the initialization is:*

$$\operatorname{Var}_{\boldsymbol{\theta}}[\ell(\boldsymbol{\rho}; \boldsymbol{\theta})] = \sum_\alpha \frac{\operatorname{Tr}\left((\boldsymbol{O}^\alpha)^2\right) \operatorname{Tr}\left((\boldsymbol{\rho}^\alpha)^2\right)}{\dim_{\mathbb{R}}(\operatorname{Aut}(\mathcal{A}_\alpha))}. \tag{22}$$

This follows immediately from the variance of elements of Wishart-distributed random matrices; see Appendix C.1 for an explicit calculation. Despite the simplicity of the proof (given Theorem 1), this result generalizes Theorem 1 of Ragone et al. (2024), which only considered when either $i\boldsymbol{O}$ or $i\boldsymbol{\rho}$ was in the algebra generating $\operatorname{Aut}(\mathcal{A}_\alpha)$. However, our result holds for *all* variational ansatzes due to our general Jordan algebraic formulation of QNNs. Intriguingly, $\operatorname{Tr}\left((\boldsymbol{\rho}^\alpha)^2\right)$ can be thought of as probing the *generalized entanglement* (Barnum et al., 2004) of $\boldsymbol{\rho}$ with respect to the Jordan algebraic structure of $\mathcal{A}$. That is, entanglement induces barren plateaus, as previously seen in a nonalgebraic setting by Ortiz Marrero et al. (2021). A similar phenomenon was also previously noted by Ragone et al. (2024) with respect to the Lie algebraic structure of LASAs.

## 5.2 The Quantum Neural Tangent Kernel

We now connect our results to the quantum neural tangent kernel (QNTK) literature (Liu et al., 2022; 2023; You et al., 2022; García-Martín et al., 2023; Girardi & Palma, 2024; García-Martín et al., 2024). The landmark result in this field is that, in certain settings, QNN loss functions are asymptotically Gaussian processes. However, this same body of work has noted that such a Gaussian process description cannot generally hold. For instance, if the objective observable $\boldsymbol{O}$ is rank-1 and the algebra $\mathcal{A}$ associated with the QNN is the space of complex Hermitian matrices, it is known that the loss is proportional to an exponentially-distributed random variable (Boixo et al., 2018).

Our results can be seen as a unifying model of neural network loss landscapes, including both when convergence to a Gaussian process is achieved and when it is not. Recall that Theorem 1 demonstrated the asymptotic expression for the QNN loss at any $\boldsymbol{\theta}$ (left implicit) to be:

$$\hat{\ell}(\boldsymbol{\rho}) = \sum_{\alpha} \frac{I_\alpha \overline{o}_\alpha}{r_\alpha} \operatorname{Tr}_\alpha (\boldsymbol{\rho}^\alpha \boldsymbol{W}_\alpha). \tag{23}$$

This correctly captures the exponential behavior when $\mathcal{A}$ is the space of complex Hermitian matrices and $\operatorname{rank}(\boldsymbol{O}) = r = 1$. This is because the diagonal entries of such a complex Wishart matrix are (up to a constant) $\chi^2$-distributed with two degrees of freedom each, which is identical to an exponential distribution.

Indeed, our more general result can be used to *exactly* characterize when the loss functions of QNNs asymptotically form Gaussian processes. First, we consider normalizing $\hat{\ell}$ by some $\mathcal{N} \geq 1$ such that $\mathcal{N}\hat{\ell}$ has nonvanishing variance asymptotically. Under this normalization, convergence to a Gaussian process occurs when $\mathcal{N}\hat{\ell}$ has higher-order cumulants asymptotically vanishing. In other words, using $\kappa_i(\cdot)$ to denote the $i$th cumulant and $\mathcal{R}$ to denote the data set, this occurs when (see Appendix C.2 for an explicit calculation):

$$\max_{\substack{\boldsymbol{\rho} \in \mathcal{R} \\ i > 2}} \kappa_i \left( \mathcal{N}\hat{\ell}(\boldsymbol{\rho}) \right) \sim \max_{\substack{\boldsymbol{\rho} \in \mathcal{R} \\ \alpha}} \frac{\mathcal{N}^3 I_\alpha^3 \overline{o}_\alpha^3 \operatorname{Tr}_\alpha \left( (\boldsymbol{\rho}^\alpha)^3 \right)}{r_\alpha^2} = o(1). \tag{24}$$

We thus can state this pair of conditions as follows.

**Corollary 5** (Exact conditions for convergence to a Gaussian process, informal). *Let $\ell$ be as in Theorem 1. $\mathcal{N}\ell(\boldsymbol{\rho})$ is asymptotically a Gaussian process over $\boldsymbol{\rho} \in \mathcal{R}$ if and only if Eq. (24) is satisfied and $\mathcal{N}^2 \operatorname{Var}[\ell(\boldsymbol{\rho})]$ (as in Corollary 4) is nonvanishing for all $\boldsymbol{\rho} \in \mathcal{R}$.*

*When these conditions are met, the covariance is given by:*

$$\mathcal{K}(\boldsymbol{\rho}, \boldsymbol{\rho}') = \lim_{N \to \infty} \mathcal{K}_N(\boldsymbol{\rho}, \boldsymbol{\rho}')$$

$$= \lim_{N \to \infty} \mathcal{N}^2 \sum_\alpha \frac{\beta_\alpha \operatorname{Tr}\left( (\boldsymbol{O}^\alpha)^2 \right)}{2 \dim_{\mathbb{R}}(\operatorname{Aut}(\mathcal{A}_\alpha)) r_\alpha} I_\alpha \mathbb{E}\left[ \operatorname{Tr}_\alpha (\boldsymbol{\rho}^\alpha (\boldsymbol{W}_\alpha - r_\alpha)) \operatorname{Tr}_\alpha (\boldsymbol{\rho}'^\alpha (\boldsymbol{W}_\alpha - r_\alpha)) \right]. \tag{25}$$

It is now easy to see why (for simple $\mathcal{A}$) the $r = 1$ case does not converge to a Gaussian process: in this setting $\mathcal{N}$ must be $o(N)$ such that Eq. (24) is satisfied and higher-order cumulants vanish, but for such a choice the variance is vanishing.

In Appendix C.2 we are also more explicit on the form of Eq. (25). We use known results for the mixed cumulants of Wishart matrix elements to show that the covariance can be written as an explicit, quadratic expression in $\boldsymbol{\rho}^\alpha$ and $\boldsymbol{\rho}'^\alpha$, depending only on $\mathcal{A}$. We further show in Appendix C.2 that, when the $\boldsymbol{\rho} \in \mathcal{R}$ can be mutually diagonalized,

$$\mathcal{K}_N(\boldsymbol{\rho}, \boldsymbol{\rho}') = \mathcal{N}^2 \sum_\alpha \frac{\operatorname{Tr}\left( (\boldsymbol{O}^\alpha)^2 \right)}{\dim_{\mathbb{R}}(\operatorname{Aut}(\mathcal{A}_\alpha))} \operatorname{Tr}(\boldsymbol{\rho}^\alpha \boldsymbol{\rho}'^\alpha). \tag{26}$$

This form of the covariance can then be immediately fed into neural tangent kernel results to reason about the training behavior (Neal, 1996; Choromanska et al., 2015; Chaudhari, 2018; Lee et al.,

2018) and generalization ability (Jacot et al., 2018; Wei et al., 2022) of such networks. For instance, by the results of Girardi & Palma (2024), this covariance suggests that gradient descent does not reach the global minimum in time polynomial in the model size for unstructured QNNs. This is discussed in more detail in Appendix C.2.

## 5.3 LOCAL MINIMA

We end by examining the distribution of local minima of the loss landscape. This has been done previously in the more restricted setting where no structure is imposed on the ansatz or loss function outside of locality (Anschuetz, 2022; Anschuetz & Kiani, 2022). In this section we consider a general QNN with $p$ trained parameters under the conditions of Theorem 3, as well as assume that all

$$\gamma_\alpha \equiv \frac{p}{\beta_\alpha r_\alpha} \tag{27}$$

are held constant as we take the asymptotic limit $\dim(\mathcal{A}) \to \infty$. We call the $\gamma_\alpha$ the *overparameterization ratios* of a given QNN architecture, associated with the various simple components of the Jordan algebra $\mathcal{A}$ in the JAWS description of the network.

We calculate the distribution of local minima using the *Kac–Rice formula* (Adler & Taylor, 2007), which gives the expected density of local minima of a random field $\ell$ at a function value $z$. Anschuetz (2022) demonstrated that the assumptions for the Kac–Rice formula are satisfied for variational loss landscapes, and when rotationally invariant on the $p$-torus the formula takes the form:

$$\mathbb{E}\left[\mathrm{Crt}_0\left(z\right)\right] = (2\pi)^p \, \mathbb{E}\left[\det\left(\boldsymbol{H}_z\right) \mathbf{1}\left\{\boldsymbol{H}_z \succeq \boldsymbol{0}\right\}\right] \mathbb{P}\left[\boldsymbol{G}_z = \boldsymbol{0}\right] \mathbb{P}\left[\ell = z\right]. \tag{28}$$

Here, $\mathbb{E}\left[\mathrm{Crt}_0\left(z\right)\right]$ is the expected density of local minima at a function value $z$, $\boldsymbol{G}_z$ is the gradient conditioned on $\ell = z$, and $\boldsymbol{H}_z$ is the Hessian conditioned on $\ell = z$ and $\boldsymbol{G} = \boldsymbol{0}$. In a slight abuse of notation, here $\mathbb{P}\left[\cdot\right]$ denotes the probability density associated with the event $\cdot$.

We can evaluate this expression using Theorems 1, 2, and 3. To simplify the expression here, we assume the addition of a regularization term of the form:

$$R_L\left(\boldsymbol{\theta}\right) = L \left\|\boldsymbol{\theta} - \boldsymbol{C}\right\|_2^2 \tag{29}$$

to the loss; we also present here only the relative density rather than the total number of minima at a loss function value $z$. The full expression counting local minima—both with and without the regularization of Eq. (29)—are described in detail in Appendix C.3.

**Corollary 6** (Density of local minima, informal). *Consider the setting of Theorem 3 with an additional regularization term of the form of Eq.* (29). *The density of local minima at a loss function value $z > 0$ is, to multiplicative leading order, given by the convolution over all $\alpha$ with $\gamma_\alpha < 1$:*

$$\kappa\left(z\right) = \operatorname*{\text{\Large$*$}}_{\alpha:\gamma_\alpha<1} f_\Gamma\left(\frac{z}{\overline{o}_\alpha \operatorname{Tr}\left(\boldsymbol{\rho}^\alpha\right)}; \frac{\beta_\alpha r_\alpha}{2}, \frac{2}{\beta_\alpha r_\alpha}\right). \tag{30}$$

*Here, $f_\Gamma\left(\cdot; k, \theta\right)$ denotes the gamma distribution with shape $k$ and scale $\theta$ parameters:*

$$f_\Gamma\left(x; k, \theta\right) = \frac{1}{\Gamma\left(k\right)\theta^k} x^{k-1} \exp\left(-\frac{x}{\theta}\right). \tag{31}$$

Examples of this distribution are plotted in Figure 1(c), where we give a finite width to the density in the *overparameterized regime*—where all $\gamma_\alpha \geq 1$—to represent potential finite-size effects. Intriguingly, the variance of this distribution when all $\gamma_\alpha < 1$—the *underparameterized regime*—corresponds to the variance of the loss function itself, i.e., as given in Corollary 4. However, due to the exponential tails of the gamma distribution, even when there are no barren plateaus in the loss landscape there is only an exponentially small fraction of local minima in the vicinity of the global minimum in this regime. That is, both barren plateaus and poor local minima are potential obstructions to efficient trainability that must be taken into account in the design of practical QNNs.

## 6 CONCLUSION

Taken together, our results show that the natural model for quantum neural networks is not one of Gaussian processes obeying a quantum neural tangent kernel, but rather one of *Wishart processes*.

This Wishart process model unifies all of the recent major thrusts in calculating properties of quantum neural network loss landscapes. Indeed, our results allow us to propose a simple operational definition for the "trainability" of quantum neural networks, which to date has been a term used heuristically without any formal definition.[4]

**Definition 7** (Trainability of quantum neural networks). Consider a QNN with $p$ trained parameters composed of $N \times N$ unitary matrices. Let $\mathcal{A} \cong \bigoplus_\alpha \mathcal{A}_\alpha$ be the corresponding Jordan algebra as in Sec. 3. Define the *degrees of freedom* parameters:

$$r_\alpha \equiv \frac{\mathrm{Tr}_\alpha \left(\boldsymbol{O}^\alpha\right)^2}{\mathrm{Tr}_\alpha \left(\left(\boldsymbol{O}^\alpha\right)^2\right)}. \tag{32}$$

We say that the QNN is *trainable* if and only if, as $N \to \infty$, the QNN satisfies:

1. *Absence of barren plateaus* (Corollary 4):

$$\sum_\alpha \frac{\mathrm{Tr}\left(\left(\boldsymbol{O}^\alpha\right)^2\right) \mathrm{Tr}\left(\left(\boldsymbol{\rho}^\alpha\right)^2\right)}{\dim_{\mathbb{R}}\left(\mathrm{Aut}\left(\mathcal{A}_\alpha\right)\right)} = \Omega\left(\mathrm{poly}\left(\log\left(N\right)\right)^{-1}\right). \tag{33}$$

2. *Absence of poor local minima* (Corollary 6):

$$p \geq \max_\alpha \beta_\alpha r_\alpha. \tag{34}$$

Given knowledge of $\mathcal{A}$, this yields a quantum algorithm for determining the asymptotic trainability of a QNN architecture: one need only measure $\mathrm{Tr}\left(\left(\boldsymbol{O}^\alpha\right)^2\right)$ and $\mathrm{Tr}\left(\left(\boldsymbol{\rho}^\alpha\right)^2\right)$ on a quantum computer. This can be done through the measurement of basis elements of $\mathcal{A}$ given copies of $\boldsymbol{\rho}$ and, for the former, block encodings of $\boldsymbol{O}$ as prepared by the standard linear combination of unitaries subroutine (Childs & Wiebe, 2012). We provide explicit error bounds for this procedure in Appendix E.

Practically, where does this leave quantum neural networks? For one, it seems unlikely that there exists any computational quantum advantage during the training of QNNs outside of HHL-like speedups (Harrow et al., 2009; Biamonte et al., 2017) and training algorithms that leverage some existing knowledge of the data to be learned (Liu et al., 2021; Hastings & O'Donnell, 2022; Huang et al., 2024). This is due to efficient classical simulation algorithms for quantum systems algebraically constrained to explore a low-dimensional subspace (Anschuetz et al., 2023a; Goh et al., 2023), as is required for efficient trainability per Definition 7. This was noted in Cerezo et al. (2023) for deep QNNs, where the authors postulated that deep QNNs exhibiting no barren plateaus are classically simulable (outside of certain special cases). Our results demonstrate a similar phenomenon for *shallow* QNNs: poor local minima in polynomially-sized circuits can only be avoided when the effective Hilbert space dimension grows at most polynomially quickly with the system size. This leaves the space for a practical, superpolynomial quantum advantage when training with a problem-agnostic algorithm such as gradient descent even narrower than previously believed: a veritable Amity Island in a sea of negative results. One ray of hope is the known existence of *polynomial* quantum advantages during inference in such a setting (Anschuetz et al., 2023b; Anschuetz & Gao, 2024). Intelligent warm-starting of the optimization procedure may be another way to circumvent poor training (Puig-i-Valls et al., 2024), though more must be done to fully understand how precise warm-starting must be for training to be efficient. Given the exact asymptotic form of the loss landscape we give here, such an analysis may be possible in the future.

Our work gives a unified understanding of quantum neural networks as Wishart processes. Great strides have been made in the classical machine learning literature in understanding the training dynamics (Choromanska et al., 2015; Chaudhari, 2018; Jacot et al., 2018; Allen-Zhu et al., 2019) and generalization behavior (Jacot et al., 2018; Wei et al., 2022) of classical neural networks via their connections to Gaussian processes, which unfortunately only port over in the specific settings where the Wishart process itself approaches a Gaussian process. Our results encourage an understanding of how specific properties of Wishart processes, not just Gaussian processes, influence the learning behavior of quantum networks in order to more fully grasp how quantum neural networks learn.

---

[4]The concept of "efficient learning" has been previously studied (Gil-Fuster et al., 2024), but we give the first *operational* definition in terms of the structural properties of a given QNN architecture.

REPRODUCIBILITY STATEMENT

The main results discussed in Secs. 3 and 4 are formally stated—with an explicit listing of formal assumptions—in Appendix B. Definitions and background necessary for these formal theorem and assumption statements are laid out in Appendix A. Formal proofs of the main results are given in Appendix D. Formal statements and proofs of the corollaries discussed in Sec. 5 are given in Appendix C. A proof of the claim made in Sec. 6 that one can efficiently estimate the quantities in Definition 7 using an LCU block encoding is given in Appendix E. A discussion of the settings where our results do not hold due to a violation of our formal assumptions is given in Appendix F. The remainder of the Appendices prove claims and helper lemmas used in Appendices B, C, and D.

ACKNOWLEDGMENTS

E.R.A. is grateful to Pablo Bermejo, Marco Cerezo, Diego García-Martín, Bobak T. Kiani, Martín Larocca, Thomas Schuster, and Alissa Wilms for enlightening discussion and suggestions that aided in the preparation of this manuscript. E.R.A. also thanks Nathan Wiebe for advocating for a formal definition of trainability in quantum neural networks at the PennyLane Research Retreat of 2023, which inspired parts of this work. E.R.A. was funded in part by the Walter Burke Institute for Theoretical Physics at Caltech.

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

# A  PRELIMINARIES FOR FORMAL DISCUSSION OF RESULTS

We begin by reviewing concepts that we will use in proving our results. We also give a summary of the notation we use throughout in Table 2.

## A.1  QUANTUM NEURAL NETWORKS

We first review *quantum neural networks* (QNNs). These are defined by a parameterized *ansatz*

$$\boldsymbol{U}\left(\boldsymbol{\theta}\right) = \prod_{i=p}^{1} \exp\left(-\mathrm{i}\theta_i \boldsymbol{A}_i\right), \tag{35}$$

which belongs to some path-connected subgroup of the unitary group $\mathrm{U}\left(N\right)$. Though $\boldsymbol{U}\left(\boldsymbol{\theta}\right)$ can in principle parameterize all of $\mathrm{U}\left(N\right)$, it is often taken to instead parameterize some path-connected subgroup $G \subseteq \mathrm{U}\left(N\right)$. This might be due to enforcing some global structure on the model (Meyer et al., 2023), or can be a model of the finiteness of the reverse light cones of shallow quantum circuits (Anschuetz & Kiani, 2022).

Given a choice of ansatz, the goal of a QNN is to minimize an empirical risk of the form:

$$f\left(\boldsymbol{\theta}\right) = \frac{1}{|\mathcal{R}|} \sum_{\boldsymbol{\rho}\in\mathcal{R}} \ell\left(\boldsymbol{\theta}; \boldsymbol{\rho}\right) = \frac{1}{|\mathcal{R}|} \sum_{\boldsymbol{\rho}\in\mathcal{R}} \mathrm{Tr}\left(\boldsymbol{U}\left(\boldsymbol{\theta}\right) \boldsymbol{\rho} \boldsymbol{U}\left(\boldsymbol{\theta}\right)^{\dagger} \boldsymbol{O}\right). \tag{36}$$

Here, $\mathcal{R}$ can be thought of a data set comprising multiple input states $\boldsymbol{\rho}$, and $\ell$ the loss function. Historically, when $|\mathcal{R}| = 1$ QNNs have been referred to as *variational quantum algorithms* (VQAs) (Peruzzo et al., 2014) due to their connection to finding variational approximations to the ground states of quantum Hamiltonians. There are known quantum-classical separations for the expressivity of quantum neural networks even when taking into account the requirement that the training procedure is efficient (Liu et al., 2021; Hastings & O'Donnell, 2022; Huang et al., 2024), though they require very specific training algorithms that take advantage of the structure of the data.

There has been recent hope that, with enough ansatz structure, efficient training may follow just via a simple application of gradient descent. This follows from the *Lie algebra-supported ansatz* (LASA) literature, where it has been shown that if the generators $\mathrm{i}\boldsymbol{A}_i$ of the ansatz $\boldsymbol{U}$ as well as the (scaled) objective observable $\mathrm{i}\boldsymbol{O}$ belong to the same Lie algebra $\mathfrak{g}$—called the *dynamical Lie algebra* (Larocca et al., 2022)—generating $G$, gradients scale inversely with the dimension of $\mathfrak{g}$ (Fontana et al., 2024; Ragone et al., 2024). It is also conjectured that when $\dim\left(\mathfrak{g}\right)$ scales polynomially with the system size there exist polynomial-depth ansatzes that do not have poor local minima, which together with the large gradients would imply efficient trainability of these loss functions via gradient descent. Conditioned on this conjecture there have been results demonstrating expressivity separations in quantum machine learning where the QNN is efficiently trainable through a simple application of gradient descent (Anschuetz et al., 2023b; Anschuetz & Gao, 2024).

Though the LASA framework gives sufficient conditions for loss functions to have large gradients, it is known that they are not necessary. One example of this was demonstrated in Diaz et al. (2023), where it was shown that parameterized matchgate circuits with an objective observable given by constant-degree polynomials in Majorana fermions are efficiently trainable though they are not a part of the LASA setting. We claim that both settings are special cases of a Jordan algebraic understanding of variational loss functions. In preparation of discussing this connection we now review Jordan algebras.

| | |
|---|---|
| $\mathbb{N}_0$ | Natural numbers including 0 |
| $\mathbb{N}_1$ | Natural numbers excluding 0 |
| $[n]$ | Natural numbers from 1 through $n$ |
| $\mathcal{H}^n(\mathbb{F})$ | Jordan algebra of $n \times n$ Hermitian matrices over $\mathbb{F}$ |
| $\mathbb{F}$ | Field, here one of $\mathbb{R}, \mathbb{C}, \mathbb{H}$ |
| $\beta$ | $\beta = 1, 2, 4$ when associated field $\mathbb{F} = \mathbb{R}, \mathbb{C}, \mathbb{H}$, respectively |
| $\dim_{\mathbb{F}}(\cdot)$ | Dimension of $\cdot$ as a vector space over $\mathbb{F}$ |
| $\mathcal{N}^\beta(0, 1)$ | Standard normal distribution over $\mathbb{F}$ (given by $\beta$) |
| $\mathcal{W}_n^\beta(r, \boldsymbol{\Sigma})$ | $\beta$-Wishart distribution of $n \times n$ matrices with $r$ degrees of freedom and scale matrix $\boldsymbol{\Sigma}$ |
| $\rightsquigarrow$ | Convergence in distribution |
| $\xrightarrow{p}$ | Convergence in probability |
| $\odot$ | Hadamard product |
| $\mathcal{A}_\alpha$ | Simple components of semisimple Jordan algebra $\mathcal{A}$ |
| $G_\alpha$ | Lie group isomorphic to a connected component of $\mathrm{Aut}(\mathcal{A}_\alpha)$ |
| $\mathfrak{g}_\alpha$ | Lie algebra generating $G_\alpha$ |
| $\cdot^\alpha$ | Defining representation of the projection of $\cdot$ into $\mathcal{A}_\alpha$ |
| $N_\alpha$ | Dimension of vector space on which the defining representation of $\mathcal{A}$ acts |
| $N$ | Sum of $N_\alpha$ |
| $\mathrm{Tr}_\alpha(\cdot)$ | Trace of $\cdot$ in the defining representation of $\mathcal{A}_\alpha$ |
| $\mathrm{Tr}(\cdot)$ | Trace of $\cdot$ according to its representation in $\mathcal{H}^N(\mathbb{C}) \supseteq \mathcal{A}$ |
| $\|\cdot\|_{\mathrm{op}}$ | Operator norm of $\cdot \in \mathcal{A}_\alpha$ in its defining representation |
| $\|\cdot\|_*$ | Trace (nuclear) norm of $\cdot \in \mathcal{A}_\alpha$ in its defining representation |
| $\|\cdot\|_{\mathrm{F}}$ | Frobenius norm of $\cdot \in \mathcal{A}_\alpha$ in its defining representation |
| $\mathrm{O}(\mathrm{poly}(N))$ | $\mathrm{O}(N^\kappa)$ for some constant $\kappa > 0$ |
| WLOG | without loss of generality |
| w.h.p. | with high probability |

Table 2: **Table of notation.** Notation used in the presentation of our results are given in the left column with corresponding meaning given in the right column. We also note here that exponents written as decimals throughout our results are chosen somewhat arbitrarily, i.e., can be improved by any sufficiently small constant $\epsilon > 0$ in the exponent. Finally, we note that we will often forgo writing "...a sequence of [ansatzes, loss functions, observables, QNNs, ...]..." when describing our asymptotic convergence results for brevity.

## A.2 JORDAN ALGEBRAS

A *Jordan algebra* over the reals is formally a real vector space $V$ with a commutative multiplication operation $\circ$ acting on $u, v \in V$ satisfying the *Jordan identity*:

$$u \circ ((u \circ u) \circ v) = (u \circ u) \circ (u \circ v), \tag{37}$$

which ensures the associativity of the power. A simple example of a Jordan algebra over the reals is the real algebra $\mathcal{H}^N(\mathbb{C})$ of $N \times N$ complex-valued Hermitian matrices with $\circ$ given by half the anticommutator. In particular, for any finite-dimensional quantum system both $\boldsymbol{O}$ and the inputs $\boldsymbol{\rho}$ in Eq. (36) belong to $\mathcal{H}^N(\mathbb{C})$. We emphasize that though this algebra is typically written in terms of complex matrices it is still a *real* Jordan algebra. This is because, for instance, $\mathrm{i} \notin \mathbb{R}$ multiplying a Hermitian matrix is no longer Hermitian. In fact, the Jordan algebra of Hermitian matrices is *Euclidean* (or *formally real*) as it satisfies the defining property (Koecher, 1999c):

$$\forall A, B \in \mathcal{A}, \ A \circ A + B \circ B = 0 \iff A = B = 0. \tag{38}$$

It is apparent that all subalgebras of a formally real algebra are also formally real.

The real vector space $V$ with which a Jordan algebra is associated also has a natural linear transformation $L(u)$:

$$L(u)v \equiv u \circ v, \tag{39}$$

which can be viewed as the Jordan algebraic analogue of the adjoint representation of Lie algebras. This linear transformation gives rise to the *canonical trace form*:

$$\tau(u, v) \equiv \mathrm{Tr}(L(u \circ v)). \tag{40}$$

When $\tau$ is nonsingular we call the associated Jordan algebra $\mathcal{A}$ *semisimple*. As an example, for the algebra $\mathcal{H}^N(\mathbb{F})$ of Hermitian matrices over the field $\mathbb{F}$, $\tau(u, v)$ is just the Frobenius inner product between $u$ and $v$. Just as is the case for semisimple Lie algebras, all symmetric bilinear forms on a semisimple Jordan algebra are identical up to an overall scaling. Specifically, for any symmetric bilinear form $\sigma$, there exists an element $z$ in the center of $\mathcal{A}$ such that (Theorem 10, Koecher (1999a)):

$$\sigma(u, v) = \tau(z \circ u, v). \tag{41}$$

For $\mathcal{A} = \mathcal{H}^N(\mathbb{C})$ in the defining representation the center is just real multiples of the identity matrix, i.e., there exists a real $I \geq 0$ such that:

$$I\sigma(u, v) = \tau(u, v). \tag{42}$$

We call $I$ the *index* of $\sigma$ in analogy with Fontana et al. (2024) for Lie algebras.

Though perhaps not as famous as the classification of compact Lie groups, the semisimple Euclidean Jordan algebras have also been classified.

**Theorem 8** (Classification of semisimple Euclidean Jordan algebras (Koecher, 1999d)). *Any semisimple Euclidean Jordan algebra is isomorphic to a direct sum of the simple Euclidean Jordan algebras:*

- $\mathcal{L}_N$ *for $N \geq 1$: the* spin factor, *with vector space equal to $\mathbb{R}^N$ and $\circ$ given by the operation*

$$\boldsymbol{x} \circ \boldsymbol{y} = (\boldsymbol{x} \cdot \boldsymbol{y}, x_1 \overline{\boldsymbol{y}} + y_1 \overline{\boldsymbol{x}}), \tag{43}$$

  *where $\overline{\cdot}$ denotes $\cdot$ with the first coordinate projected out;*

- $\mathcal{H}^N(\mathbb{R})$ *for $N \geq 3$: $N \times N$ symmetric matrices over $\mathbb{R}$, with $\circ$ half the anticommutator;*

- $\mathcal{H}^N(\mathbb{C})$ *for $N \geq 3$: $N \times N$ Hermitian matrices over $\mathbb{C}$, with $\circ$ half the anticommutator;*

- $\mathcal{H}^N(\mathbb{H})$ *for $N \geq 3$: $N \times N$ Hermitian matrices over $\mathbb{H}$, with $\circ$ half the anticommutator;*

- $\mathcal{H}^3(\mathbb{O})$: $3 \times 3$ *Hermitian matrices over $\mathbb{O}$, with $\circ$ half the anticommutator.*

We use the term *defining representation* to speak of the described matrix representations of these algebras. As the Hermitian octonion case is not an infinite family it is often called *exceptional*. We here are interested in asymptotic sequences of Jordan algebras so we will not be considering the exceptional case.

Every Jordan algebra $\mathcal{A}$ has associated with it an automorphism group $\text{Aut}(\mathcal{A})$. As an example, for $\mathcal{H}^N(\mathbb{R})$ in the defining representation this is just given by the action of conjugation (under the usual matrix multiplication) by orthogonal matrices. More generally, the nonexceptional simple cases have automorphism groups (Orlitzky, 2024):

- $\text{Aut}(\mathcal{L}_N)$: left-action of $\{1\} \times \text{O}(N-1)$;

- $\text{Aut}(\mathcal{H}^N(\mathbb{R}))$: conjugation action of $\text{PO}(N)$;

- $\text{Aut}(\mathcal{H}^N(\mathbb{C}))$: disjoint union of the conjugation action of $\text{PU}(N)$, and the transpose followed by the conjugation action of $\text{PU}(N)$;

- $\text{Aut}(\mathcal{H}^N(\mathbb{H}))$: conjugation action of $\text{PSp}(N)$.

We are here primarily interested in the connected component $\text{Aut}_1(\cdot) \subseteq \text{Aut}(\cdot)$ containing the identity transformation, so we will only concern ourselves with the path-connected automorphism subgroups:

- $\text{SO}(N-1) \cong \text{Aut}_1(\mathcal{L}_N) \subset \text{Aut}(\mathcal{L}_N)$;

- $\text{SO}(N) \cong \text{Aut}_1(\mathcal{H}^N(\mathbb{R})) \subseteq \text{Aut}(\mathcal{H}^N(\mathbb{R}))$;

- $\text{SU}(N) \cong \text{Aut}_1(\mathcal{H}^N(\mathbb{C})) \subset \text{Aut}(\mathcal{H}^N(\mathbb{C}))$;

- $\text{Sp}(N) \cong \text{Aut}_1(\mathcal{H}^N(\mathbb{H})) \subset \text{Aut}(\mathcal{H}^N(\mathbb{H}))$.

Surprisingly, the automorphism groups of semisimple Jordan algebras are also classified. Given a decomposition of a semisimple Euclidean Jordan algebra $\mathcal{A}$ into simple components:

$$\mathcal{A} \cong \bigoplus_\alpha \mathcal{A}_\alpha, \tag{44}$$

the connected component of $\mathrm{Aut}_1(\mathcal{A}) \subseteq \mathrm{Aut}(\mathcal{A})$ containing the identity is isomorphic to the direct product (Theorem 10, Koecher (1999b)):

$$\mathrm{Aut}_1(\mathcal{A}) \cong \bigtimes_\alpha \mathrm{Aut}_1(\mathcal{A}_\alpha). \tag{45}$$

### A.3 $\epsilon$-APPROXIMATE $t$-DESIGNS OVER $\mathrm{Aut}_1(\mathcal{A})$

In order to discuss the structure of the loss function in any detail we will need to consider some choice of randomness over loss functions. To achieve this we will use $\epsilon$-*approximate $t$-designs* over $\mathrm{Aut}_1(\mathcal{A})$. Our use of these designs formalizes the notion of approximate "independence" or "uniform initialization" of the ansatz with respect to the eigenbasis of a given objective observable, as when $\epsilon \to 0$ and $t \to \infty$ the ansatz is chosen in a completely group-invariant way over $\mathrm{Aut}_1(\mathcal{A})$. By Eq. (45) these designs are (up to isomorphism) direct products of $\epsilon$-approximate $t$-designs over $\mathrm{SO}(N)$, $\mathrm{SU}(N)$, and $\mathrm{Sp}(N)$.

Before defining $\epsilon$-approximate $t$-designs we define the Haar measure on compact Lie groups such as $\mathrm{Aut}_1(\mathcal{A})$. This is the unique group-invariant normalized measure on these compact Lie groups.

**Definition 9** (Haar measure on $G$ (Haar, 1933)). Let $G$ be a compact Lie group. The unique measure satisfying:

$$\int_G \mathrm{d}\mu = 1 \tag{46}$$

as well as

$$\mathrm{d}\mu = \mathrm{d}(g\mu) \tag{47}$$

for all $g \in G$ is the *Haar measure on $G$*. The existence and uniqueness of this measure follows from Haar (1933).

We now define $\epsilon$-approximate $t$-designs. We will here use the "trace norm definition" out of convenience, though all of the commonly used definitions are roughly equivalent (Harrow & Mehraban, 2023).

**Definition 10** ($\epsilon$-approximate $t$-designs over $G$). Let $G$ be a compact Lie group with defining representation over an $N$-dimensional space, and $\mu$ the Haar measure over $G$. A measure $\nu$ satisfying:

$$\sum_m |\mathbb{E}_{U\sim\mu}[m(U)] - \mathbb{E}_{U\sim\nu}[m(U)]| \leq \epsilon, \tag{48}$$

where the sum is over all degree-$(t,t)$ monomials $m(U)$ in the entries of $U$ and $U^*$, is an $\epsilon$-*approximate $t$-design over $G$*.

### A.4 WISHART MATRICES

Our main results will be given in terms of Wishart-distributed random matrices, so before proceeding we give a brief review of this distribution. We will use $\mathcal{W}_n^\beta(r, \boldsymbol{\Sigma})$ to denote the $\beta$-*Wishart distribution* (Dubbs et al., 2013), where:

- $\beta = 1, 2, 4$ indicates it is over a field $\mathbb{R}, \mathbb{C}, \mathbb{H}$, respectively;
- $r \in \mathbb{N}_1$ is the *degrees of freedom* parameter of the distribution;
- $\boldsymbol{\Sigma} \in \mathcal{H}^n(\mathbb{R})$ is the symmetric, real-valued, positive-definite *scale matrix* parameter of the distribution.

When we do not specify $\beta$ we implicitly are referring to the usual Wishart distribution where $\beta = 1$. The $\beta$-Wishart distribution is over positive semidefinite matrices of rank $r$, constructed by drawing i.i.d. standard Gaussian entries over $\mathbb{F}$:

$$X_{i,j} \sim \mathcal{N}^\beta(0, 1) \tag{49}$$

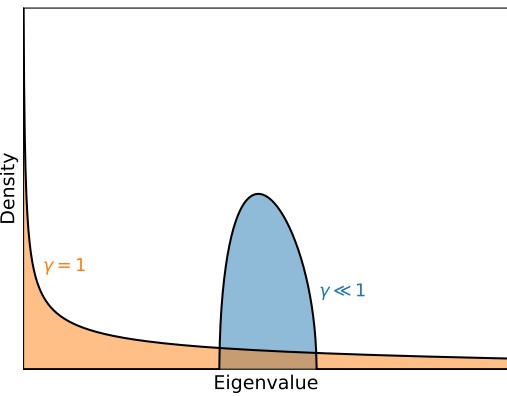

Figure 2: **Marčenko–Pastur distribution.** The density of the Marčenko–Pastur distribution—the asymptotic empirical eigenvalue distribution of normalized Wishart matrices—in the regime where $\gamma \ll 1$ and where $\gamma = 1$. At $\gamma = 1$ the associated Wishart matrix transitions from being full-rank to being low-rank.

for $i \in [n]$ and $j \in [r]$, and considering the distribution of:

$$\boldsymbol{W} = \sqrt{\boldsymbol{\Sigma}} \boldsymbol{X} \boldsymbol{X}^{\dagger} \sqrt{\boldsymbol{\Sigma}}. \tag{50}$$

Here there exists the so-called *Bartlett decomposition* for $\boldsymbol{W} \sim \mathcal{W}_n^\beta (r, \boldsymbol{I}_n)$:

$$\boldsymbol{W} \sim \sqrt{\boldsymbol{\Sigma}} \boldsymbol{L} \boldsymbol{L}^{\dagger} \sqrt{\boldsymbol{\Sigma}}, \tag{51}$$

where $\boldsymbol{L}$ is lower-triangular. When $r \geq n$, the the entries below the diagonal of $\boldsymbol{L}$ are i.i.d. $\mathcal{N}^\beta (0, 1)$-distributed and the diagonal entries are i.i.d. distributed as (Rouault, 2007):

$$\beta L_{i,i}^2 \sim \chi^2 \left( \beta \left( r - i + 1 \right) \right), \tag{52}$$

i.e., the $L_{i,i}$ are $\chi$-distributed up to an overall scaling of $\beta^{-\frac{1}{2}}$. When $r < n$, $\boldsymbol{L}$ is $n \times r$ with first $r$ rows as above and all other entries i.i.d. according to $\mathcal{N}^\beta (0, 1)$ (Srivastava, 2003; Li & Xue, 2010; Yu et al., 2014). This completely characterizes the marginal distribution of the entries of any $\boldsymbol{W} \sim \mathcal{W}_n^\beta (r, \boldsymbol{I}_n)$.

Our final results will be written in terms of real Wishart matrices $\boldsymbol{W}$, i.e., those with $\beta = 1$. When we analyze the asymptotic density of local minima of a JAWS it will turn out we will need to consider the asymptotic spectrum of a real Wishart matrix, which we now discuss. When the scale matrix $\boldsymbol{\Sigma}$ of $\boldsymbol{W}$ is the identity, as $n, r \to \infty$ with $\frac{n}{r} = \gamma$ held constant, the spectrum of $r^{-1} \boldsymbol{W}$ is known to almost surely converge weakly to a fixed distribution (Marčenko & Pastur, 1967). This fixed distribution is the *Marčenko–Pastur distribution*, given by

$$\frac{\mathrm{d}\mu_{\mathrm{MP}}^\gamma}{\mathrm{d}\lambda} = \left( 1 - \gamma^{-1} \right) \delta \left( \lambda \right) \mathbf{1} \left\{ \gamma \geq 1 \right\} + \frac{\sqrt{\left( \gamma_+ - \lambda \right) \left( \lambda - \gamma_- \right)}}{2\pi\gamma\lambda} \mathbf{1} \left\{ \gamma_- \leq \lambda \leq \gamma_+ \right\}, \tag{53}$$

where

$$\gamma_\pm \equiv \left( 1 \pm \sqrt{\gamma} \right)^2 \tag{54}$$

and $\mathbf{1}$ is the indicator function. This distribution is illustrated in Figure 2.

### A.5 CONVERGENCE OF RANDOM VARIABLES

We finally discuss in more detail the various of notions of convergence of random variables that we use throughout our paper. Given a sequence of a set $\left( X_i^N \right)_i$ of real-valued random variables as $N \to \infty$, we say $\left( X_i^N \right)_i$ *weakly converges* (or *converges in distribution*) to $(X_i^\infty)_i$ if the joint cumulative distribution function $F^N$ of the $X_i^N$ converges to the joint cumulative distribution function $F^\infty$ of the $X_i^\infty$ at every point at which $F^\infty$ is continuous. That is,

$$\lim_{N \to \infty} F^N \left( \boldsymbol{x} \right) = F^\infty \left( \boldsymbol{x} \right) \tag{55}$$

for all points $\boldsymbol{x}$ at which $F^\infty$ is continuous. We denote this at the level of random variables with the notation:

$$\left(X_i^N\right)_i \rightsquigarrow \left(X_i^\infty\right)_i. \tag{56}$$

One way to quantify the rate of this convergence is through the use of the *Lévy–Prokhorov metric*. Letting $p^N$ and $p^\infty$ be the densities associated with $F^N$ and $F^\infty$, respectively, the Lévy–Prokhorov metric is given by:

$$\pi\left(p^N, p^\infty\right) = \inf\left\{\epsilon > 0 \mid \forall A, F^N(A) \leq F^\infty(A^\epsilon) + \epsilon\right\}, \tag{57}$$

where here $A^\epsilon$ is an $\epsilon$-neighborhood in $\infty$-norm of $A$. $\pi\left(p^N, p^\infty\right) \to 0$ if and only if $\left(X_i^N\right)_i \rightsquigarrow \left(X_i^\infty\right)_i$.

Convergence in distribution is closely related to the pointwise convergence of probability densities. Indeed, the latter implies the former by Scheffé's theorem (Scheffé, 1947). The former implies the latter if the $p^N$ are equicontinuous and uniformly bounded as $N \to \infty$ (Boos, 1985).

We finally discuss *convergence in probability*. This is the statement that, for all $\epsilon > 0$,

$$\lim_{N \to \infty} \mathbb{P}\left(\left\|\boldsymbol{X}^N - \boldsymbol{X}^\infty\right\|_\infty > \epsilon\right) = 0. \tag{58}$$

We denote this convergence using the notation:

$$\left(X_i^N\right)_i \xrightarrow{p} \left(X_i^\infty\right)_i. \tag{59}$$

One way to quantify convergence in probability is through the *Ky Fan metric*, which is given by:

$$\alpha\left(p^N, p^\infty\right) = \inf\left\{\epsilon > 0 \mid \mathbb{P}\left(\left\|\boldsymbol{X}^N - \boldsymbol{X}^\infty\right\|_\infty > \epsilon\right) \leq \epsilon\right\}. \tag{60}$$

$\alpha\left(p^N, p^\infty\right) \to 0$ if and only if $\left(X_i^N\right)_i \xrightarrow{p} \left(X_i^\infty\right)_i$. The Ky Fan metric upper bounds the Lévy–Prokhorov metric (Strassen, 1965), consistent with the notion of convergence in probability implying convergence in distribution.

We will in what follows often be loose with language and say two sequences of distributions converge to one another at a rate $K$, either weakly or in probability, rather than state convergence to a fixed distribution. This should be understood to mean that their distance in associated metric asymptotically vanishes as $K^{-1}$.

# B  FORMAL DISCUSSION OF THE MAIN RESULTS

## B.1  THE JORDAN ALGEBRAIC STRUCTURE OF VARIATIONAL LOSS LANDSCAPES

With these preliminaries in place we can write the variational loss function given in Eq. (36) as the canonical trace form on $\mathcal{H}^N(\mathbb{C})$:

$$\ell(\boldsymbol{\theta}; \boldsymbol{\rho}) = \tau(\boldsymbol{\rho}, T(\boldsymbol{\theta})\boldsymbol{O}), \tag{61}$$

where here $\boldsymbol{\rho}, \boldsymbol{O} \in \mathcal{H}^N(\mathbb{C})$, and $T(\boldsymbol{\theta})$ parameterizes some path-connected compact Lie group $\mathcal{T}$. Note that $\{T(\boldsymbol{\theta})\boldsymbol{O}\}_{\boldsymbol{\theta}}$ generates a Jordan subalgebra $\mathcal{A} \subseteq \mathcal{H}^N(\mathbb{C})$ such that $\boldsymbol{O} \in \mathcal{A}$ and $\mathcal{T} \subseteq \mathrm{Aut}_1(\mathcal{A})$. As $T(\boldsymbol{\theta})\boldsymbol{O} \in \mathcal{A}$ for all $\boldsymbol{\theta}$, the trace form is zero for components of $\boldsymbol{\rho}$ orthogonal to $\mathcal{A}$. Because of this we will often also consider $\boldsymbol{\rho}$ an element of $\mathcal{A}$ in a slight abuse of notation.

We call a variational loss function of this form a *Jordan algebra-supported ansatz* (JASA), in analogy with the term *Lie-algebra supported ansatz* (LASA) introduced in Fontana et al. (2024). However, whereas variational loss functions are LASAs only if $i\boldsymbol{O}$ belongs to the dynamical Lie algebra generating the ansatz, *all* variational loss functions are JASAs (up to assuming the path-connectedness of $\mathcal{T}$). Indeed, in Appendix H we give a direct mapping from both LASAs and the variational matchgate formalism of Diaz et al. (2023) to JASAs and show that a mapping in the other direction is generally not possible.

We are often interested in the case where our variational loss landscape has some sort of structure, i.e., when $\mathcal{A} \cong \bigoplus_\alpha \mathcal{A}_\alpha \neq \mathcal{H}^N(\mathbb{C})$. In Appendix G we demonstrate that in the context of examining variational loss landscapes, the spin factor sectors ($\mathcal{A}_\alpha = \mathcal{L}_{N_\alpha}$) effectively reduce to real symmetric ($\mathcal{A}_\alpha = \mathcal{H}^{N_\alpha}(\mathbb{R})$) sectors. Because of this, we will from here on out focus on the case when $\mathcal{A}_\alpha$ is

the Jordan algebra of Hermitian matrices over a field, i.e., $\mathcal{A}_\alpha \cong \mathcal{H}^{N_\alpha}(\mathbb{F}_\alpha)$ for $\mathbb{F}_\alpha = \mathbb{R}, \mathbb{C}, \mathbb{H}$. We will also throughout use $\beta_\alpha = 1, 2, 4$ to label these three cases, respectively, as is commonly done in the physics literature. We use $\mathrm{Tr}_\alpha(\cdot)$ to denote the trace of $\cdot$ in the defining representation of $\mathcal{A}_\alpha$ and $\mathrm{Tr}(\cdot)$ to denote the trace of $\cdot$ according to its representation in $\mathcal{H}^N(\mathbb{C}) \supseteq \mathcal{A}$.

Recalling from Appendix A.2 the universality of the trace form and the direct product decomposition of $\mathrm{Aut}_1(\mathcal{A})$, we can rewrite the loss function of Eq. (61) as:

$$\ell(\boldsymbol{\theta}; \boldsymbol{\rho}) = \sum_\alpha \ell^\alpha(\boldsymbol{\theta}; \boldsymbol{\rho}) \equiv \sum_\alpha I_\alpha \tau^\alpha\left(\boldsymbol{\rho}^\alpha, T^\alpha(\boldsymbol{\theta})\, \boldsymbol{O}^\alpha\right), \tag{62}$$

where here $\cdot^\alpha$ is used to denote the component of $\cdot$ in $\mathcal{A}_\alpha$, $\tau^\alpha$ is the canonical trace form on $\mathcal{A}_\alpha$, and $I_\alpha > 0$ is the constant due to Eq. (42).

We now give an explicit form for $T^\alpha(\boldsymbol{\theta})$. Recall that elements of $\mathrm{Aut}_1(\mathcal{A}_\alpha)$ are, in the defining representation of $\mathcal{A}_\alpha$, given by the conjugation action by elements of $\mathrm{SO}(N_\alpha)$, $\mathrm{SU}(N_\alpha)$, or $\mathrm{Sp}(N_\alpha)$ when $\mathbb{F}_\alpha = \mathbb{R}, \mathbb{C}, \mathbb{H}$, respectively. We let $G_\alpha$ denote the corresponding Lie group and $\mathfrak{g}_\alpha$ the associated Lie algebra. We also denote $G$ as the direct product over $G_\alpha$, and $\mathfrak{g}$ the corresponding Lie algebra. Following the typical structure of variational quantum algorithms (Peruzzo et al., 2014) we will assume $T^\alpha(\boldsymbol{\theta})$ in the defining representation corresponds to conjugation by:

$$U_{\boldsymbol{g},h}^\alpha(\boldsymbol{\theta}) = g_0^{\alpha\dagger}\left(\prod_{i=1}^p g_i^\alpha \exp\left(\theta_i A_i^\alpha\right) g_i^{\alpha\dagger}\right) h^\alpha \in G_\alpha \tag{63}$$

for some $g_i, h \in G$ and $A_i \in \mathfrak{g}$, where $\cdot^\alpha$ denotes the projection onto $G_\alpha$ or $\mathfrak{g}_\alpha$ appropriately. With this choice of ansatz it is also easy to see the derivatives of $\ell$ have algebraic interpretations. For instance, at $\boldsymbol{\theta} = \boldsymbol{0}$, $i \geq j$,

$$\partial_i \ell(\boldsymbol{\theta}; \boldsymbol{\rho}) = \sum_\alpha I_\alpha \tau^\alpha\left(g_0^\alpha \boldsymbol{\rho}^\alpha g_0^{\alpha\dagger}, \left[g_i^\alpha A_i^\alpha g_i^{\alpha\dagger}, h^\alpha \boldsymbol{O}^\alpha h^{\alpha\dagger}\right]\right), \tag{64}$$

$$\partial_i \partial_j \ell(\boldsymbol{\theta}; \boldsymbol{\rho}) = \sum_\alpha I_\alpha \tau^\alpha\left(g_0^\alpha \boldsymbol{\rho}^\alpha g_0^{\alpha\dagger}, \left[g_j^\alpha A_j^\alpha g_j^{\alpha\dagger}, \left[g_i^\alpha A_i^\alpha g_i^{\alpha\dagger}, h^\alpha \boldsymbol{O}^\alpha h^{\alpha\dagger}\right]\right]\right), \tag{65}$$

where the commutator action of the automorphism group of a Jordan algebra is canonically defined through its representation $L$ defined in Eq. (39) (see Lemma 7 of Koecher (1999b)).

## B.2 Jordan Algebraic Wishart Systems

We have demonstrated that all variational loss landscapes are Jordan algebra-supported ansatzes (JASAs). We are finally ready to discuss our main results, which give an explicit expression for the loss landscape of a JASA when its ansatz takes the form of Eq. (63). To do this we first define a *Jordan algebraic Wishart system* (JAWS), leaving for now ambiguous the connection to Wishart matrices.

**Definition 11** (Jordan algebraic Wishart system). Let $O$ belong to a Jordan algebra $\mathcal{A} \subseteq \mathcal{H}^N(\mathbb{C})$ and let $\mathcal{T}$ be a path-connected subspace $\mathcal{T} \subseteq \mathrm{Aut}_1(\mathcal{A})$. Let

$$\mathcal{A} \cong \bigoplus_\alpha \mathcal{A}_\alpha \tag{66}$$

be the decomposition of $\mathcal{A}$ into simple Euclidean components as in Theorem 8 with associated decomposition $\mathcal{T} \cong \bigtimes_\alpha \mathcal{T}_\alpha$ as in Eq. (45). Let $\mathcal{G}_i^\alpha, \mathcal{H}^\alpha$ be independent distributions over $\mathcal{T}_\alpha$, and $\boldsymbol{A} = \{A_i\}_{i=1}^p$ elements of the Lie algebra of $\mathrm{Aut}_1(\mathcal{A})$. We call

$$\mathcal{J} = (\mathcal{A}, \mathcal{T}, \boldsymbol{\mathcal{G}}, \boldsymbol{\mathcal{H}}, \boldsymbol{A}, O) \tag{67}$$

a *Jordan algebraic Wishart system* (JAWS).

Every JAWS has associated with it a loss function as discussed in Appendix B.1. Using $\cdot^\alpha$ to label the projection of $\cdot$ into $\mathcal{A}_\alpha$ in its defining representation, the associated loss function takes the form:

$$\ell(\boldsymbol{\theta}; \boldsymbol{\rho}) = \sum_\alpha \ell^\alpha(\boldsymbol{\theta}; \boldsymbol{\rho}) = \sum_\alpha I_\alpha \mathrm{Tr}_\alpha\left(\boldsymbol{\rho}^\alpha \boldsymbol{U}^\alpha(\boldsymbol{\theta})\, \boldsymbol{O}^\alpha \boldsymbol{U}^\alpha(\boldsymbol{\theta})^\dagger\right), \tag{68}$$

where $\boldsymbol{U}^\alpha(\boldsymbol{\theta}) \in G_\alpha$ for all $\boldsymbol{\theta}$ and is of the form:

$$\boldsymbol{U}^\alpha(\boldsymbol{\theta}) \equiv g_0^{\alpha\dagger}\left(\prod_{i=1}^p g_i^\alpha \exp\left(\theta_i A_i^\alpha\right) g_i^{\alpha\dagger}\right) h^\alpha. \tag{69}$$

Here, $g_i^\alpha$ is the defining representation of an element drawn from from the distribution $\mathcal{G}_i^\alpha$, and $h^\alpha$ is similar for $\mathcal{H}^\alpha$.

Our main result is a concise, asymptotic description of $\ell(\boldsymbol{\theta};\boldsymbol{\rho})$ when the ansatz is chosen "sufficiently independently" from $\boldsymbol{O}$ (up to respecting the algebra $\mathcal{A}$). More formally, the rate at which $\ell(\boldsymbol{\theta};\boldsymbol{\rho})$ converges to its asymptotic limit depends on the parameters defined by the following assumption.

**Assumption 12** ($\epsilon$-approximate $t$-design of observable basis)**.** Each $\mathcal{H}^\alpha$ is an $\epsilon$-approximate $t$-design over $\mathcal{T}_\alpha = \mathrm{Aut}_1(\mathcal{A}_\alpha)$.

If there is a sense of geometric locality in the system, i.e., if the reverse lightcone of $\boldsymbol{O}^\alpha$ is of the form:

$$\boldsymbol{U}^\alpha(\boldsymbol{\theta})\,\boldsymbol{O}^\alpha\boldsymbol{U}^\alpha(\boldsymbol{\theta})^\dagger = \tilde{\boldsymbol{O}}^\alpha \otimes \boldsymbol{I} \tag{70}$$

for some $\tilde{\boldsymbol{O}}^\alpha$ acting on a space of dimension at most $D \equiv \exp\left(\mathrm{O}\left(p^d\right)\right)$ for constant geometric dimension $d$, one can explicitly enforce this with only a modest overhead in circuit depth. Indeed, one may embed $\mathcal{A}_\alpha$ into qubits and use the construction of Harrow & Mehraban (2023) to give a $d$-dimensional random circuit of depth $\mathrm{O}\left(\mathrm{poly}\left(t\right)p\right)$ that achieves this over $\mathrm{Aut}_1\left(\mathcal{H}^{2^n}(\mathbb{C})\right) \supset \mathrm{Aut}_1(\mathcal{A}_\alpha)$ for some $n = \mathrm{O}\left(p^d\right)$ and $\epsilon = \exp\left(-\Omega\left(p\right)\right)$.

Given Assumption 12 we are able to prove our main result on the convergence of loss functions. We are able to prove our convergence for any $N$-dependent overall normalization of the loss function $\mathcal{N}$, the only requirement being that central moments of sufficiently large (constant) order have a finite limit as $N \to \infty$. This is always true for $\mathcal{N} = 1$ and, depending on the specific choice of $t$-design, potentially holds for any $\mathcal{N} \le \mathrm{O}\left(N^{1-\delta}\right)$ where $\delta > 0$ is constant.

**Theorem 13** (Loss function distribution)**.** *Let $\mathcal{J}$ be a JAWS satisfying Assumption 12 with loss function $\ell(\boldsymbol{\theta};\boldsymbol{\rho})$ for $\boldsymbol{\rho} \in \mathcal{R}$. Further assume that*

$$\mathrm{Tr}\left((\boldsymbol{\rho}^\alpha)^2\right) = \Omega\left(\frac{1}{N_\alpha^{0.999}}\right). \tag{71}$$

*Fix $\boldsymbol{\theta}$. Assume $\mathcal{N} \le \mathrm{O}\left(\min_\alpha N_\alpha^{0.99}\right)$ is such that there exists a constant $k^*$ such that the central moments of order $k > k^*$ of $(\mathcal{N}\ell(\boldsymbol{\theta};\boldsymbol{\rho}))_{\boldsymbol{\rho}\in\mathcal{R}}$ are finite as all $\epsilon^{-1},t,N_\alpha \to \infty$. We have the convergence in joint distributions:*

$$(\mathcal{N}\ell(\boldsymbol{\theta};\boldsymbol{\rho}))_{\boldsymbol{\rho}\in\mathcal{R}} \rightsquigarrow \left(\mathcal{N}\hat{\ell}(\boldsymbol{\theta};\boldsymbol{\rho})\right)_{\boldsymbol{\rho}\in\mathcal{R}} \tag{72}$$

*as all $\epsilon^{-1},t,N_\alpha \to \infty$, where*

$$\hat{\ell}(\boldsymbol{\theta};\boldsymbol{\rho}) \equiv \sum_\alpha \hat{\ell}^\alpha(\boldsymbol{\theta};\boldsymbol{\rho}) \equiv \sum_\alpha \frac{I_\alpha \overline{o}^\alpha}{r_\alpha}\,\mathrm{Tr}_\alpha\left(\boldsymbol{\rho}^\alpha \boldsymbol{W}^\alpha\right), \tag{73}$$

*$\overline{o}^\alpha$ is the arithmetic mean eigenvalue of $\boldsymbol{O}^\alpha$ in the defining representation:*

$$\overline{o}^\alpha = N_\alpha^{-1}\,\mathrm{Tr}_\alpha\left(\boldsymbol{O}^\alpha\right), \tag{74}$$

*and the $\boldsymbol{W}^\alpha$ are independent Wishart-distributed random matrices with $\beta_\alpha r_\alpha$ degrees of freedom, where*

$$r_\alpha \equiv \frac{\|\boldsymbol{O}^\alpha\|_*^2}{\|\boldsymbol{O}^\alpha\|_F^2}. \tag{75}$$

*In particular, the distributions differ in Lévy–Prokhorov metric by*

$$\pi = \mathrm{O}\left(\frac{\log\log\left(\mathcal{N}^{-t}\epsilon^{-1}\right)}{\log\left(\mathcal{N}^{-t}\epsilon^{-1}\right)} + \frac{\log(t)}{\sqrt{t}} + \sum_\alpha \sqrt{\frac{\mathcal{N}\log\left(N_\alpha\right)}{N_\alpha^{1.001}}}\right) = \mathrm{o}\left(1\right). \tag{76}$$

In Appendix I we show that Eq. (71) can effectively be taken WLOG. This is because all $\boldsymbol{\rho}^\alpha$ for which this is not true are such that $\ell^\alpha(\boldsymbol{\theta}; \boldsymbol{\rho})$ have asymptotically vanishing variance even when rescaled by the potentially exponentially large $\mathcal{N} = \mathrm{O}(N)$. This is why we do not take Eq. (71) as a numbered Assumption. Examples of the loss density are illustrated in Figure 1(a).

We can strengthen the statement of weak convergence in Theorem 13 to one of pointwise convergence of densities given equicontinuity and boundedness of the appropriately normalized loss function density. Whether this is true depends on the specifics of the distributions $\mathcal{G}, \mathcal{H}$, particularly whether the loss is equicontinuous and bounded at the same $\mathcal{N} = \sqrt{r_\alpha}$ scale as $\hat{\ell}^\alpha$. We give some standard examples where this is true in Appendix K.

**Corollary 14** (Convergence of loss function densities). *Let $\mathcal{J}, \ell, \hat{\ell}$ be as in Theorem 13. Assume the density of $\left(\sqrt{r_\alpha}\ell^\alpha(\boldsymbol{\theta}; \boldsymbol{\rho})\right)_{\boldsymbol{\rho} \in \mathcal{R}, \alpha}$ is equicontinuous and bounded as all $\epsilon^{-1}, t, N_\alpha \to \infty$. Then the joint density of $\left(\sqrt{r_\alpha}\ell^\alpha(\boldsymbol{\theta}; \boldsymbol{\rho})\right)_{\boldsymbol{\rho} \in \mathcal{R}, \alpha}$ is pointwise equal to that of $\left(\sqrt{r_\alpha}\hat{\ell}^\alpha(\boldsymbol{\theta}; \boldsymbol{\rho})\right)_{\boldsymbol{\rho} \in \mathcal{R}, \alpha}$ up to an additive error $\mathrm{O}(\pi)$.*

As-written our result only gives the loss function distribution at a single point $\boldsymbol{\theta}$ in parameter space. However, any finite set $\Theta$ of points in parameter space can be considered by taking

$$\boldsymbol{\rho}(\boldsymbol{\theta}) = \boldsymbol{U}^\alpha(\boldsymbol{\theta}) \boldsymbol{\rho} \boldsymbol{U}^{\alpha\dagger}(\boldsymbol{\theta}) \tag{77}$$

to be elements in an augmented set of input states:

$$\mathcal{R}' = \mathcal{R} \times \Theta. \tag{78}$$

We can give a more concrete form for the parameter-dependence of the loss landscape by considering the joint distribution of $\ell$ with its derivatives. We begin with the gradient. In order to consider the joint distribution over what can potentially be many gradient components—a number growing with the number of parameters $p$—we take the following additional assumption on the growth of $t$ with $p$ so we are able to fully capture correlations between the derivatives.

**Assumption 15** (Scaling of parameter space dimension with $t$-design). The number of trained parameters $p$ satisfies:[5]

$$p^2 \leq \mathrm{o}\left(\min\left(\frac{\log(\epsilon^{-1})}{\log\log(\epsilon^{-1})}, \frac{\sqrt{t}}{\log(t)}\right)\right), \tag{79}$$

where the $\mathcal{G}_i^\alpha$ are i.i.d. $\epsilon$-approximate $t$-designs over $\mathcal{T}_\alpha = \mathrm{Aut}_1(\mathcal{A}_\alpha)$, and

$$p \leq \mathrm{O}\left(\mathrm{poly}\left(\min_\alpha N_\alpha\right)\right). \tag{80}$$

It is possible to weaken this assumption and instead assume that the $\mathcal{G}_i^\alpha$ are $\epsilon$-approximate 2-designs rather than $t$-designs, but this is at the expense of requiring $\mathcal{N} \leq \mathrm{o}\left(N^{\frac{2}{3}}\right)$ and only considering jointly at most $\sim \log(N)$ components of the gradient. We leave further details of this alternative setting to Appendix J.

It will also be convenient for the rest of our discussion to consider a concrete set of $\boldsymbol{A}_i$.

**Assumption 16** (Concrete choice of $\boldsymbol{A}_i$). For $\mathbb{F}_\alpha = \mathbb{C}, \mathbb{H}$, the $\boldsymbol{A}_i^\alpha$ are rank-1. For $\mathbb{F}_\alpha = \mathbb{R}$, the $\boldsymbol{A}_i^\alpha$ are rank-2.

The distinction between the cases $\mathbb{F}_\alpha = \mathbb{C}, \mathbb{H}$ and $\mathbb{F}_\alpha = \mathbb{R}$ is due to there being no rank 1 operators in the defining representation of $\mathfrak{so}(N_\alpha)$. This choice of $\boldsymbol{A}_i^\alpha$ may seem unphysical due to their nonlocality, but one can emulate the behavior of high-rank (e.g., Pauli) rotations by considering a factor of $N_\alpha$ more layers $p_\alpha$ in a given simple sector—each, for instance, performing a rotation under each eigenvector of $\boldsymbol{A}_i^\alpha$—and then tying the associated parameters together. This breaks no other assumptions as taking $p_\alpha \to N_\alpha p_\alpha$ maintains Eq. (80).

We then have the following theorem. It is stated assuming only a single input $\boldsymbol{\rho}$ which is rank-1 when projected to the defining representation of each simple sector to simplify the final result. However, in Appendix D.3 we give a full, exact expression of the joint distribution in terms of Wishart matrix elements for any choice of $\mathcal{R} \ni \boldsymbol{\rho}$ and independent of the scaling of $N_\alpha - r_\alpha$.

---

[5]We here use a bound on $p^2$ so that later we can reuse this assumption for the Hessian; at this stage we really only need this bound for $p$.

**Theorem 17** (Gradient distribution). *Consider the setting of Theorem 13, with the additional Assumptions 15 and 16. Assume $\frac{\sigma_o^\alpha}{\overline{o}}$ is bounded—where $\sigma_o^\alpha$ is the standard deviation of the eigenvalues of $\boldsymbol{O}^\alpha$—and assume $|\mathcal{R}| = 1$ with element $\boldsymbol{\rho}$ such that each $\boldsymbol{\rho}^\alpha$ is rank-1 in its defining representation. We have the convergence in joint distributions:*

$$\left(\mathcal{N}\ell\left(\boldsymbol{\theta};\boldsymbol{\rho}\right), \mathcal{N}\partial_i\ell\left(\boldsymbol{\theta};\boldsymbol{\rho}\right)\right)_{i\in[p]} \rightsquigarrow \left(\mathcal{N}\hat{\ell}\left(\boldsymbol{\theta};\boldsymbol{\rho}\right), \mathcal{N}\hat{\ell}_{;i}\left(\boldsymbol{\theta};\boldsymbol{\rho}\right)\right)_{i\in[p]} \tag{81}$$

*as all $\epsilon^{-1}, t, N_\alpha \to \infty$, where conditioned on*

$$\hat{\ell}^\alpha\left(\boldsymbol{\theta};\boldsymbol{\rho}\right) = z^\alpha \tag{82}$$

*we have that*

$$\hat{\ell}_{;i|z^\alpha}^\alpha\left(\boldsymbol{\theta};\boldsymbol{\rho}\right) = \frac{2I_\alpha\sigma_o^\alpha\operatorname{Tr}_\alpha\left(\boldsymbol{\rho}^\alpha\right)}{N_\alpha}\sqrt{\frac{\beta_\alpha z^\alpha}{I_\alpha\overline{o}^\alpha}}\chi_i^\alpha G_i^\alpha, \tag{83}$$

*where*

$$\hat{\ell}_{;i}\left(\boldsymbol{\theta};\boldsymbol{\rho}\right) \mid \left\{\hat{\ell}^\alpha\left(\boldsymbol{\theta},\boldsymbol{\rho}\right) = z^\alpha\right\}_\alpha \equiv \hat{\ell}_{;i|\boldsymbol{z}}\left(\boldsymbol{\theta};\boldsymbol{\rho}\right) \equiv \sum_\alpha \hat{\ell}_{;i|z^\alpha}^\alpha\left(\boldsymbol{\theta};\boldsymbol{\rho}\right). \tag{84}$$

*Here, the $\chi_i^\alpha$ are independent $\chi$-distributed random variables with $\max\left(2, \beta_\alpha\right)$ degrees of freedom and the $G_i^\alpha$ are i.i.d. standard normal random variables. In particular, the distributions differ in Lévy–Prokhorov metric by*

$$\pi' = \operatorname{O}\left(\frac{p\log\log\left(\mathcal{N}^{-t}\epsilon^{-1}\right)}{\log\left(\mathcal{N}^{-t}\epsilon^{-1}\right)} + \frac{p\log\left(t\right)}{\sqrt{t}} + \sum_\alpha\sqrt{\frac{\mathcal{N}\log\left(N_\alpha\right)}{N_\alpha^{1.001}}} + \frac{\mathcal{N}^{\frac{1}{3}}\log\left(N\right)}{N^{\frac{5}{12}}}\right) = \operatorname{o}\left(1\right). \tag{85}$$

This result is stronger than typical barren plateau results as it gives the full distributional form of the gradient—even when conditioned on the loss contribution $z^\alpha$—rather than just its variance (McClean et al., 2018; Larocca et al., 2022; Fontana et al., 2024; Ragone et al., 2024). The conditional distribution as in Eq. (83) is illustrated in Figure 1(b).

Just as with the loss function, we can strengthen Theorem 17 to show pointwise convergence in probability densities assuming equicontinuity of the original distribution.

**Corollary 18** (Convergence of gradient densities). *Let $\mathcal{J}, \ell, \hat{\ell}$ be as in Corollary 14. Assume the density of $\left(\sqrt{r_\alpha}\ell_{;i}^\alpha\left(\boldsymbol{\theta};\boldsymbol{\rho}\right)\right)_\alpha$ is equicontinuous and bounded for all $z^\alpha \in \mathbb{R}_{\geq 0}$ as all $\epsilon^{-1}, t, N_\alpha \to \infty$. Then the joint density of $\left(\sqrt{r_\alpha}\ell_{;i}^\alpha\left(\boldsymbol{\theta};\boldsymbol{\rho}\right)\right)_\alpha$ is pointwise equal to that of $\left(\sqrt{r_\alpha}\hat{\ell}_{;i}^\alpha\left(\boldsymbol{\theta};\boldsymbol{\rho}\right)\right)_\alpha$ up to an additive error $\operatorname{O}\left(\pi'\right)$.*

We now give our final result, which specifies the joint distribution of not only the loss and gradient but also the Hessian at critical points. This is required to reason about the critical point distribution of the loss landscape using the so-called Kac–Rice formula (Adler & Taylor, 2007) that we will discuss in detail in Appendix C.3. We state our result assuming $\sigma_o \ll \overline{o}$ for simplicity, giving as we did for the gradient the full expression in terms of Wishart matrix elements in Appendix D.3.

**Theorem 19** (Hessian distribution). *Let $\mathcal{J}, \ell, \hat{\ell}$ be as in Theorem 17. Assume all $\frac{\sigma_o^\alpha}{\overline{o}^\alpha} = \operatorname{o}\left(1\right)$ as $N_\alpha \to \infty$. We have the convergence in joint distributions:*

$$\left(\ell\left(\boldsymbol{\theta};\boldsymbol{\rho}\right), \partial_i\ell\left(\boldsymbol{\theta};\boldsymbol{\rho}\right), \partial_i\partial_j\ell\left(\boldsymbol{\theta};\boldsymbol{\rho}\right)\right)_{\boldsymbol{\rho}\in\mathcal{R}, i\geq j\in[p]} \rightsquigarrow \left(\hat{\ell}\left(\boldsymbol{\theta};\boldsymbol{\rho}\right), \hat{\ell}_{;i}\left(\boldsymbol{\theta};\boldsymbol{\rho}\right), \hat{\ell}_{;i,j}\left(\boldsymbol{\theta};\boldsymbol{\rho}\right)\right)_{\boldsymbol{\rho}\in\mathcal{R}, i\geq j\in[p]} \tag{86}$$

*as all $\epsilon^{-1}, t, N_\alpha \to \infty$, where conditioned on*

$$\hat{\ell}^\alpha\left(\boldsymbol{\theta};\boldsymbol{\rho}\right) = z^\alpha \tag{87}$$

*and*

$$\hat{\ell}_{;i|z^\alpha}^\alpha\left(\boldsymbol{\theta};\boldsymbol{\rho}\right) = 0 \tag{88}$$

*we have that*

$$\hat{\ell}_{;i,j|z^\alpha,0}^\alpha\left(\boldsymbol{\theta};\boldsymbol{\rho}\right) = \frac{2I_\alpha\sigma_o^\alpha\operatorname{Tr}_\alpha\left(\boldsymbol{\rho}^\alpha\right)}{N_\alpha^2}\sqrt{\frac{z^\alpha}{I_\alpha\overline{o}_\alpha}}G_i^\alpha\chi_j^\alpha\left\langle i\right|\boldsymbol{W}^\alpha\left|j\right\rangle, \tag{89}$$

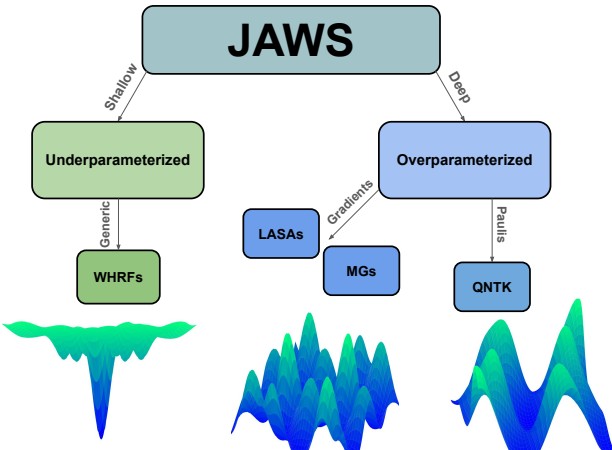

Figure 3: **Relation between models for quantum variational loss landscapes.** Our introduced theory for the loss landscapes of quantum neural networks (QNNs) are *Jordan algebraic Wishart systems* (JAWS), which relate the algebraic structure of a given QNN architecture to an asymptotic random process description of the loss landscape. This JAWS description reduces to the previously-studied Wishart hypertoroidal random fields (WHRFs) and quantum neural tangent kernel (QNTK) in different settings, and also reproduces the loss function variance known for Lie algebra-supported ansatzes (LASAs) and matchgate (MG) networks. Cartoons of the loss landscapes associated with previously studied models are shown.

*where*

$$\hat{\ell}_{;i,j}\left(\boldsymbol{\theta};\boldsymbol{\rho}\right) \mid \left\{\hat{\ell}^{\alpha}\left(\boldsymbol{\theta},\boldsymbol{\rho}\right) = z^{\alpha}, \left(\hat{\ell}^{\alpha}_{;i}\left(\boldsymbol{\theta};\boldsymbol{\rho}\right)\right)_{i\in[p]} = \boldsymbol{0}\right\}_{\alpha} \equiv \hat{\ell}_{;i,j|\boldsymbol{z},\boldsymbol{0}}\left(\boldsymbol{\theta};\boldsymbol{\rho}\right) \equiv \sum_{\alpha}\hat{\ell}^{\alpha}_{;i,j|z^{\alpha},0}\left(\boldsymbol{\theta};\boldsymbol{\rho}\right).$$

(90)

*Here, the $G_i^{\alpha}$ are i.i.d. standard normal random variables, the $\chi_i^{\alpha}$ are independent $\chi$-distributed random variables with $\max\left(2,\beta_{\alpha}\right)$ degrees of freedom, and the $\boldsymbol{W}^{\alpha}$ are independent Wishart-distributed random matrices with $\beta_{\alpha}r_{\alpha}$ degrees of freedom. In particular, the distributions differ in Lévy–Prokhorov metric by*

$$\pi'' = \mathrm{O}\left(\frac{p^2\log\log\left(\mathcal{N}^{-t}\epsilon^{-1}\right)}{\log\left(\mathcal{N}^{-t}\epsilon^{-1}\right)} + \frac{p^2\log\left(t\right)}{\sqrt{t}} + \sum_{\alpha}\frac{\mathcal{N}^{\frac{1}{3}}\log\left(N_{\alpha}\right)}{N_{\alpha}^{\frac{1}{3}}}\right) = \mathrm{o}\left(1\right).$$

(91)

## C  FORMAL DISCUSSION OF THE CONSEQUENCES OF OUR RESULTS

Before proceeding with proofs of our results we examine their implications. These are summarized in Figure 3.

### C.1  BARREN PLATEAUS

The first implication of our results that we will discuss is the unification of barren plateau results (Fontana et al., 2024; Ragone et al., 2024) in the large $N_{\alpha}$ limit.

**Corollary 20** (General expression for the loss function variance). *Let $\ell$ be as in Theorem 13. The variance is:*

$$\mathrm{Var}\left[\ell\left(\boldsymbol{\theta};\boldsymbol{\rho}\right)\right] = \sum_{\alpha}\frac{\mathrm{Tr}\left(\left(\boldsymbol{O}^{\alpha}\right)^2\right)\mathrm{Tr}\left(\left(\boldsymbol{\rho}^{\alpha}\right)^2\right)}{\dim_{\mathbb{R}}\left(\mathfrak{g}_{\alpha}\right)} + \mathrm{O}\left(\pi\right).$$

(92)

*Here,* $\dim_{\mathbb{R}}(\mathfrak{g}_\alpha)$ *is the dimension of the automorphism group of* $\mathcal{A}_\alpha$ *and* $\pi$ *is as in Eq. (76).*[6]

*Proof.* Note that

$$\dim_{\mathbb{R}}(\mathfrak{g}_\alpha) = (\beta_\alpha - 1) N_\alpha + \frac{\beta_\alpha N_\alpha (N_\alpha - 1)}{2} = \frac{\beta_\alpha N_\alpha^2}{2} + \mathrm{O}(N_\alpha). \tag{93}$$

Similarly,

$$\frac{(\overline{o}^\alpha)^2}{r_\alpha} = \frac{\|\boldsymbol{O}^\alpha\|_{\mathrm{F}}^2}{N_\alpha^2}. \tag{94}$$

By Eq. (42), we also have that

$$I_\alpha \|\boldsymbol{O}^\alpha\|_{\mathrm{F}}^2 = \mathrm{Tr}\left((\boldsymbol{O}^\alpha)^2\right), \tag{95}$$

with $\mathrm{Tr}(\cdot)$ the trace on the full $N$-dimensional Hilbert space. Using the fact that the diagonal entries of $\beta_\alpha \boldsymbol{W}^\alpha$ are i.i.d. $\chi^2$-distributed with $\beta_\alpha r_\alpha$ degrees of freedom (see Appendix A.4), we then have from Theorem 13 that as $N_\alpha \to \infty$:

$$
\begin{aligned}
\mathrm{Var}\left[\ell(\boldsymbol{\theta}; \boldsymbol{\rho})\right] &= \sum_\alpha \mathrm{Var}\left[\hat{\ell}^\alpha(\boldsymbol{\theta}; \boldsymbol{\rho})\right] + \mathrm{O}(\pi) \\
&= \sum_\alpha \frac{2 I_\alpha^2 (\overline{o}^\alpha)^2}{\beta_\alpha r_\alpha} \mathrm{Tr}_\alpha\left((\boldsymbol{\rho}^\alpha)^2\right) + \mathrm{O}(\pi) \\
&= \sum_\alpha \frac{2 I_\alpha (\overline{o}^\alpha)^2}{\beta_\alpha r_\alpha} \mathrm{Tr}\left((\boldsymbol{\rho}^\alpha)^2\right) + \mathrm{O}(\pi) \\
&= \sum_\alpha \frac{\mathrm{Tr}\left((\boldsymbol{O}^\alpha)^2\right) \mathrm{Tr}\left((\boldsymbol{\rho}^\alpha)^2\right)}{\dim_{\mathbb{R}}(\mathfrak{g}_\alpha)} + \mathrm{O}(\pi).
\end{aligned}
\tag{96}
$$

$\square$

This result immediately implies Theorem 1 of Ragone et al. (2024) in the case of Lie algebra-supported ansatzes. However, our result holds for *all* variational ansatzes, i.e., it does not rely on either i$\boldsymbol{O}$ or i$\boldsymbol{\rho}$ being a member of the dynamical Lie algebra generating the ansatz.

In the language of Ragone et al. (2024), $\mathrm{Tr}\left((\boldsymbol{\rho}^\alpha)^2\right)$ is the $\mathcal{A}_\alpha$-purity $\mathcal{P}_{\mathcal{A}_\alpha}$ of $\boldsymbol{O}$, and be thought of as measuring the *generalized entanglement* (Barnum et al., 2004) of $\boldsymbol{\rho}$ with respect to the Jordan algebraic structure of $\mathcal{A}$. In this sense, this barren plateau result can also be thought of as generalizing the entanglement-induceed barren plateaus previously studied by Ortiz Marrero et al. (2021).

## C.2 The Quantum Neural Tangent Kernel

We now connect our results to the quantum neural tangent kernel (QNTK) literature (Liu et al., 2022; 2023; You et al., 2022; García-Martín et al., 2023; Girardi & Palma, 2024; García-Martín et al., 2024). The landmark result in this field is that, under certain conditions on $\mathcal{R}$, $\boldsymbol{O}$, and $\mathcal{A}$, variational loss functions are asymptotically a Gaussian process when $\epsilon^{-1}, t, N \to \infty$. However, this same body of work has noted that such a Gaussian process description cannot generally be true: for instance, if $\boldsymbol{O}$ is rank-1 and $\mathcal{A}$ is the space of complex Hermitian matrices, the loss should be Porter–Thomas distributed as this reduces to a random circuit sampling setting (Boixo et al., 2018).

Our results can be seen as a unifying model of neural network loss landscapes, including both when convergence to a Gaussian process is achieved and when it is not. Recall that Theorem 13 demonstrated that the asymptotic expression for the variational loss with objective observable $\boldsymbol{O}$ is distributed as a Wishart process:

$$\hat{\ell}(\boldsymbol{\rho}) = \sum_\alpha \frac{I_\alpha \overline{o}^\alpha}{r_\alpha} \mathrm{Tr}_\alpha\left(\boldsymbol{\rho}^\alpha \boldsymbol{W}^\alpha\right), \tag{97}$$

---

[6]Through careful examination of our proof this error is actually identically zero when $t \geq 2$ and $\epsilon = 0$ in Assumption 12 as here we are only taking a second moment.

even when rescaled by a quantity $\mathcal{N}$ exponentially large in the problem size. This correctly captures the Porter–Thomas behavior when $\mathcal{A}$ is the space of complex Hermitian matrices and $\mathrm{rank}\,(\boldsymbol{O}) = r = 1$. This is because the diagonal entries of a complex Wishart matrix are $\chi^2$-distributed with two degrees of freedom each, which is identical to the exponential distribution.[7]

Indeed, our more general result can be used to *exactly* characterize when QNNs asymptotically form Gaussian processes. This occurs when the loss is scaled by some $\mathcal{N}$ such that Eq. (97) has nonvanishing variance yet the higher-order cumulants vanish. We compute the third-order cumulant:

$$\kappa_3\left(\mathcal{N}\hat{\ell}(\boldsymbol{\rho})\right) = \mathrm{O}\left(\mathcal{N}^3 \sum_\alpha \frac{I_\alpha^3\,(\overline{o}^\alpha)^3}{r_\alpha^2}\,\mathrm{Tr}_\alpha\left((\boldsymbol{\rho}^\alpha)^3\right)\right). \tag{98}$$

We thus can state this pair of conditions formally as follows.

**Corollary 21** (Exact conditions for convergence to a Gaussian process). *Let $\mathcal{J}, \ell$ be as in Theorem 13. $\mathcal{N}\ell\,(\boldsymbol{\theta};\boldsymbol{\rho})$ is asymptotically a Gaussian process as $\epsilon^{-1}, t, N \to \infty$ if and only if*

$$\mathcal{N} \le \mathrm{o}\left(\min_\alpha \frac{r_\alpha^{\frac{2}{3}}}{I_\alpha \overline{o}^\alpha\,\mathrm{Tr}_\alpha\left((\boldsymbol{\rho}^\alpha)^3\right)^{\frac{1}{3}}}\right) \tag{99}$$

*and $\mathcal{N}^2\,\mathrm{Var}\,[\ell\,(\boldsymbol{\theta};\boldsymbol{\rho})]$ (with the variance as given in Corollary 20) is nonvanishing for all $\boldsymbol{\rho} \in \mathcal{R}$.*

*When these conditions are met, the covariance is given by:*

$$\mathcal{K}\,(\boldsymbol{\rho},\boldsymbol{\rho}') = \lim_{N\to\infty} \mathcal{K}_N\,(\boldsymbol{\rho},\boldsymbol{\rho}')$$

$$= \lim_{N\to\infty} \mathcal{N}^2 \sum_\alpha \frac{\beta_\alpha\,\mathrm{Tr}\left((\boldsymbol{O}^\alpha)^2\right)}{2\dim_{\mathbb{R}}\,(\mathfrak{g}_\alpha)\,r_\alpha} I_\alpha \mathbb{E}\left[\mathrm{Tr}_\alpha\left(\boldsymbol{\rho}^\alpha\,(\boldsymbol{W}^\alpha - r_\alpha)\right)\mathrm{Tr}_\alpha\left(\boldsymbol{\rho}'^\alpha\,(\boldsymbol{W}^\alpha - r_\alpha)\right)\right]. \tag{100}$$

It is now easy to see why (for simple $\mathcal{A}$) the $r = 1$ case does not converge to a Gaussian process: when $r = 1$ we may only choose a normalization $\mathcal{N} \le \mathrm{o}\,(N)$ such that higher-order cumulants vanish asymptotically, but then the variance vanishes asymptotically. In contrast, our results demonstrate convergence to a Wishart process through *any* finite number of cumulants, not just the first two.

We can also be more concrete on the form of $\mathcal{K}\,(\boldsymbol{\rho},\boldsymbol{\rho}')$. Indeed, we are able to show that each algebraic sector $\mathcal{A}_\alpha$ contributes to the covariance an explicit, quadratic expression in $\boldsymbol{\rho}^\alpha$ and $\boldsymbol{\rho}'^\alpha$, depending only on the field $\mathbb{F}_\alpha$ associated with its defining representation. In other words, whether or not a given quantum neural network is asymptotically a Gaussian process just depends on how SWAP tests of the inputs scale. As the expression is complicated we do not reproduce it here, instead giving references to where it may be found for each of $\mathbb{F}_\alpha = \mathbb{R}, \mathbb{C}, \mathbb{H}$.

**Theorem 22** ($\mathcal{K}$ is quadratic in $\boldsymbol{\rho},\boldsymbol{\rho}'$). *$\mathcal{K}\,(\boldsymbol{\rho},\boldsymbol{\rho}')$ can be written as a closed-form, explicit function of only the $\mathrm{Tr}\,(\boldsymbol{\rho}^\alpha)$, $\mathrm{Tr}\,(\boldsymbol{\rho}'^\alpha)$, $\mathrm{Tr}\left((\boldsymbol{\rho}^\alpha)^2\right)$, $\mathrm{Tr}\left((\boldsymbol{\rho}'^\alpha)^2\right)$, and $\mathrm{Tr}\,(\boldsymbol{\rho}^\alpha\boldsymbol{\rho}'^\alpha)$.*

*Proof.* By the Haar invariance of $\boldsymbol{W}^\alpha$ there are explicit formulas for covariances of the form of $\mathcal{K}\,(\boldsymbol{\rho},\boldsymbol{\rho}')$ as degree-2 polynomials in $\boldsymbol{\rho}$ and $\boldsymbol{\rho}'$. These formulas are given in:

1. Theorem 1 of Redelmeier (2011) when $\mathbb{F} = \mathbb{R}$;

2. Theorem 3.5 of Mingo & Speicher (2006) when $\mathbb{F} = \mathbb{C}$;

3. Sec. 5.3 of Redelmeier (2021) when $\mathbb{F} = \mathbb{H}$.

The result then follows by noting that $I_\alpha\,\mathrm{Tr}_\alpha\,(\cdot^\alpha) = \mathrm{Tr}\,(\cdot^\alpha)$. $\qquad\square$

---

[7] It is also easy to check when $r = 1$ and the ansatz unitaries are Haar random that the assumptions we use in proving our results hold for any $\mathcal{N} \le \mathrm{O}\,(N)$.

We give a simple example of the calculation of $\mathcal{K}$ when all inputs $\boldsymbol{\rho_0}$ can be mutually diagonalized. By the unitary invariance of Wishart matrices we can assume WLOG that all inputs are diagonal. Inputs are then completely parameterized by the eigenvalues $\boldsymbol{x} \in \mathbb{R}^{N_\alpha}_{\geq 0}$ of the input states, where $\|\boldsymbol{x}\|_1 = 1$. By the independence of diagonal entries of a Wishart matrix we then have that (when $\mathcal{N} = 1$):

$$
\begin{aligned}
\mathcal{K}_N\left(\boldsymbol{x}, \boldsymbol{x'}\right) &= \mathcal{N}^2 \sum_\alpha \frac{\beta_\alpha \operatorname{Tr}\left(\left(\boldsymbol{O}^\alpha\right)^2\right)}{2 \dim_\mathbb{R}\left(\mathfrak{g}_\alpha\right) r_\alpha} \sum_{i=1}^N x_i^\alpha x_i'^\alpha \operatorname{Var}\left[\langle i|\, \boldsymbol{W}^\alpha\, |i\rangle\right] + \mathrm{o}\left(1\right) \\
&= \mathcal{N}^2 \sum_\alpha \frac{\operatorname{Tr}\left(\left(\boldsymbol{O}^\alpha\right)^2\right)}{\dim_\mathbb{R}\left(\mathfrak{g}_\alpha\right)} \boldsymbol{x}^\alpha \cdot \boldsymbol{x'}^\alpha + \mathrm{o}\left(1\right),
\end{aligned}
\tag{101}
$$

where in the final line we used similar simplifications as in Appendix C.1. Noting that $\boldsymbol{x}^\alpha \cdot \boldsymbol{x'}^\alpha$ is just the overlap of $\boldsymbol{\rho}^\alpha$ and $\boldsymbol{\rho'}^\alpha$ yields an equivalent formulation:

$$
\mathcal{K}_N\left(\boldsymbol{\rho}, \boldsymbol{\rho'}\right) = \mathcal{N}^2 \sum_\alpha \frac{\operatorname{Tr}\left(\left(\boldsymbol{O}^\alpha\right)^2\right)}{\dim_\mathbb{R}\left(\mathfrak{g}_\alpha\right)} \operatorname{Tr}\left(\boldsymbol{\rho}^\alpha \boldsymbol{\rho'}^\alpha\right) + \mathrm{o}\left(1\right) = \sum_\alpha \mathrm{O}\left(\frac{\mathcal{N}^2 r_\alpha}{N_\alpha^2}\right) \operatorname{Tr}\left(\boldsymbol{\rho}^\alpha \boldsymbol{\rho'}^\alpha\right) + \mathrm{o}\left(1\right).
\tag{102}
$$

In particular, when $\mathcal{A}$ is simple, a $\mathcal{N} = \Omega\left(\frac{N}{\sqrt{r}}\right)$ normalization is required for the network to asymptotically form a Gaussian process over pure inputs.

While previous results have shown that Gaussian processes efficiently train, we argue that this may paint an overly optimistic picture for generic QNNs. Focusing on the case where the ansatz is an $\Theta\left(\log\left(n\right)\right)$-depth, 2-dimensional circuit on $n$ qubits, we have that the QNN forms a Gaussian process when:

$$
\mathcal{N} = \exp\left(\Omega\left(\log\left(n\right)^2\right)\right),
\tag{103}
$$

i.e., at a normalization superpolynomial in $n$. Thus while it is known that such networks can achieve a constant improvement in $\mathcal{N}\hat{\ell}$ in time polynomial in $n$ via gradient descent (Girardi & Palma, 2024), this translates to a superpolynomially vanishing improvement in the loss function value when in the physical normalization of $\mathcal{N} = 1$. More generally, the covariance we here derive can be used with results in the classical neural tangent kernel literature to reason about the training behavior (Neal, 1996; Choromanska et al., 2015; Chaudhari, 2018; Lee et al., 2018) and generalization ability (Jacot et al., 2018; Wei et al., 2022) of this class of quantum neural networks, which we hope to analyze in more detail in the future.

### C.3  LOCAL MINIMA

We end by examining the distribution of local minima of the loss landscape. This has been done previously in the more restricted setting where no structure is imposed on the loss function (Anschuetz, 2022; Anschuetz & Kiani, 2022). In this section we assume the assumptions of Theorem 19 and that as all $\epsilon^{-1}, t, N_\alpha \to \infty$,

$$
\gamma_\alpha \equiv \frac{p_\alpha}{\beta_\alpha r_\alpha}
\tag{104}
$$

is held constant. Here, $p_\alpha$ is the number of $\boldsymbol{A}_i$ that are nontrivial on the simple component $\mathcal{A}_\alpha$ of $\mathcal{A}$. $\gamma_\alpha$ is the so-called *overparameterization ratio* discussed in Anschuetz (2022); Anschuetz & Kiani (2022). We also assume for the simplicity of our expressions that each simple sector is fully controllable, i.e., that each ansatz generator $\boldsymbol{A}_i$ is only nontrivial on a single simple component of $\mathcal{A}$. In principle a more general expression could be written as well, though the gradient density would be a convolution over complicated probability densities, and the Hessian distributed as a free convolution of multiple Wishart matrices.

We calculate the distribution of local minima via the *Kac–Rice formula* (Adler & Taylor, 2007), which gives the expected density of local minima of a function $\ell$ at a function value $z$. Anschuetz (2022) demonstrated that the assumptions for the Kac–Rice formula are satisfied for variational loss landscapes, and when rotationally invariant on the $p$-torus takes the form:

$$
\mathbb{E}\left[\operatorname{Crt}_0\left(z\right)\right] = \left(2\pi\right)^p \mathbb{E}\left[\det\left(\boldsymbol{H}_z\right)\boldsymbol{1}\left\{\boldsymbol{H}_z \succeq \boldsymbol{0}\right\}\right] \mathbb{P}\left[\boldsymbol{G}_z = \boldsymbol{0}\right] \mathbb{P}\left[\ell = z\right].
\tag{105}
$$

Here, $\mathbb{E}\left[\mathrm{Crt}_0\left(z\right)\right]$ is the expected density of local minima at a function value $z$, $\boldsymbol{G}_z$ is the gradient conditioned on $\ell = z$, and $\boldsymbol{H}_z$ is the Hessian conditioned on $\ell = z$ and $\boldsymbol{G} = \boldsymbol{0}$. In a slight abuse of notation, here $\mathbb{P}\left[\cdot\right]$ denotes the probability density associated with the event $\cdot$.

We can evaluate this expression using Theorems 13, 17, and 19.

**Corollary 23** (Density of local minima to multiplicative leading order)**.** *Let $\mathcal{J}, \hat{\ell}$ be as in Corollary 18 and Theorem 19. Assume each ansatz generator $\boldsymbol{A}_i$ is only nontrivial on a single simple component of $\mathcal{A}$, and further assume that $\boldsymbol{\rho}^\alpha$ is rank-1 in its defining representation. Assume as well that the overparameterization ratios:*

$$\gamma_\alpha \equiv \frac{p_\alpha}{\beta_\alpha r_\alpha} \tag{106}$$

*remain fixed as $N \to \infty$. Let $\mu_{\boldsymbol{H}_z^\alpha}$ be the empirical spectral measure of*

$$\tilde{\boldsymbol{H}}_z^\alpha \equiv N_\alpha^{-1} \boldsymbol{S}^\alpha \odot \boldsymbol{W}^\alpha, \tag{107}$$

*where (for $i \geq j$) $S_{i,j}^\alpha = |G_i^\alpha| \chi_j^\alpha$ with random variables defined as in Theorem 19 and $\odot$ denotes the Hadamard product. Let $\lambda_{z,\alpha}^*$ denote the infimum of the support of $\mu_{\boldsymbol{H}_z^\alpha}$. Then the expected density of local minima of $\hat{\ell}$ at a loss function value $z$ is:*

$$\mathbb{E}\left[\mathrm{Crt}\left(z\right)\right] = \bigstar_\alpha \mathbb{E}\left[\mathrm{Crt}_0^\alpha\right]\left(z\right), \tag{108}$$

*where*

$$\begin{aligned}
p^{-1}\ln\left(\mathbb{E}\left[\mathrm{Crt}_0^\alpha\left(z\right)\right]\right) = {} & \ln\left(\frac{\pi \max\left(2, \beta_\alpha\right)}{4\sqrt{\beta_\alpha}}\right) + \frac{1}{2\gamma_\alpha}\left(1 - \frac{z}{\mathrm{Tr}\left(\boldsymbol{\rho}^\alpha\right)\overline{o}^\alpha} + \ln\left(\frac{z}{\mathrm{Tr}\left(\boldsymbol{\rho}^\alpha\right)\overline{o}^\alpha}\right)\right) \\
& + p^{-1}\ln\mathbb{E}\left[\exp\left(p\int \mathrm{d}\mu_{\tilde{\boldsymbol{H}}_z^\alpha}\left(\lambda\right)\ln\left(\lambda\right)\right)\mathbf{1}\left\{\lambda_{z,\alpha}^* \geq 0\right\}\right] + \mathrm{o}\left(1\right).
\end{aligned} \tag{109}$$

*Proof.* We will first consider a single simple component $\mathcal{A}_\alpha$ with pure $\boldsymbol{\rho}^\alpha$ (and with $\alpha$ labels implicit for clarity of notation), and describe in the end how a full distribution of local minima can be determined from this via a convolution. From Corollary 14 it follows that the density of the loss is:[8]

$$\mathbb{P}\left[\hat{\ell} = z\right] = \frac{\left(\frac{\beta r z}{2 I \overline{o}}\right)^{\frac{\beta r}{2} - 1} \exp\left(-\frac{\beta r z}{2 I \overline{o}}\right)}{2\,\Gamma\left(\frac{\beta r}{2}\right)}. \tag{110}$$

Similarly, using Corollary 18 and recalling that $\beta$ is one of $1$, $2$, or $4$, we can evaluate the density of the gradient at zero:

$$\mathbb{P}\left[\boldsymbol{G}_z = \boldsymbol{0}\right] = \left(\frac{1}{\sqrt{2\pi}\sigma_z}\int_0^\infty \mathrm{d}x\,\frac{f_{\chi^{\max(2,\beta)}}\left(x\right)}{x}\right)^p = \left(\frac{\max\left(2, \beta\right)}{4\sigma_z}\right)^p, \tag{111}$$

where $f_{\chi^{\max(2,\beta)}}$ is the density of a $\chi$-distributed random variable with $\max\left(2, \beta\right)$ degrees of freedom and

$$\sigma_z = \frac{2 I \sigma_o}{N}\sqrt{\frac{\beta z}{I \overline{o}}}. \tag{112}$$

Finally, we consider the Hessian determinant. Recall Theorem 19 for the Hessian components, which gives the Hessian as:

$$\frac{N}{2 I \sigma_o}\sqrt{\frac{I \overline{o}}{z}}\boldsymbol{H}_z = N^{-1}\tilde{\boldsymbol{S}} \odot \boldsymbol{W}, \tag{113}$$

where $\tilde{S}_{i,j} = G_i \chi_j$. We now claim that $\boldsymbol{H}_z$ can only be positive semidefinite if all $G_i \geq 0$. To see this, note that $\chi_i W_{i,i}$ is always nonnegative. When $G_i < 0$, then, the $(i, i)$ entry of $\tilde{\boldsymbol{S}} \odot \boldsymbol{W}$ is negative and thus is not positive semidefinite. We therefore can consider $|G_i|$ rather than $G_i$ up

---

[8]Technically we need to consider the density of the loss rescaled by $\sqrt{N}$ to achieve pointwise convergence in densities, but the results are equivalent after rescaling $z$.

to pulling out a factor of $2^{-p}$ from the expectation. Putting everything together, to multiplicative leading order as $N \to \infty$,

$$
\begin{aligned}
p^{-1} \ln \left( \mathbb{E} \left[ \mathrm{Crt}_0 \left( z \right) \right] \right) = & \ln \left( \frac{\pi \max \left( 2, \beta \right)}{4 \sqrt{\beta}} \right) + \frac{1}{2\gamma} \left( 1 - \frac{z}{I \bar{o}} + \ln \left( \frac{z}{I \bar{o}} \right) \right) \\
& + p^{-1} \ln \mathbb{E} \left[ \exp \left( p \int \mathrm{d}\mu_{\tilde{\boldsymbol{H}}_z} \left( \lambda \right) \ln \left( \lambda \right) \right) \mathbf{1} \left\{ \lambda_z^* \geq 0 \right\} \right] + \mathrm{o} \left( 1 \right).
\end{aligned}
\tag{114}
$$

This completes our proof for a single simple component $\mathcal{A}_\alpha$ with pure $\boldsymbol{\rho}^\alpha$. To calculate the loss landscape of a JAWS associated with a nonsimple Jordan algebra, note that:

$$
\mathbb{P} \left[ \ell = z \right] = \underset{\alpha}{\text{\Large $\ast$}} \, \mathbb{P} \left[ \ell^\alpha = z^\alpha \right],
\tag{115}
$$

where $\ast$ denotes convolution. The relative weights of the various sectors introduced by $\mathrm{Tr}_\alpha \left( \boldsymbol{\rho}^\alpha \right)$ can be accounted for by taking:

$$
I_\alpha \to I_\alpha \, \mathrm{Tr}_\alpha \left( \boldsymbol{\rho}^\alpha \right) = \mathrm{Tr} \left( \boldsymbol{\rho}^\alpha \right).
\tag{116}
$$

From Eq. (105) and our simplifying assumptions, then,

$$
\mathbb{E} \left[ \mathrm{Crt}_0 \left( z \right) \right] = \underset{\alpha}{\text{\Large $\ast$}} \, \mathbb{E} \left[ \mathrm{Crt}_0^\alpha \right]_{I_\alpha \to \mathrm{Tr}(\boldsymbol{\rho}^\alpha)} \left( z \right),
\tag{117}
$$

where $\mathbb{E} \left[ \mathrm{Crt}_0^\alpha \right]$ denotes Eq. (114) associated with the algebraic sector $\mathcal{A}_\alpha$. $\qquad \square$

While Corollary 23 is exact, it is obtuse almost to the point of obscurity, particularly due to the expectation over the Hessian. The obstruction to further simplification is the presence of the Hadamard product between $\boldsymbol{S}^\alpha$ and $\boldsymbol{W}$ in $\tilde{\boldsymbol{H}}_z^\alpha$, which is difficult to handle analytically. To get around this, we consider a slightly modified quantity where we condition both sides of Eq. (105) on the events:

$$
|G_i| = \chi_i.
\tag{118}
$$

In effect this can be considered as a regularization scheme, where new parameters $\tilde{\theta}_i$ are introduced as Lagrange multipliers with associated derivatives:[9]

$$
\tilde{\ell}_{;i} \equiv \tilde{\theta}_i + |G_i| - \chi_i
\tag{119}
$$

and we consider a sufficiently small neighborhood of $\tilde{\theta}_i = 0$. In this setting the nontrivial components of $\tilde{\boldsymbol{H}}_z^\alpha$ take the much more manageable form:

$$
\tilde{\boldsymbol{H}}_z^\alpha = N_\alpha^{-1} \sqrt{\boldsymbol{\Sigma}^\alpha} \boldsymbol{W}^\alpha \sqrt{\boldsymbol{\Sigma}^\alpha},
\tag{120}
$$

where $\boldsymbol{\Sigma}^\alpha$ is a diagonal matrix with entries i.i.d. $\chi^2$-distributed with $\max \left( 2, \beta_\alpha \right)$ degrees of freedom. Analyzing the expected determinant of this random matrix leads us to prove the following.

**Corollary 24** (Density of local minima to multiplicative leading order, regularized). *Consider the setting of Corollary 23 conditioned on $|G_i| = \chi_i$ such that $\tilde{\boldsymbol{H}}_z^\alpha$ is as in Eq. (120). Let $\mu_{MP}^\gamma$ be the Marčenko–Pastur distribution with parameter $\gamma$. Then the expected density of local minima of $\hat{\ell}$ at a loss function value $z$ is:*

$$
\mathbb{E} \left[ \mathrm{Crt} \left( z \right) \right] = \underset{\alpha : \gamma_\alpha < 1}{\text{\Large $\ast$}} \, \mathbb{E} \left[ \mathrm{Crt}_0^\alpha \right] \left( z \right),
\tag{121}
$$

*where:*

$$
\begin{aligned}
p^{-1} \ln \left( \mathbb{E} \left[ \mathrm{Crt}_0^\alpha \left( z \right) \right] \right) = & \ln \left( \frac{\pi \max \left( 2, \beta_\alpha \right)}{2 \sqrt{\beta_\alpha}} \right) + \frac{1}{2\gamma_\alpha} \left( 1 - \frac{z}{\mathrm{Tr} \left( \boldsymbol{\rho}^\alpha \right) \bar{o}_\alpha} + \ln \left( \frac{z}{\mathrm{Tr} \left( \boldsymbol{\rho}^\alpha \right) \bar{o}_\alpha} \right) \right) \\
& + \frac{\max \left( 2, \beta_\alpha \right)}{2} - 1 - \gamma + \int \mathrm{d}\mu_{MP}^{\gamma_\alpha} \left( \lambda \right) \ln \left( \lambda \right) + \mathrm{o} \left( 1 \right).
\end{aligned}
\tag{122}
$$

---

[9]The Kac–Rice formula as stated in Adler & Taylor (2007) allows one to consider this modified gradient jointly with the original loss.

*Proof.* As in Corollary 23 we focus on a single simple component $\mathcal{A}_\alpha$. $\boldsymbol{\Sigma}$ is positive definite with probability 1. As it is diagonal with i.i.d. $\chi^2$-distributed random variables each with $\max(2,\beta)$ degrees of freedom,

$$\exp\left(p\int \mathrm{d}\mu_{\boldsymbol{\Sigma}}(\lambda)\ln(\lambda)\right) = \exp\left(p\left(\frac{\max(2,\beta)}{2}-1-\gamma+\ln(2)\right)\right), \tag{123}$$

where $\mathrm{d}\mu_{\boldsymbol{\Sigma}}$ is the empirical spectral distribution of $\boldsymbol{\Sigma}$ and $\gamma$ is the Euler–Mascheroni constant. What remains to be considered is the spectrum of

$$\boldsymbol{D}_z \equiv \boldsymbol{\Sigma}^{-\frac{1}{2}}\tilde{\boldsymbol{H}}_z\boldsymbol{\Sigma}^{-\frac{1}{2}} = N^{-1}\boldsymbol{W} = \left(\frac{\beta r}{N}\right)(\beta r)^{-1}\boldsymbol{W}. \tag{124}$$

Recall that by assumption $\frac{\sigma_o}{\overline{o}} = \mathrm{o}(1)$ so by, e.g., Lemma 25 (proved in Appendix D.1):

$$\ln\left(\frac{\beta r}{N}\right) = \mathrm{o}(1). \tag{125}$$

We need only focus on $(\beta r)^{-1}\boldsymbol{W}$, then. As discussed in Appendix A.4, this random matrix has empirical eigenvalue spectrum weakly converging almost surely to the Marčenko–Pastur distribution with parameter $\gamma$. In principle, large deviations in this convergence—even if they occur with exponentially small probability—can contribute corrections to the expected determinant due to its exponential sensitivity on the eigenvalues of $\boldsymbol{D}_z$. However, we show in Appendix L that these large deviations are dominated in the expectation by the Marčenko–Pastur distribution. Noting that the Marčenko–Pastur distribution with parameter $\gamma$ has support at the origin if and only if $\gamma > 1$ and taking convolutions over simple algebraic sectors as in the proof of Corollary 23 then yields the final result. $\qquad\square$

Dropping multiplicatively subleading factors from Eq. (122), we effectively have demonstrated that the density $\kappa_\alpha(z)$ of local minima for a given simple component $\mathcal{A}_\alpha$ is asymptotically given by:

$$\kappa_\alpha(z) = f_\Gamma\left(\frac{z}{\mathrm{Tr}(\boldsymbol{\rho}^\alpha)\,\overline{o}^\alpha}; \frac{p_\alpha}{2\gamma_\alpha}, \frac{2\gamma_\alpha}{p_\alpha}\right) \tag{126}$$

if $\gamma_\alpha < 1$, and otherwise:

$$\kappa_\alpha(z) = \delta(z). \tag{127}$$

We here have used the expression for the density of the gamma distribution:

$$f_\Gamma(x; k, \theta) = \frac{1}{\Gamma(k)\theta^k}x^{k-1}\exp\left(-\frac{x}{\theta}\right). \tag{128}$$

Convolving $\kappa_\alpha$ over many simple sectors thus yields the final density for $z > 0$:

$$\kappa(z) = \underset{\alpha:\gamma_\alpha<1}{\scalebox{1.5}{$\ast$}}\, f_\Gamma\left(\frac{z}{\overline{o}^\alpha\,\mathrm{Tr}(\boldsymbol{\rho}^\alpha)}; \frac{\beta_\alpha r_\alpha}{2}, \frac{2}{\beta_\alpha r_\alpha}\right). \tag{129}$$

This distribution is illustrated in Figure 1(C) for various parameter regimes. To multiplicative leading order in $\gamma$ this agrees exactly with the asymptotic local minima distribution studied in Anschuetz (2022); Anschuetz & Kiani (2022). See for instance Eq. (1) of Anschuetz (2022), which studies the case $\mathcal{A} = \mathcal{H}^N(\mathbb{C})$; i.e., one takes $\beta_\alpha \to 2$, $r_\alpha \to m$, and $\frac{z}{\mathrm{Tr}(\boldsymbol{\rho}^\alpha)\overline{o}_\alpha} \to x$ to translate from our setting to their setting.

We can simplify this expression even further by noting that, asymptotically, the convolution of many gamma distributions is also gamma-distributed by the Welch–Satterthwaite equation (Satterthwaite, 1946; Welch, 1947). This yields:

$$\kappa(z) = f_\Gamma(z; k_{\mathrm{eff}}, \theta_{\mathrm{eff}}), \tag{130}$$

where

$$k_{\mathrm{eff}} = \frac{\left(\sum_{\alpha:\gamma_\alpha<1}\overline{o}^\alpha\,\mathrm{Tr}(\boldsymbol{\rho}^\alpha)\right)^2}{\sum_{\alpha:\gamma_\alpha<1}\frac{2(\overline{o}^\alpha)^2\,\mathrm{Tr}(\boldsymbol{\rho}^\alpha)^2}{\beta_\alpha r_\alpha}} \xrightarrow{\{N_\alpha\to\infty\}_\alpha} \overline{\ell}_{\mathrm{u.p.}}^2\left(\sum_{\alpha:\gamma_\alpha<1}\frac{\mathrm{Tr}\left((\boldsymbol{O}^\alpha)^2\right)\mathrm{Tr}\left((\boldsymbol{\rho}^\alpha)^2\right)}{\dim_{\mathbb{R}}(\mathfrak{g}_\alpha)}\right)^{-1}, \tag{131}$$

$$\theta_{\mathrm{eff}} = \frac{\sum_{\alpha:\gamma_\alpha<1}\frac{2(\overline{o}^\alpha)^2\,\mathrm{Tr}(\boldsymbol{\rho}^\alpha)^2}{\beta_\alpha r_\alpha}}{\sum_{\alpha:\gamma_\alpha<1}\overline{o}^\alpha\,\mathrm{Tr}(\boldsymbol{\rho}^\alpha)} \xrightarrow{\{N_\alpha\to\infty\}_\alpha} \overline{\ell}_{\mathrm{u.p.}}^{-1}\sum_{\alpha:\gamma_\alpha<1}\frac{\mathrm{Tr}\left((\boldsymbol{O}^\alpha)^2\right)\mathrm{Tr}\left((\boldsymbol{\rho}^\alpha)^2\right)}{\dim_{\mathbb{R}}(\mathfrak{g}_\alpha)}. \tag{132}$$

Here,

$$\overline{\ell}_{\text{u.p.}} \equiv \sum_{\alpha:\gamma_\alpha<1} \overline{o}^\alpha \operatorname{Tr}(\boldsymbol{\rho}^\alpha) \tag{133}$$

is the mean loss function value over the underparameterized sectors, and the limit in each line is due to the identities (recalling that here we assume that $\boldsymbol{\rho}^\alpha$ is rank-1 in its defining representation):

$$(\overline{o}^\alpha)^2 \operatorname{Tr}(\boldsymbol{\rho}^\alpha)^2 = I_\alpha^2 (\overline{o}^\alpha)^2 \operatorname{Tr}_\alpha\left((\boldsymbol{\rho}^\alpha)^2\right) = I_\alpha (\overline{o}^\alpha)^2 \operatorname{Tr}\left((\boldsymbol{\rho}^\alpha)^2\right), \tag{134}$$

$$I_\alpha(\overline{o}^\alpha)^2 = \frac{I_\alpha}{N_\alpha^2}\operatorname{Tr}_\alpha(\boldsymbol{O}^\alpha)^2 = \frac{I_\alpha r_\alpha}{N_\alpha^2}\operatorname{Tr}_\alpha\left((\boldsymbol{O}^\alpha)^2\right) = \frac{r_\alpha}{N_\alpha^2}\operatorname{Tr}\left((\boldsymbol{O}^\alpha)^2\right), \tag{135}$$

as well as the identities considered in Appendix C.1.

Intriguingly, the relevant features of this density in the underparameterized regime are controlled by the $\mathcal{A}_\alpha$-purities of both $\boldsymbol{O}$ and $\boldsymbol{\rho}$ in the sectors in which they are underparameterized. This is the same quantity which controls the variance of the loss function (see Appendix C.1). However, even when there are no barren plateaus in the loss landscape—for instance, if the variance of the loss function are polynomially vanishing in $N$—the density $\kappa(z)$ may still have exponentially small measure near $z = 0$ as the gamma distribution has exponential tails.

We reemphasize that our calculation of the local minima density was performed assuming the variance of the spectral distribution of each $\boldsymbol{O}^\alpha$ (in units of the mean eigenvalue) asymptotically vanishes. This was also the setting studied in previous work on the local minima of QNNs (Anschuetz, 2022; Anschuetz & Kiani, 2022). This assumption allows us to dramatically simplify the Hessian to the form given in Theorem 19. Though it holds for low-weight fermionic (Feng et al., 2019) and local spin Hamiltonians (Erdős & Schröder, 2014), it does *not* hold for the Gaussian unitary ensemble (GUE) or nonlocal spin systems; these systems are also known to have efficient quantum algorithms that prepare their low-energy states, unlike their local cousins (Chen et al., 2024). We hope in the future to analyze whether this property also has an impact on the behavior of local minima of QNNs.

We finally note an interesting connection between our results and algorithmic hardness. We here only calculate the expected density of local minima at a given function value. If the second moment of the local minima density is also sufficiently well-behaved, then it could be the case that the asymptotic density of local minima has a fixed distributional form; this is the case in (classical) spherical spin glass models (Subag, 2017). The function value at which these local minima proliferate w.h.p. is conjectured to also hold as a *general* (i.e., beyond gradient descent) *algorithmic threshold*, that is, it is generically believed to be the function value at which better approximations to the ground state become algorithmically intractable to find. This conjecture is known to be true for the specific case of pure spherical spin glass models (Huang & Sellke, 2023). Studying second moments of quantum spin glass local minima distributions may thus be a tractable avenue for studying the algorithmic hardness of quantum problems.

# D  PROOFS OF THE MAIN RESULTS

## D.1  PRELIMINARIES

We now give in full detail the proofs of the main results discussed in Appendix B.2. We begin by giving definitions and notational conventions that we will use throughout our proofs. Recall that, given a JAWS $\mathcal{J} = \left(\bigoplus_\alpha \mathcal{A}_\alpha, \mathcal{T}, \mathcal{G}, \mathcal{H}, \boldsymbol{A}, \boldsymbol{O}\right)$, we are interested in the joint distribution of the loss $\ell(\boldsymbol{\theta}; \boldsymbol{\rho})$ and its first two derivatives over a set of input states $\boldsymbol{\rho} \in \mathcal{R}$, where:

$$\ell(\boldsymbol{\theta}; \boldsymbol{\rho}) = \sum_\alpha \ell^\alpha(\boldsymbol{\theta}; \boldsymbol{\rho}) \equiv \sum_\alpha I_\alpha \operatorname{Tr}_\alpha\left(\boldsymbol{\rho}^\alpha \boldsymbol{U}^\alpha(\boldsymbol{\theta})\boldsymbol{O}^\alpha \boldsymbol{U}^{\alpha\dagger}(\boldsymbol{\theta})\right) \tag{136}$$

and

$$\boldsymbol{U}^\alpha(\boldsymbol{\theta}) \equiv \boldsymbol{g}_0^{\alpha\dagger}\left(\prod_{i=1}^p \boldsymbol{g}_i^\alpha \exp\left(\theta_i \boldsymbol{A}_i^\alpha\right)\boldsymbol{g}_i^{\alpha\dagger}\right)\boldsymbol{h}^\alpha \tag{137}$$

with $\boldsymbol{g}_i^\alpha, \boldsymbol{h}^\alpha \in G_\alpha$, and $\boldsymbol{A}_i^\alpha \in \mathfrak{g}_\alpha$. As previously discussed we will use $\cdot^\alpha$ to denote the defining representation of the projection of $\cdot$ into $\mathcal{A}_\alpha$. As each sector labeled by $\alpha$ is independent we will here only consider a single $\alpha$ WLOG, with nonzero $\boldsymbol{\rho}^\alpha$ and $\boldsymbol{O}^\alpha$. This will also allow us to remove

the cumbersome notation of labeling all objects with the index $\alpha$ for the remainder of this section. To further simplify the language, we will use the term "unitary" to refer to "orthogonal," "unitary," or "hyperunitary" in the context of $\mathbb{F} = \mathbb{R}, \mathbb{C}, \mathbb{H}$, respectively, unless otherwise explicitly stated. Similarly, we will later see that our results hold for any $\boldsymbol{\theta}$; we will thus leave the $\boldsymbol{\theta}$-dependence of $\ell$ implicit from here on out to save on notation. In the following we will use $|\mu\rangle$ (i.e., Greek letters), $0 \leq \mu \leq N_\alpha - 1$ to denote basis vectors in the vector space on which the defining representation of $\mathcal{H}^{N_\alpha}(\mathbb{F}_\alpha)$ acts. We will later use $|i\rangle$ (i.e., Latin letters), $0 \leq i \leq p$ to denote vectors in the vector space on which the defining representation of $\mathcal{H}^{p+1}(\mathbb{F}_\alpha)$ acts.

We now detail our choice of $\boldsymbol{A}_i$ given Assumption 16. Assumption 16 has a nice interpretation as taking the $\boldsymbol{A}_i$ to be low-rank (representations of) basis elements of the Cartan subalgebra $\mathfrak{h} \subseteq \mathfrak{g}$. Up to the adjoint action of $G$ and an overall normalization (which can be absorbed into $\theta_i$), then, we can take WLOG:

$$\boldsymbol{A}_i = |0\rangle\langle 1| - |1\rangle\langle 0| \tag{138}$$

when $\mathbb{F} = \mathbb{R}$ and

$$\boldsymbol{A}_i = \mathrm{i}\,|0\rangle\langle 0| \tag{139}$$

when $\mathbb{F} = \mathbb{C}$. When $\mathbb{F} = \mathbb{H}$ we will take a parameterization of the form:

$$\exp\left(\theta_i \boldsymbol{A}_i\right) = \exp\left(\mathrm{i}\theta_i^{(\mathrm{i})}\,|0\rangle\langle 0|\right)\exp\left(\mathrm{j}\theta_i^{(\mathrm{j})}\,|0\rangle\langle 0|\right)\exp\left(\mathrm{k}\theta_i^{(\mathrm{k})}\,|0\rangle\langle 0|\right); \tag{140}$$

that is, we will assume we have full control over the quaternionic phase. We have chosen here for the $\boldsymbol{A}_i$ to be $i$-independent for convenience, moving any $i$-dependence to the conjugating unitaries of each layer $\boldsymbol{g}_i^\alpha \in G_\alpha$.

Finally, before continuing we give a convenient relation between

$$r = \frac{\|\boldsymbol{O}\|_*^2}{\|\boldsymbol{O}\|_{\mathrm{F}}^2} \tag{141}$$

and the standard deviation $\sigma_o$ of the eigenvalues of $\boldsymbol{O}$. This relation will be used to simplify some of our later expressions.

**Lemma 25** ($r$ and $\sigma_o$ relation). *Let $\boldsymbol{O}$ be an $N \times N$ Hermitian operator and $r$ be as in Eq. (141). Let $\sigma_o$ be the standard deviation of the eigenvalues of $\boldsymbol{O}$, and $\bar{o}$ the arithmetic mean. Then:*

$$\sqrt{\frac{N}{r} - 1} = \frac{\sigma_o}{\bar{o}}. \tag{142}$$

*Proof.* Let $\bar{o}_{\mathrm{RMS}}$ be the root mean square of the eigenvalues of $\boldsymbol{O}$. Then:

$$\begin{aligned}
\sqrt{\frac{N}{r} - 1} &= \sqrt{\frac{N\|\boldsymbol{O}\|_{\mathrm{F}}^2}{\|\boldsymbol{O}\|_*^2} - 1} \\
&= \sqrt{\frac{\bar{o}_{\mathrm{RMS}}^2}{\bar{o}^2} - 1} \\
&= \frac{\sigma_o}{\bar{o}}.
\end{aligned} \tag{143}$$

$\square$

## D.2 ASYMPTOTIC EXPRESSION FOR THE LOSS

We now proceed with the proofs of our main results. This subsection is devoted to a series of reductions that will allow us to consider the $\boldsymbol{h}\boldsymbol{O}\boldsymbol{h}^\dagger$ as $\beta$-Wishart matrices up to a controlled error in Lévy–Prokhorov metric, thus proving Theorem 13. Along the way we will also prove a reduction to taking the $\boldsymbol{g}_i$ to be Haar random, once again up to some bounded error in Lévy–Prokhorov metric. In proving these results we will heavily rely on various lemmas on convergence in distribution given in Appendix M.

### D.2.1 REDUCTION TO HAAR RANDOM $\boldsymbol{h}, \boldsymbol{g}_i$

We first argue that, under Assumption 12, the $\boldsymbol{h}$ in Eq. (137) can be assumed to be Haar random over $G$ up to some bounded error in Lévy–Prokhorov metric. We also show that this remains true when scaling by any choice of $N$-dependent normalization $\mathcal{N}$ with the only requirement being that central moments of large (constant) order have a well-defined, finite limit as $N \to \infty$; for which maximal choice of $\mathcal{N}$ this is true depends on the exact form of the $\epsilon$-approximate $t$-design, but is always true for $\mathcal{N} = 1$. In the following we implicitly consider a sequence of objects as $\epsilon^{-1}, t, N \to \infty$.

**Lemma 26** (Weak convergence to Haar random $\boldsymbol{h}$). *Assume $\boldsymbol{h}$ forms an $\epsilon$-approximate $t$-design. Let $L_i$, $i \in [d]$ be multilinear functions of the form:*

$$L_i = \mathrm{Tr}\left(\boldsymbol{M}_i \boldsymbol{h} \boldsymbol{O} \boldsymbol{h}^\dagger\right), \tag{144}$$

*where the $\boldsymbol{M}_i$ have bounded operator norm and $\boldsymbol{h} \sim \mathcal{H}$. Let $\tilde{L}_i$ be the same with $\boldsymbol{h} \to \tilde{\boldsymbol{h}}$, where $\tilde{\boldsymbol{h}}$ is Haar random. Assume any $N$-dependent normalization $\mathcal{N} \leq \mathrm{O}(N)$, and assume that there exists a constant $k^*$ such that all central moments of constant order $k > k^*$ of the $\mathcal{N} L_i$ have a finite limit as $\epsilon^{-1}, t, N \to \infty$. Then the joint distribution of $\mathcal{N} L_i$ differs from that of $\mathcal{N} \tilde{L}_i$ by an error at most*

$$\pi_1 = \mathrm{O}\left(\frac{d \log\log\left(\mathcal{N}^{-t}\epsilon^{-1}\right)}{\log\left(\mathcal{N}^{-t}\epsilon^{-1}\right)} + \frac{d \log(t)}{\sqrt{t}}\right) \tag{145}$$

*in Lévy–Prokhorov metric as $\epsilon^{-1}, t, N \to \infty$.*

*Proof.* As the $L_i - \mathbb{E}[L_i]$ are bounded random variables, there exists some constant $C > 0$ such that:

$$\mathbb{E}\left[\prod_{i \in s_k}\left(L_i - \mathbb{E}[L_i]\right)\right] \leq \left(C\sqrt{k}\right)^k \tag{146}$$

for all index multisets $s_k$ of cardinality $k$. By the given assumption on $\mathcal{N}$ a similar bound also holds for central moments of the $\mathcal{N} L_i$ of sufficiently high order. The bound $\mathcal{N} = \mathrm{O}(N)$ ensures that this central moment bound also holds true for the $\mathcal{N} \tilde{L}_i$ by sub-Gaussianity (Meckes, 2019; Vershynin, 2018), and also ensures that the $\mu$ of Corollary 38 is subleading to Eq. (145). The conditions of Corollary 38 are then satisfied and the final result yielded. $\square$

By incurring this error in Lévy–Prokhorov metric we now can assume that the $\boldsymbol{h}$ (given Assumption 12) are Haar random when considering just the distribution of the loss (so $d = 1$ in Lemma 26). When considering the loss, gradient, and Hessian jointly, the addition of Assumption 15 means that we can assume that both the $\boldsymbol{h}$ and the $\boldsymbol{g}_i$ are Haar random. We note, however, that up to requiring a worse bound on $\mathcal{N}$ (i.e., $\mathcal{N} \leq \mathrm{o}\left(N^{\frac{2}{3}}\right)$) the $\boldsymbol{g}_i$ being drawn from an $\epsilon$-approximate 2-design suffices. This argument is given in detail in Appendix J.

When the $\boldsymbol{h}$ and $\boldsymbol{g}_i$ are reduced to being drawn i.i.d. from the Haar distribution it is apparent that the joint distribution of the loss with its first two derivatives is invariant under translations of the parameters. This justifies us fixing $\boldsymbol{\theta} = \boldsymbol{0}$ in the sequel.

### D.2.2 REDUCTION TO GAUSSIAN $\boldsymbol{h}, \boldsymbol{g}_i$

We now show that certain marginal distributions of the the entries of $\sqrt{\mathcal{N}}\boldsymbol{h}, \sqrt{\mathcal{N}}\boldsymbol{g}_i$ can be approximated as random Gaussian matrices over $\mathbb{F}$ up to a small error in Ky Fan metric whenever $\mathcal{N} \leq \mathrm{O}(N)$. This will allow us to replace $\sqrt{\mathcal{N}}\boldsymbol{h}, \sqrt{\mathcal{N}}\boldsymbol{g}_i$ with random matrices with i.i.d. Gaussian entries, simplifying our results further.

**Lemma 27** (Convergence in probability of Haar marginals to Gaussian matrices). *Let $L_i$, $i \in [d]$ be uniformly bounded multilinear functions of the form:*

$$L_i = \mathrm{Tr}\left(\boldsymbol{M}_i \boldsymbol{h} \boldsymbol{\rho} \boldsymbol{h}^\dagger\right), \tag{147}$$

*where there exists some $D$ such that*

$$\frac{1}{\mathrm{Tr}(\boldsymbol{\rho}^2)} \leq D = \mathrm{o}\left(\frac{N}{\log(N)}\right). \tag{148}$$

*Assume as well that $d \leq \mathrm{O}\left(\mathrm{poly}\left(N\right)\right)$ and the $\boldsymbol{M}_i$ are independent from the $\boldsymbol{h}$. Let $\tilde{L}_i$ be the same, where now the $\boldsymbol{h}$ are random matrices $\boldsymbol{G}$ with i.i.d. standard Gaussian entries over $\mathbb{F}$. For sufficiently large $N$, the joint distribution of $\mathcal{N} L_i$ differs from the joint distribution of $\frac{N}{N}\tilde{L}_i$ by an error in Ky Fan metric of at most:*

$$\alpha_3 = \mathrm{O}\left(\frac{\sqrt{\mathcal{N}D\log\left(N\right)}}{N} + \frac{\mathcal{N}}{D}\right) = \mathrm{o}\left(1\right) \tag{149}$$

*for all $\mathcal{N} \leq \mathrm{O}\left(N\right)$. Alternatively, if $\mathrm{rank}\left(\boldsymbol{\rho}\right) \leq D$,*

$$\alpha_3 = \mathrm{O}\left(\frac{\sqrt{\mathcal{N}D\log\left(N\right)}}{N}\right) = \mathrm{o}\left(1\right). \tag{150}$$

*Proof.* We assume $\boldsymbol{\rho}$ has trace norm 1 by absorbing $\mathrm{Tr}\left(\boldsymbol{\rho}\right)$ into $\boldsymbol{M}_i$. We first argue that if the high-purity condition is assumed then $\boldsymbol{\rho}$ has an approximation to small error in trace norm that is low-rank. To see this, let $r_i$ be the eigenvalues of $\boldsymbol{\rho}$ in non-increasing order. If

$$\sum_{i=D+1}^{N_\alpha} r_i \geq \frac{1}{D+1} \tag{151}$$

then it must be that $\boldsymbol{\rho}$ has low purity, i.e.,

$$\mathrm{Tr}\left(\boldsymbol{\rho}^2\right) \leq \frac{D+1}{\left(D+1\right)^2} = \frac{1}{D+1} < \frac{1}{D}, \tag{152}$$

breaking our assumption on the purities of the inputs (Eq. (148)). Thus,

$$\sum_{i=D+1}^{N_\alpha} r_i < \frac{1}{D+1}. \tag{153}$$

In particular, $\boldsymbol{\rho}$ has a rank-$D$ approximation that agrees up to an $\mathrm{O}\left(\frac{1}{D}\right)$ additive error in trace distance.

Consider now the low-rank case. Let

$$\delta \equiv \frac{D\ln\left(N\right)}{N}. \tag{154}$$

$\delta < \frac{1}{4}$ for sufficiently large $N$ by the assumed scaling of $D$. Following the proof of Corollary 1.1 in Jiang (2010), then, for sufficiently large $N$ and $t > 10\sqrt{\frac{N\delta}{N}}$,

$$\mathbb{P}\left[\max_{\substack{i\in[N]\\j\in[D]}}\left|\sqrt{N}h_{i,j} - G_{i,j}\right| \geq \sqrt{\frac{N}{\mathcal{N}}}t\right] = \exp\left(-\Omega\left(\log\left(N\right)^{\frac{3}{2}}\right)\right) + \exp\left(-\Omega\left(\frac{N^2}{\mathcal{N}D}\right)\right). \tag{155}$$

The right-hand side decays superpolynomially with $N$ whenever $\mathcal{N} \leq \mathrm{O}\left(N\right)$. This implies the error in Ky Fan metric is dominated by the cutoff at $t = \Theta\left(\sqrt{\frac{N\delta}{N}}\right)$. This in conjunction with the error from the initial low-rank approximation yields the final result. $\qquad\square$

We will mostly be concerned with the specific case when $D = N^{0.999}$, yielding for $\mathcal{N} \leq \mathrm{O}\left(N^{0.99}\right)$:

$$\alpha_3 = \mathrm{O}\left(\sqrt{\frac{\mathcal{N}\log\left(N\right)}{N^{1.001}}}\right) = \mathrm{o}\left(1\right). \tag{156}$$

### D.2.3  REDUCTION TO SEMI-ISOTROPIC $\boldsymbol{O}$

We end with a reduction to a more convenient form for the spectrum of $\boldsymbol{O}$. Let $|\mu\rangle$ be the eigenbasis of $\boldsymbol{O}$. Consider the Hermitian $\tilde{\boldsymbol{O}}$:

$$\tilde{\boldsymbol{O}} \equiv k \sum_{\mu=0}^{r-1} |\mu\rangle \langle\mu| , \tag{157}$$

where

$$r \equiv \frac{\|\boldsymbol{O}\|_*^2}{\|\boldsymbol{O}\|_{\mathrm{F}}^2} \tag{158}$$

is assumed to be an integer and (for $\overline{o}$ the mean eigenvalue of $o$)

$$k \equiv \frac{\|\boldsymbol{O}\|_*}{r} = \frac{N\overline{o}}{r}. \tag{159}$$

By construction

$$\mathrm{Tr}\left(\boldsymbol{O}\right) = \mathrm{Tr}\left(\tilde{\boldsymbol{O}}\right), \tag{160}$$

$$\mathrm{Tr}\left(\boldsymbol{O}^2\right) = \mathrm{Tr}\left(\tilde{\boldsymbol{O}}^2\right). \tag{161}$$

We claim that replacing $\boldsymbol{O}$ with $\tilde{\boldsymbol{O}}$ incurs only a vanishingly small error in Ky Fan metric. Intuitively this follows from the Welch–Satterthwaite approximation of a weighted sum of Wishart random matrices as a single Wishart random matrix (Khuri et al., 1994). We formally state this as the following lemma.

**Lemma 28** (Reduction to semi-isotropic $\boldsymbol{O}$). *Let $L_i$, $i \in [d]$ be of the form:*

$$L_i = \mathrm{Tr}\left(\boldsymbol{M}_i \boldsymbol{X}\left(\boldsymbol{O} - \tilde{\boldsymbol{O}}\right)\boldsymbol{X}^\dagger\right), \tag{162}$$

*where $\sqrt{N}\boldsymbol{X}$ has i.i.d. standard Gaussian entries over $\mathbb{F}$, $\boldsymbol{M}_i$ is independent from $\boldsymbol{X}$ with bounded trace norm, and $d = \mathrm{O}\left(\mathrm{poly}\left(N\right)\right)$. Let $\tilde{L}_i$ be the same multilinear functions, where instead of $\boldsymbol{O}$ one has $\tilde{\boldsymbol{O}}$ as defined in Eq. (157). The joint distribution of $\mathcal{N}L_i$ over $\boldsymbol{X}$ differs from $\boldsymbol{0}$ by an error at most*

$$\alpha_4 = \mathrm{O}\left(\frac{\mathcal{N}\Delta\log\left(\frac{dN}{\mathcal{N}\Delta}\right)}{N}\right) \tag{163}$$

*in Ky Fan metric, where*

$$\Delta \equiv 1 - \frac{\overline{o}}{\|\boldsymbol{O}\|_{op}} - \frac{\sigma_o^2}{\|\boldsymbol{O}\|_{op}\overline{o}}. \tag{164}$$

*Here, $o_i$ is the $i$th eigenvalue of $\boldsymbol{O}$ in non-increasing order, $\sigma_o$ the standard deviation of the eigenvalues of $\boldsymbol{O}$, and $\overline{o}$ the mean.*

*Proof.* $\boldsymbol{O}$ and $\tilde{\boldsymbol{O}}$ can be assumed to be diagonal WLOG as $\boldsymbol{X}$ is unitarily invariant. Therefore, $\boldsymbol{X}\left(\boldsymbol{O} - \tilde{\boldsymbol{O}}\right)\boldsymbol{X}^\dagger$ can be written as a weighted sum of standard $\beta$-Wishart matrices each with a single degree of freedom:

$$\boldsymbol{X}\left(\boldsymbol{O} - \tilde{\boldsymbol{O}}\right)\boldsymbol{X}^\dagger = \frac{1}{N}\sum_{i=1}^{N}(o_i - \tilde{o}_i)|x_i\rangle\langle x_i| , \tag{165}$$

where $|x_i\rangle$ is the (unnormalized) $i$th column of $\sqrt{N}\boldsymbol{X}$ and $o_i, \tilde{o}_i$ are the eigenvalues of $\boldsymbol{O}, \tilde{\boldsymbol{O}}$, respectively, in nonincreasing order. Note that for any normalized $|\mu\rangle$, $|\langle x_i|\mu\rangle|$ is gamma-distributed. Thus, from a generalization of Bernstein's inequality (see, e.g., Theorem 2.8.1 of Vershynin (2018)), for any $|\mu\rangle$,

$$\mathbb{P}\left[\left|\frac{1}{N}\sum_{i=1}^{N}(o_i - \tilde{o}_i)|\langle x_i|\mu\rangle|^2\right| \geq \epsilon\right]$$

$$\leq 2\exp\left(-cN\min\left(\frac{\epsilon}{\max_i|o_i - \tilde{o}_i|}, \frac{\epsilon^2}{\frac{1}{N}\sum_{i=1}^{N}(o_i - \tilde{o}_i)^2}\right)\right) \tag{166}$$

for some constant $c > 0$. Due to the equality of traces of $\boldsymbol{O}$ and $\tilde{\boldsymbol{O}}$ we calculate:

$$
\begin{aligned}
\frac{1}{N} \sum_{i=1}^{N} (o_i - \tilde{o}_i)^2 &= \frac{1}{N} \operatorname{Tr}\left(\left(\boldsymbol{O} - \tilde{\boldsymbol{O}}\right)^2\right) \\
&= \frac{2}{N}\left(\operatorname{Tr}\left(\boldsymbol{O}^2\right) - \operatorname{Tr}\left(\boldsymbol{O}\tilde{\boldsymbol{O}}\right)\right) \\
&\leq \frac{2}{N}\left(\operatorname{Tr}\left(\boldsymbol{O}^2\right) - \operatorname{Tr}\left(\boldsymbol{O}\right)k + ko_{r+1}\right) \\
&= \frac{2}{r}\overline{o}o_{r+1} \\
&= \mathrm{O}\left(\frac{o_{r+1}}{N}\right).
\end{aligned}
\tag{167}
$$

Furthermore,

$$
\begin{aligned}
\max_i |o_i - \tilde{o}_i| &= \mathrm{O}\left(o_1 - k\right) + \mathrm{O}\left(o_{r+1}\right) \\
&= \mathrm{O}\left(o_1 - \overline{o} - \frac{\sigma_o^2}{\overline{o}} + o_{r+1}\right) \\
&= \mathrm{O}\left(1 - \frac{\overline{o}}{\|\boldsymbol{O}\|_{\mathrm{op}}} - \frac{\sigma_o^2}{\|\boldsymbol{O}\|_{\mathrm{op}}\overline{o}}\right).
\end{aligned}
\tag{168}
$$

The desired convergence follows by the union bound and taking $\epsilon \to \frac{\epsilon}{N}$. $\qquad\square$

We now claim these suffice to prove Theorem 13.

*Proof of Theorem 13.* For $\boldsymbol{X}$ with i.i.d. standard Gaussian entries over $\mathbb{F}$ and $\tilde{\boldsymbol{O}}$ as in Lemma 28, $k^{-1}\boldsymbol{X}\tilde{\boldsymbol{O}}\boldsymbol{X}^\dagger$ is $\beta$-Wishart-distributed with $r$ degrees of freedom. The result then follows from Lemmas 26, 27, and 28 applied in sequence, with a final error in Lévy–Prokhorov metric of:

$$
\pi = \pi_1 + \alpha_3 + \alpha_4 = \mathrm{O}\left(\frac{\log\log\left(\mathcal{N}^{-t}\epsilon^{-1}\right)}{\log\left(\mathcal{N}^{-t}\epsilon^{-1}\right)} + \frac{\log(t)}{\sqrt{t}} + \sqrt{\frac{\mathcal{N}\log(N)}{N^{1.001}}}\right).
\tag{169}
$$

$\qquad\square$

## D.3 ASYMPTOTIC EXPRESSIONS FOR THE FIRST TWO DERIVATIVES

We will now move on and construct an explicit asymptotic expression for the first two derivatives of the loss function. As previously mentioned, we need only consider $\boldsymbol{\theta} = \boldsymbol{0}$ by Lemma 26. Here we have that the first derivatives are distributed as:

$$
\partial_i \ell = I \operatorname{Tr}\left(\rho\left[\boldsymbol{g}_i \boldsymbol{A}_i \boldsymbol{g}_i^\dagger, \boldsymbol{h}\boldsymbol{O}\boldsymbol{h}^\dagger\right]\right).
\tag{170}
$$

The second derivatives are of the form (for $i \geq j$):

$$
\partial_i \partial_j \ell = I \operatorname{Tr}\left(\rho\left[\boldsymbol{g}_j \boldsymbol{A}_j \boldsymbol{g}_j^\dagger\left[\boldsymbol{g}_i \boldsymbol{A}_i \boldsymbol{g}_i^\dagger, \boldsymbol{h}\boldsymbol{O}\boldsymbol{h}^\dagger\right]\right]\right).
\tag{171}
$$

Furthermore, using the lemmas from Appendix D.2, up to a cost:

$$
\pi_1 + \alpha_3 + \alpha_4 = \mathrm{O}\left(\frac{d\log\log\left(\mathcal{N}^{-t}\epsilon^{-1}\right)}{\log\left(\mathcal{N}^{-t}\epsilon^{-1}\right)} + \frac{d\log(t)}{\sqrt{t}} + \sqrt{\frac{\mathcal{N}\log(N)}{N}}\right)
\tag{172}
$$

in Lévy–Prokhorov metric—where $d$ is either $p$ or $p^2$—we may reduce to considering:

$$
\hat{\ell} = \frac{I}{N} \operatorname{Tr}\left(\rho \boldsymbol{X}_0^\dagger \boldsymbol{O} \boldsymbol{X}_0\right),
\tag{173}
$$

$$
\hat{\ell}_{;i} = \frac{I}{N^2} \operatorname{Tr}\left(\rho \boldsymbol{X}_0^\dagger\left[\boldsymbol{X}_i \boldsymbol{A}_i \boldsymbol{X}_i^\dagger, \boldsymbol{O}\right]\boldsymbol{X}_0\right),
\tag{174}
$$

$$
\hat{\ell}_{;i,j} = \frac{I}{N^3} \operatorname{Tr}\left(\rho \boldsymbol{X}_0^\dagger\left[\boldsymbol{X}_j \boldsymbol{A}_j \boldsymbol{X}_j^\dagger, \left[\boldsymbol{X}_i \boldsymbol{A}_i \boldsymbol{X}_i^\dagger, \boldsymbol{O}\right]\right]\boldsymbol{X}_0\right),
\tag{175}
$$

where:

1. $\boldsymbol{X}_0$ is an $N \times \text{rank}(\boldsymbol{\rho})$ random matrix with i.i.d. standard Gaussian entries over $\mathbb{F}$, and the $\boldsymbol{X}_i$ are $N \times p$ or $N \times 2p$ (depending on the rank of the $\boldsymbol{A}_i$) random matrices with i.i.d. standard Gaussian entries over $\mathbb{F}$.

2. The initial states $\boldsymbol{\rho}$ are of rank at most $\text{O}\left(N^{0.999}\right)$.

Note that these quantities depend only on $M = \text{O}(\text{rank}(\boldsymbol{\rho}))$ rows of $\boldsymbol{X}_0$ and either 1 ($\mathbb{F} = \mathbb{C}, \mathbb{H}$) or 2 ($\mathbb{F} = \mathbb{R}$) rows of $\boldsymbol{X}_i$. We can therefore further simplify this by collecting all of the relevant rows of $\boldsymbol{X}_0, \boldsymbol{X}_i$ into a single $\boldsymbol{X}$ of dimensions $N \times (p + M)$ (or $N \times (2p + M)$ when $\mathbb{F} = \mathbb{R}$), writing these expressions as:[10]

$$\hat{\ell} = \frac{I\bar{o}}{r} \text{Tr}\left(\boldsymbol{\rho} \boldsymbol{X}^\dagger \tilde{\boldsymbol{O}} \boldsymbol{X}\right), \tag{176}$$

$$\hat{\ell}_{;i} = \frac{I\bar{o}}{rN} \text{Tr}\left(\boldsymbol{\rho} \boldsymbol{X}^\dagger \left[\boldsymbol{X} |i\rangle \langle i| \boldsymbol{X}^\dagger, \tilde{\boldsymbol{O}}\right] \boldsymbol{X}\right), \tag{177}$$

$$\hat{\ell}_{;i,j} = \frac{I\bar{o}}{rN^2} \text{Tr}\left(\boldsymbol{\rho} \boldsymbol{X}^\dagger \left[\boldsymbol{X} |j\rangle \langle j| \boldsymbol{X}^\dagger, \left[\boldsymbol{X} |i\rangle \langle i| \boldsymbol{X}^\dagger, \tilde{\boldsymbol{O}}\right]\right] \boldsymbol{X}\right). \tag{178}$$

Here, we took another error of $\alpha_4$ in Lévy–Prokhorov metric to change $\boldsymbol{O}$ to

$$\tilde{\boldsymbol{O}} = \sum_{\mu=0}^{r-1} |\mu\rangle \langle \mu| \tag{179}$$

as in Eq. (157). Though this expression for the full distribution is unwieldy, it completely characterizes the joint distribution of the loss and first two derivatives in terms of a single Gaussian random matrix $\boldsymbol{X}$. To simplify further we will assume $\boldsymbol{\rho}$ is a single state which is rank-1 when projected into the simple sector we are considering, taken WLOG to be $|0\rangle \langle 0|$. We will also assume in the following that $\frac{\sigma_o}{\bar{o}}$ is asymptotically bounded.

Details of further simplifications vary depending on whether $\mathbb{F}$ is algebraically closed or not (i.e., whether or not we are working in $\mathbb{R}$). We will thus consider the two cases separately.

D.3.1  $\mathbb{F} = \mathbb{C}, \mathbb{H}$

When $\mathbb{F} = \mathbb{C}, \mathbb{H}$, we are considering the joint distribution of:

$$\hat{\ell} = \frac{I\bar{o}}{r} \text{Tr}\left(|0\rangle \langle 0| \boldsymbol{X}^\dagger \tilde{\boldsymbol{O}} \boldsymbol{X}\right), \tag{180}$$

$$\hat{\ell}_{;i} = \frac{\mathrm{i}I\bar{o}}{rN} \text{Tr}\left(|0\rangle \langle 0| \boldsymbol{X}^\dagger \left[\boldsymbol{X} |i\rangle \langle i| \boldsymbol{X}^\dagger, \tilde{\boldsymbol{O}}\right] \boldsymbol{X}\right), \tag{181}$$

$$\hat{\ell}_{;i,j} = -\frac{I\bar{o}}{rN^2} \text{Tr}\left(|0\rangle \langle 0| \boldsymbol{X}^\dagger \left[\boldsymbol{X} |j\rangle \langle j| \boldsymbol{X}^\dagger, \left[\boldsymbol{X} |i\rangle \langle i| \boldsymbol{X}^\dagger, \tilde{\boldsymbol{O}}\right]\right] \boldsymbol{X}\right), \tag{182}$$

where now $\boldsymbol{X}$ is a $N \times (p + 1)$ random matrix with i.i.d. standard Gaussian entries over $\mathbb{F}$. Here, when $\mathbb{F} = \mathbb{H}$, "i" should be taken to mean one of $\mathrm{i}, \mathrm{j}, \mathrm{k}$, depending on which parameter of Eq. (140) the derivative is being taken with respect to. We take the two contributions of quaternionic phase to be equal in the second derivative expression as, by the anticommutivity of quaternions, this expression is otherwise identically zero. In particular, this abuse of notation will not change any of the following analysis unless specifically stated otherwise.

Recall that

$$\tilde{\boldsymbol{O}} = \sum_{\mu=0}^{r-1} |\mu\rangle \langle \mu|. \tag{183}$$

Let $\boldsymbol{C}$ be the complement of $\tilde{\boldsymbol{O}}$, i.e.,

$$\boldsymbol{C} = \sum_{\mu=r}^{N-1} |\mu\rangle \langle \mu|. \tag{184}$$

---

[10]We have slightly abused notation here; the $|i\rangle, |j\rangle$ here are taken to be orthogonal to the nonzero eigenvectors of $\boldsymbol{\rho}$.

In particular, $\boldsymbol{C}$ and $\tilde{\boldsymbol{O}}$ sum to the identity:

$$\boldsymbol{C} + \tilde{\boldsymbol{O}} = \boldsymbol{I}_N. \tag{185}$$

Furthermore,

$$\boldsymbol{W} \equiv \boldsymbol{X}^\dagger \tilde{\boldsymbol{O}} \boldsymbol{X}, \tag{186}$$

$$\boldsymbol{V} \equiv \boldsymbol{X}^\dagger \boldsymbol{C} \boldsymbol{X}, \tag{187}$$

are each $\beta$-Wishart-distributed matrices; the former with $r$ degrees of freedom, and the latter with $N - r$. The independence of $\boldsymbol{W}$ and $\boldsymbol{V}$ can be checked from the associated Bartlett decompositions. We thus have that the distribution we are interested in is the joint distribution of:

$$\hat{\ell} = \frac{I\bar{o}}{r} \langle 0| \boldsymbol{W} |0\rangle, \tag{188}$$

$$\hat{\ell}_{;i} = \frac{2I\bar{o}}{rN} \operatorname{Re}\left\{ \mathrm{i} \langle 0| \boldsymbol{W} |i\rangle \langle i| \boldsymbol{X}^\dagger \boldsymbol{X} |0\rangle \right\} = \frac{2I\bar{o}}{rN} \operatorname{Re}\left\{ \mathrm{i} \langle 0| \boldsymbol{W} |i\rangle \langle i| \boldsymbol{V} |0\rangle \right\}, \tag{189}$$

and:

$$
\begin{aligned}
\hat{\ell}_{;i,j} = & -\frac{I\bar{o}}{rN^2} \langle 0| \boldsymbol{X}^\dagger \left[ \boldsymbol{X} |j\rangle \langle j| \boldsymbol{X}^\dagger, \left[ \boldsymbol{X} |i\rangle \langle i| \boldsymbol{X}^\dagger, \boldsymbol{O} \right] \right] \boldsymbol{X} |0\rangle \\
= & -\frac{I\bar{o}}{rN^2} \langle 0| \boldsymbol{X}^\dagger \boldsymbol{X} |j\rangle \langle j| \boldsymbol{X}^\dagger \boldsymbol{X} |i\rangle \langle i| \boldsymbol{X}^\dagger \boldsymbol{O} \boldsymbol{X} |0\rangle \\
& -\frac{I\bar{o}}{rN^2} \langle 0| \boldsymbol{X}^\dagger \boldsymbol{O} \boldsymbol{X} |i\rangle \langle i| \boldsymbol{X}^\dagger \boldsymbol{X} |j\rangle \langle j| \boldsymbol{X}^\dagger \boldsymbol{X} |0\rangle \\
& +\frac{I\bar{o}}{rN^2} \langle 0| \boldsymbol{X}^\dagger \boldsymbol{X} |i\rangle \langle i| \boldsymbol{X}^\dagger \boldsymbol{O} \boldsymbol{X} |j\rangle \langle j| \boldsymbol{X}^\dagger \boldsymbol{X} |0\rangle \\
& +\frac{I\bar{o}}{rN^2} \langle 0| \boldsymbol{X}^\dagger \boldsymbol{X} |j\rangle \langle j| \boldsymbol{X}^\dagger \boldsymbol{O} \boldsymbol{X} |i\rangle \langle i| \boldsymbol{X}^\dagger \boldsymbol{X} |0\rangle \\
= & -\frac{I\bar{o}}{rN^2} \langle 0| (\boldsymbol{W} + \boldsymbol{V}) |j\rangle \langle j| (\boldsymbol{W} + \boldsymbol{V}) |i\rangle \langle i| \boldsymbol{W} |0\rangle \\
& -\frac{I\bar{o}}{rN^2} \langle 0| \boldsymbol{W} |i\rangle \langle i| (\boldsymbol{W} + \boldsymbol{V}) |j\rangle \langle j| (\boldsymbol{W} + \boldsymbol{V}) |0\rangle \\
& +\frac{I\bar{o}}{rN^2} \langle 0| (\boldsymbol{W} + \boldsymbol{V}) |i\rangle \langle i| \boldsymbol{W} |j\rangle \langle j| (\boldsymbol{W} + \boldsymbol{V}) |0\rangle \\
& +\frac{I\bar{o}}{rN^2} \langle 0| (\boldsymbol{W} + \boldsymbol{V}) |j\rangle \langle j| \boldsymbol{W} |i\rangle \langle i| (\boldsymbol{W} + \boldsymbol{V}) |0\rangle.
\end{aligned}
\tag{190}
$$

We can simplify the expression for the second derivative by grouping terms by their order in $\boldsymbol{W}$. Note first that all terms cubic in matrix elements of $\boldsymbol{W}$ sum to zero. For the quadratic terms, note that for the outer two terms:

$$
\begin{aligned}
& -\langle 0| \boldsymbol{V} |j\rangle \langle j| \boldsymbol{W} |i\rangle \langle i| \boldsymbol{W} |0\rangle - \langle 0| \boldsymbol{W} |j\rangle \langle j| \boldsymbol{V} |i\rangle \langle i| \boldsymbol{W} |0\rangle \\
& +\langle 0| \boldsymbol{V} |j\rangle \langle j| \boldsymbol{W} |i\rangle \langle i| \boldsymbol{W} |0\rangle + \langle 0| \boldsymbol{W} |j\rangle \langle j| \boldsymbol{W} |i\rangle \langle i| \boldsymbol{V} |0\rangle \\
= & -\langle 0| \boldsymbol{W} |j\rangle \langle j| \boldsymbol{V} |i\rangle \langle i| \boldsymbol{W} |0\rangle + \langle 0| \boldsymbol{W} |j\rangle \langle j| \boldsymbol{W} |i\rangle \langle i| \boldsymbol{V} |0\rangle,
\end{aligned}
\tag{191}
$$

and for the inner two terms:

$$
\begin{aligned}
& -\langle 0| \boldsymbol{W} |i\rangle \langle i| \boldsymbol{V} |j\rangle \langle j| \boldsymbol{W} |0\rangle - \langle 0| \boldsymbol{W} |i\rangle \langle i| \boldsymbol{W} |j\rangle \langle j| \boldsymbol{V} |0\rangle \\
& +\langle 0| \boldsymbol{V} |i\rangle \langle i| \boldsymbol{W} |j\rangle \langle j| \boldsymbol{W} |0\rangle + \langle 0| \boldsymbol{W} |i\rangle \langle i| \boldsymbol{W} |j\rangle \langle j| \boldsymbol{V} |0\rangle \\
= & -\langle 0| \boldsymbol{W} |i\rangle \langle i| \boldsymbol{V} |j\rangle \langle j| \boldsymbol{W} |0\rangle + \langle 0| \boldsymbol{V} |i\rangle \langle i| \boldsymbol{W} |j\rangle \langle j| \boldsymbol{W} |0\rangle.
\end{aligned}
\tag{192}
$$

We can then combine terms (along with the terms linear in $\boldsymbol{W}$) to see that:

$$
\begin{aligned}
\hat{\ell}_{;i,j} = & \frac{2I\bar{o}}{rN^2} \operatorname{Re}\left\{ \langle 0| \boldsymbol{V} |i\rangle \langle i| \boldsymbol{W} |j\rangle \langle j| (\boldsymbol{W} + \boldsymbol{V}) |0\rangle \right\} \\
& -\frac{2I\bar{o}}{rN^2} \operatorname{Re}\left\{ \langle 0| \boldsymbol{W} |i\rangle \langle i| \boldsymbol{V} |j\rangle \langle j| (\boldsymbol{W} + \boldsymbol{V}) |0\rangle \right\}.
\end{aligned}
\tag{193}
$$

We can simplify the Hessian even further if we assume that $\frac{\sigma_o}{\overline{o}} = o(1)$ by dropping terms that are w.h.p. subleading in $\frac{\sigma_o}{\overline{o}}$. In particular, consider the following terms:

$$\frac{1}{rN} \operatorname{Re} \left\{ \langle 0| \boldsymbol{V} |i\rangle \langle i| \boldsymbol{W} |j\rangle \langle j| \boldsymbol{V} |0\rangle \right\}$$
$$\leq \frac{N-r}{N} \left| \frac{1}{\sqrt{N-r}} \langle 0| \boldsymbol{V} |i\rangle \right| \left| \frac{1}{r} \langle i| \boldsymbol{W} |j\rangle \right| \left| \frac{1}{\sqrt{N-r}} \langle j| \boldsymbol{V} |0\rangle \right|, \tag{194}$$

$$\frac{1}{rN} \operatorname{Re} \left\{ \langle 0| \boldsymbol{W} |i\rangle \langle i| \boldsymbol{V} |j\rangle \langle j| \boldsymbol{W} |0\rangle \right\}$$
$$\leq \frac{N-r}{N} \left| \frac{1}{\sqrt{r}} \langle 0| \boldsymbol{W} |i\rangle \right| \left| \frac{1}{N-r} \langle i| \boldsymbol{V} |j\rangle \right| \left| \frac{1}{\sqrt{r}} \langle j| \boldsymbol{W} |0\rangle \right|, \tag{195}$$

$$\frac{1}{rN} \operatorname{Re} \left\{ \langle 0| \boldsymbol{W} |i\rangle \langle i| \boldsymbol{V} |j\rangle \langle j| \boldsymbol{V} |0\rangle \right\}$$
$$\leq \frac{(N-r)^{\frac{3}{2}}}{\sqrt{r}N} \left| \frac{1}{\sqrt{r}} \langle 0| \boldsymbol{W} |i\rangle \right| \left| \frac{1}{N-r} \langle i| \boldsymbol{V} |j\rangle \right| \left| \frac{1}{\sqrt{N-r}} \langle j| \boldsymbol{V} |0\rangle \right|. \tag{196}$$

Recall that the diagonal elements of a $\beta$-Wishart matrix are $\chi^2$-distributed with $\beta r$ degrees of freedom (see Appendix A.4) and thus have exponential tails. In Lemma 43—proven in Appendix N—we prove that the off-diagonal elements also have exponential tails. In particular, all three of these terms are of the form $\kappa R_1 R_2 R_3$, where $0 < \kappa < 1$ is some overall prefactor and the $R_i$ are random variables obeying tail bounds of the form:

$$\mathbb{P}[R_i \geq 1 + t] \leq K \exp\left(-C \min\left(t, t^2\right)\right) \tag{197}$$

for some universal constants $C, K > 0$. We thus have by the union bound:

$$\mathbb{P}\left[\kappa R_1 R_2 R_3 \geq \left(\kappa^{\frac{1}{3}} + t\right)^3\right]$$
$$\leq \mathbb{P}\left[\kappa \max(R_1, R_2, R_3)^3 \geq \left(\kappa^{\frac{1}{3}} + t\right)^3\right]$$
$$\leq 3 \max\left(\mathbb{P}\left[\kappa^{\frac{1}{3}} R_1 \geq \kappa^{\frac{1}{3}} + t\right], \mathbb{P}\left[\kappa^{\frac{1}{3}} R_2 \geq \kappa^{\frac{1}{3}} + t\right], \mathbb{P}\left[\kappa^{\frac{1}{3}} R_3 \geq \kappa^{\frac{1}{3}} + t\right]\right) \tag{198}$$
$$\leq 3K \exp\left(-C \min\left(\kappa^{-\frac{1}{3}} t, \kappa^{-\frac{2}{3}} t^2\right)\right).$$

Taking into account the $p^2$ matrix elements via the union bound, these terms thus converge to $0$ in Ky Fan metric at a rate $\Omega\left(\kappa^{\frac{1}{3}} \log\left(\kappa^{-\frac{1}{3}} p\right)\right)$. We can similarly use Lemma 41 to justify replacing $\boldsymbol{W}$ with a low-rank perturbation $\tilde{\boldsymbol{W}}$, which differs only in the removal of the first column of its Bartlett factor $\boldsymbol{L}_W$. Altogether, we have that we can take up to an order

$$\alpha_5 = O\left(\frac{\mathcal{N}^{\frac{1}{3}}(N-r)^{\frac{1}{3}} \log\left(\frac{p^2 N^{\frac{2}{3}}}{\mathcal{N}^{\frac{1}{3}}(N-r)^{\frac{1}{3}}}\right)}{N^{\frac{2}{3}}}\right) \leq O\left(\frac{\mathcal{N}^{\frac{1}{3}} \sigma_o^{\frac{2}{3}} \log\left(\frac{p^2 N^{\frac{1}{3}} \overline{o}^{\frac{2}{3}}}{\mathcal{N}^{\frac{1}{3}} \sigma_o^{\frac{2}{3}}}\right)}{N^{\frac{1}{3}} \overline{o}^{\frac{2}{3}}}\right) \tag{199}$$

error in Ky Fan metric:

$$\hat{\ell}_{;i,j} = \frac{2I\overline{o}}{rN^2} \operatorname{Re}\left\{ \langle 0| \boldsymbol{V} |i\rangle \langle i| \tilde{\boldsymbol{W}} |j\rangle \langle j| \boldsymbol{W} |0\rangle \right\}. \tag{200}$$

At this point it is instructive to consider this joint distribution in terms of the marginal distributions of elements in the Bartlett decompositions of $\boldsymbol{W}$ and $\boldsymbol{V}$. Let $N_{W,i,j}$ be the i.i.d. standard normally distributed random variables over $\mathbb{F}$ in the off-diagonal elements of $\boldsymbol{L}_W$ (with the convention $i < j$), and similarly for $N_{V,i,j}$ and the Bartlett factor $\boldsymbol{L}_V$ of $\boldsymbol{V}$. Furthermore, let $\beta^{-\frac{1}{2}} \chi_{W,i}$ be the i.i.d. $\chi$-distributed random variables along the diagonal of $\boldsymbol{L}_W$, and similarly for $\beta^{-\frac{1}{2}} \chi_{V,i}$ and $\boldsymbol{L}_V$.

We first condition on $\hat{\ell} = z$. From Eq. (188) this is equivalent to conditioning on $\chi_{W,0} = \sqrt{\frac{\beta r}{I\overline{o}} z}$. Conditioned on this event, the first derivative is distributed as:

$$\hat{\ell}_{;i|z} = \frac{2I\beta\overline{o}}{rN} \sqrt{\frac{r}{I\overline{o}} z} \chi_{V,0} \operatorname{Re}\left\{ i N_{W,0,i}^* N_{V,0,i} \right\}. \tag{201}$$

Similarly, the second derivative conditioned on this event is distributed as:

$$\hat{\ell}_{;i,j|z} = \frac{2I\beta\overline{o}}{rN^2}\sqrt{\frac{r}{I\overline{o}}}z\chi_{V,0}\operatorname{Re}\left\{N_{V,0,i}^*\langle i|\,\tilde{\boldsymbol{W}}\,|j\rangle\,N_{W,0,j}\right\}. \tag{202}$$

Note that $\tilde{\boldsymbol{W}}$ is independent from $N_{W,0,i}$ exactly due to the removal of the first column of the Bartlett factor of $\boldsymbol{W}$.

We will next simplify further using the fact that $\chi_{V,0}$ concentrates around its mean, in particular using the $\chi$-distribution tail bound (for $N - r > 0$, some universal constant $C > 0$, and bounded $t$):

$$\mathbb{P}\left[\frac{1}{(N-r)^{\frac{1}{4}}}\left|\chi - \sqrt{\beta(N-r)}\right| \geq t\right] \leq 2\exp\left(-C\sqrt{N-r}\min\left(t, t^2\right)\right) \tag{203}$$

which follows from Lemma 42, proved in Appendix N. Just as we accounted for $\alpha_5$, up to an order

$$\alpha_6 = \mathrm{O}\left(\frac{\mathcal{N}^{\frac{1}{3}}r^{\frac{1}{6}}(N-r)^{\frac{1}{12}}\log\left(\frac{p^2N^{\frac{2}{3}}}{\mathcal{N}^{\frac{1}{3}}r^{\frac{1}{6}}(N-r)^{\frac{1}{12}}}\right)}{N^{\frac{2}{3}}}\right) \leq \mathrm{O}\left(\frac{\mathcal{N}^{\frac{1}{3}}\sigma_o^{\frac{1}{6}}\log\left(\frac{p^2N^{\frac{5}{12}}\overline{o}^{\frac{1}{6}}}{\mathcal{N}^{\frac{1}{3}}\sigma_o^{\frac{1}{6}}}\right)}{N^{\frac{5}{12}}\overline{o}^{\frac{1}{6}}}\right) \tag{204}$$

error in Ky Fan metric,

$$\hat{\ell}_{;i|z} = \frac{2I\beta\overline{o}}{N}\sqrt{\frac{\beta}{I\overline{o}}\left(\frac{N}{r}-1\right)}z\operatorname{Re}\left\{\mathrm{i}N_{W,0,i}^*N_{V,0,i}\right\}, \tag{205}$$

$$\hat{\ell}_{;i,j|z} = \frac{2I\beta\overline{o}}{N^2}\sqrt{\frac{\beta}{I\overline{o}}\left(\frac{N}{r}-1\right)}z\operatorname{Re}\left\{N_{V,0,i}^*\langle i|\,\tilde{\boldsymbol{W}}\,|j\rangle\,N_{W,0,j}\right\}. \tag{206}$$

Using Lemma 25 we can simplify this further as:

$$\hat{\ell}_{;i|z} = \frac{2I\beta\sigma_o}{N}\sqrt{\frac{\beta z}{I\overline{o}}}\operatorname{Re}\left\{\mathrm{i}N_{W,0,i}^*N_{V,0,i}\right\}, \tag{207}$$

$$\hat{\ell}_{;i,j|z} = \frac{2I\beta\sigma_o}{N^2}\sqrt{\frac{\beta z}{I\overline{o}}}\operatorname{Re}\left\{N_{V,0,i}^*\langle i|\,\tilde{\boldsymbol{W}}\,|j\rangle\,N_{W,0,j}\right\}. \tag{208}$$

We can simplify this even further by realizing that the distribution of $N_{V,0,i}$ is invariant under multiplication by an overall phase, i.e., it is circularly symmetric. We can thus absorb the argument of $N_{W,0,i}^*$ into $N_{V,0,i}$; we simultaneously conjugate $\tilde{\boldsymbol{W}}$ by unitary operators corresponding to this phase, so we need only consider the joint distribution of:

$$\hat{\ell}_{;i|z} = \frac{2I\beta\sigma_o}{N}\sqrt{\frac{z}{I\overline{o}}}\chi_i\operatorname{Re}\left\{\mathrm{i}N_{V,0,i}\right\}, \tag{209}$$

$$\hat{\ell}_{;i,j|z} = \frac{2I\beta\sigma_o}{N^2}\sqrt{\frac{z}{I\overline{o}}}\chi_j\langle i|\operatorname{Re}\left\{N_{V,0,i}^*\tilde{\boldsymbol{W}}\right\}|j\rangle, \tag{210}$$

where here the $\chi_i$ are i.i.d. $\chi$-distributed random variables with $\beta$ degrees of freedom given by:

$$\chi_i \equiv \sqrt{\beta}\,|N_{W,0,i}|. \tag{211}$$

Noting $\sqrt{\beta}\operatorname{Re}\{\mathrm{i}N_{V,0,i}\}$ is distributed as a standard normal random variable proves Theorem 17 for sectors with $\mathbb{F} = \mathbb{C}, \mathbb{H}$, with accumulated error in Lévy–Prokhorov distance:

$$\pi' = \pi_1 + \alpha_3 + 2\alpha_4 + \alpha_6 = \mathrm{O}\left(\frac{p\log\log\left(\mathcal{N}^{-t}\epsilon^{-1}\right)}{\log\left(\mathcal{N}^{-t}\epsilon^{-1}\right)} + \frac{p\log(t)}{\sqrt{t}} + \sqrt{\frac{\mathcal{N}\log(N)}{N^{1.001}}} + \frac{\mathcal{N}^{\frac{1}{3}}\log(N)}{N^{\frac{5}{12}}}\right). \tag{212}$$

We finally consider the second derivative conditioned on both $\hat{\ell} = z$ and all $\hat{\ell}_{;i|z} = 0$ (for $z > 0$). Note that the $\chi$ distribution density is zero at $0$. In particular, this conditioning is equivalent to conditioning on $\operatorname{Re}\{\mathrm{i}N_{V,0,i}\} = 0$. Furthermore, when $\mathbb{F} = \mathbb{H}$, recall that "i" could be any one

of i, j, k due to Eq. (140). In particular, this is conditioning on *all* nonreal components of $N_{V,0,i}$ equaling zero. Taking this all together yields:

$$\hat{\ell}_{;i,j|z,\mathbf{0}} = \frac{2I\beta\sigma_o}{N^2} \sqrt{\frac{z}{I\bar{o}}} \chi_j \operatorname{Re}\left\{N_{V,0,i}^*\right\} \langle i | \operatorname{Re}\left\{\tilde{\boldsymbol{W}}\right\} | j \rangle. \tag{213}$$

As the real parts of complex or symplectic Wishart matrices are real Wishart with a factor of $\beta$ more degrees of freedom and rescaled by a factor of $\beta^{-1}$, and similarly the real parts of standard complex or symplectic Gaussian random variables are standard real Gaussian random variables rescaled by a factor of $\beta^{-\frac{1}{2}}$, we can more conveniently rewrite this as

$$\hat{\ell}_{;i,j|z,\mathbf{0}} = \frac{2I\sigma_o}{N^2} \sqrt{\frac{z}{I\beta\bar{o}}} \hat{N}_i \chi_j \langle i | \hat{\boldsymbol{W}} | j \rangle, \tag{214}$$

where now $\hat{\boldsymbol{W}}$ is a real Wishart matrix with $\beta r$ degrees of freedom and the $\hat{N}_i$ are i.i.d. standard normal random variables. This proves Theorem 19 for sectors with $\mathbb{F} = \mathbb{C}, \mathbb{H}$, with accumulated error in Lévy–Prokhorov distance:

$$\pi'' = \pi_1 + \alpha_3 + 2\alpha_4 + \alpha_5 + \alpha_6 = \mathrm{O}\left(\frac{p^2 \log\log\left(\mathcal{N}^{-t}\epsilon^{-1}\right)}{\log\left(\mathcal{N}^{-t}\epsilon^{-1}\right)} + \frac{p^2 \log(t)}{\sqrt{t}} + \frac{\mathcal{N}^{\frac{1}{3}} \log(N)}{N^{\frac{1}{3}}}\right). \tag{215}$$

### D.3.2 $\mathbb{F} = \mathbb{R}$

We now consider when $\mathbb{F} = \mathbb{R}$, where we are interested in the joint distribution of:

$$\hat{\ell} = \frac{I\bar{o}}{r} \operatorname{Tr}\left(\boldsymbol{\rho}\boldsymbol{X}^\intercal \tilde{\boldsymbol{O}} \boldsymbol{X}\right), \tag{216}$$

$$\hat{\ell}_{;i} = \frac{I\bar{o}}{rN} \operatorname{Tr}\left(\boldsymbol{\rho}\boldsymbol{X}^\intercal \left[\boldsymbol{X}\left(|2i\rangle\langle 2i+1| - |2i+1\rangle\langle 2i|\right)\boldsymbol{X}^\intercal, \tilde{\boldsymbol{O}}\right] \boldsymbol{X}\right), \tag{217}$$

$$\hat{\ell}_{;i,j} = \frac{I\bar{o}}{rN^2} \operatorname{Tr}\Big(\boldsymbol{\rho}\boldsymbol{X}^\intercal \Big[\boldsymbol{X}\left(|2j\rangle\langle 2j+1| - |2j+1\rangle\langle 2j|\right)\boldsymbol{X}^\intercal,$$
$$\Big[\boldsymbol{X}\left(|2i\rangle\langle 2i+1| - |2i+1\rangle\langle 2i|\right)\boldsymbol{X}^\intercal, \tilde{\boldsymbol{O}}\Big]\Big]\boldsymbol{X}\Big), \tag{218}$$

where now $\boldsymbol{X}$ is a $N \times (2p+1)$ random matrix with i.i.d. standard Gaussian entries over $\mathbb{R}$. As the commutator of symmetric matrices is antisymmetric, and taking note of the identity:

$$\operatorname{Tr}\left(A\left[B, C\right]\right) = \operatorname{Tr}\left(\left[C, A\right] B\right), \tag{219}$$

we can rewrite this as:

$$\hat{\ell} = \frac{I\bar{o}}{r} \operatorname{Tr}\left(\boldsymbol{\rho}\boldsymbol{X}^\intercal \tilde{\boldsymbol{O}} \boldsymbol{X}\right), \tag{220}$$

$$\hat{\ell}_{;i} = \frac{2I\bar{o}}{rN} \operatorname{Tr}\left(\boldsymbol{\rho}\boldsymbol{X}^\intercal \left[\boldsymbol{X}|2i\rangle\langle 2i+1|\boldsymbol{X}^\intercal, \tilde{\boldsymbol{O}}\right] \boldsymbol{X}\right), \tag{221}$$

$$\hat{\ell}_{;i,j} = \frac{2I\bar{o}}{rN^2} \operatorname{Tr}\left(\boldsymbol{\rho}\boldsymbol{X}^\intercal \left[\boldsymbol{X}|2j\rangle\langle 2j+1|\boldsymbol{X}^\intercal, \left[\boldsymbol{X}\left(|2i\rangle\langle 2i+1| - |2i+1\rangle\langle 2i|\right)\boldsymbol{X}^\intercal, \tilde{\boldsymbol{O}}\right]\right] \boldsymbol{X}\right). \tag{222}$$

Recall that

$$\tilde{\boldsymbol{O}} = \sum_{\mu=0}^{r-1} |\mu\rangle\langle\mu|. \tag{223}$$

Let $\boldsymbol{C}$ be the complement of $\tilde{\boldsymbol{O}}$, i.e.,

$$\boldsymbol{C} = \sum_{\mu=r}^{N-1} |\mu\rangle\langle\mu|. \tag{224}$$

In particular, $\boldsymbol{C}$ and $\tilde{\boldsymbol{O}}$ sum to the identity:

$$\boldsymbol{C} + \tilde{\boldsymbol{O}} = \boldsymbol{I}_N. \tag{225}$$

Furthermore,

$$\boldsymbol{W} \equiv \boldsymbol{X}^{\intercal} \tilde{\boldsymbol{O}} \boldsymbol{X}, \tag{226}$$

$$\boldsymbol{V} \equiv \boldsymbol{X}^{\intercal} \boldsymbol{C} \boldsymbol{X}, \tag{227}$$

are each Wishart-distributed matrices; the former with $r$ degrees of freedom, and the latter with $N - r$. The independence of $\boldsymbol{W}$ and $\boldsymbol{V}$ can be checked from the associated Bartlett decompositions. We thus have that the distribution we are interested in is the joint distribution of:

$$\hat{\ell} = \frac{I\bar{o}}{r} \langle 0| \, \boldsymbol{W} \, |0\rangle , \tag{228}$$

$$\begin{aligned}
\hat{\ell}_{;i} &= \frac{2I\bar{o}}{rN} \langle 0| \left( \boldsymbol{X}^{\intercal} \boldsymbol{X} \, |2i\rangle \langle 2i+1| \, \boldsymbol{W} - \boldsymbol{W} \, |2i\rangle \langle 2i+1| \, \boldsymbol{X}^{\intercal} \boldsymbol{X} \right) |0\rangle \\
&= -\frac{2I\bar{o}}{rN} \langle 0| \, \boldsymbol{W} \left( |2i\rangle \langle 2i+1| - |2i+1\rangle \langle 2i| \right) \boldsymbol{V} \, |0\rangle ,
\end{aligned} \tag{229}$$

and:

$$\begin{aligned}
\hat{\ell}_{;i,j} =& \frac{2I\bar{o}}{rN^2} \langle 0| \, \boldsymbol{X}^{\intercal} \left[ \boldsymbol{X} \, |2j\rangle \langle 2j+1| \, \boldsymbol{X}^{\intercal}, \left[ \boldsymbol{X} \left( |2i\rangle \langle 2i+1| - |2i+1\rangle \langle 2i| \right) \boldsymbol{X}^{\intercal}, \boldsymbol{O} \right] \right] \boldsymbol{X} \, |0\rangle \\
=& \frac{2I\bar{o}}{rN^2} \langle 0| \, \boldsymbol{X}^{\intercal} \boldsymbol{X} \, |2j\rangle \langle 2j+1| \, \boldsymbol{X}^{\intercal} \boldsymbol{X} \left( |2i\rangle \langle 2i+1| - |2i+1\rangle \langle 2i| \right) \boldsymbol{X}^{\intercal} \boldsymbol{O} \boldsymbol{X} \, |0\rangle \\
&+ \frac{2I\bar{o}}{rN^2} \langle 0| \, \boldsymbol{X}^{\intercal} \boldsymbol{O} \boldsymbol{X} \left( |2i\rangle \langle 2i+1| - |2i+1\rangle \langle 2i| \right) \boldsymbol{X}^{\intercal} \boldsymbol{X} \, |2j\rangle \langle 2j+1| \, \boldsymbol{X}^{\intercal} \boldsymbol{X} \, |0\rangle \\
&- \frac{2I\bar{o}}{rN^2} \langle 0| \, \boldsymbol{X}^{\intercal} \boldsymbol{X} \left( |2i\rangle \langle 2i+1| - |2i+1\rangle \langle 2i| \right) \boldsymbol{X}^{\intercal} \boldsymbol{O} \boldsymbol{X} \, |2j\rangle \langle 2j+1| \, \boldsymbol{X}^{\intercal} \boldsymbol{X} \, |0\rangle \\
&- \frac{2I\bar{o}}{rN^2} \langle 0| \, \boldsymbol{X}^{\intercal} \boldsymbol{X} \, |2j\rangle \langle 2j+1| \, \boldsymbol{X}^{\intercal} \boldsymbol{O} \boldsymbol{X} \left( |2i\rangle \langle 2i+1| - |2i+1\rangle \langle 2i| \right) \boldsymbol{X}^{\intercal} \boldsymbol{X} \, |0\rangle \\
=& \frac{2I\bar{o}}{rN^2} \langle 0| \left( \boldsymbol{W} + \boldsymbol{V} \right) |2j\rangle \langle 2j+1| \left( \boldsymbol{W} + \boldsymbol{V} \right) \left( |2i\rangle \langle 2i+1| - |2i+1\rangle \langle 2i| \right) \boldsymbol{W} \, |0\rangle \\
&+ \frac{2I\bar{o}}{rN^2} \langle 0| \, \boldsymbol{W} \left( |2i\rangle \langle 2i+1| - |2i+1\rangle \langle 2i| \right) \left( \boldsymbol{W} + \boldsymbol{V} \right) |2j\rangle \langle 2j+1| \left( \boldsymbol{W} + \boldsymbol{V} \right) |0\rangle \\
&- \frac{2I\bar{o}}{rN^2} \langle 0| \left( \boldsymbol{W} + \boldsymbol{V} \right) \left( |2i\rangle \langle 2i+1| - |2i+1\rangle \langle 2i| \right) \boldsymbol{W} \, |2j\rangle \langle 2j+1| \left( \boldsymbol{W} + \boldsymbol{V} \right) |0\rangle \\
&- \frac{2I\bar{o}}{rN^2} \langle 0| \left( \boldsymbol{W} + \boldsymbol{V} \right) |2j\rangle \langle 2j+1| \, \boldsymbol{W} \left( |2i\rangle \langle 2i+1| - |2i+1\rangle \langle 2i| \right) \left( \boldsymbol{W} + \boldsymbol{V} \right) |0\rangle .
\end{aligned} \tag{230}$$

We can simplify the expression for the second derivative by grouping terms by their order in $\boldsymbol{W}$. Note first that all terms cubic in matrix elements of $\boldsymbol{W}$ sum to zero. For the quadratic terms, note that for the outer two terms:

$$\begin{aligned}
& + \langle 0| \, \boldsymbol{V} \, |2j\rangle \langle 2j+1| \, \boldsymbol{W} \left( |2i\rangle \langle 2i+1| - |2i+1\rangle \langle 2i| \right) \boldsymbol{W} \, |0\rangle \\
& + \langle 0| \, \boldsymbol{W} \, |2j\rangle \langle 2j+1| \, \boldsymbol{V} \left( |2i\rangle \langle 2i+1| - |2i+1\rangle \langle 2i| \right) \boldsymbol{W} \, |0\rangle \\
& - \langle 0| \, \boldsymbol{V} \, |2j\rangle \langle 2j+1| \, \boldsymbol{W} \left( |2i\rangle \langle 2i+1| - |2i+1\rangle \langle 2i| \right) \boldsymbol{W} \, |0\rangle \\
& - \langle 0| \, \boldsymbol{W} \, |2j\rangle \langle 2j+1| \, \boldsymbol{W} \left( |2i\rangle \langle 2i+1| - |2i+1\rangle \langle 2i| \right) \boldsymbol{V} \, |0\rangle \\
=& + \langle 0| \, \boldsymbol{W} \, |2j\rangle \langle 2j+1| \, \boldsymbol{V} \left( |2i\rangle \langle 2i+1| - |2i+1\rangle \langle 2i| \right) \boldsymbol{W} \, |0\rangle \\
& - \langle 0| \, \boldsymbol{W} \, |2j\rangle \langle 2j+1| \, \boldsymbol{W} \left( |2i\rangle \langle 2i+1| - |2i+1\rangle \langle 2i| \right) \boldsymbol{V} \, |0\rangle ,
\end{aligned} \tag{231}$$

and for the inner two terms:

$$\begin{aligned}
& + \langle 0| \, \boldsymbol{W} \left( |2i\rangle \langle 2i+1| - |2i+1\rangle \langle 2i| \right) \boldsymbol{V} \, |2j\rangle \langle 2j+1| \, \boldsymbol{W} \, |0\rangle \\
& + \langle 0| \, \boldsymbol{W} \left( |2i\rangle \langle 2i+1| - |2i+1\rangle \langle 2i| \right) \boldsymbol{W} \, |2j\rangle \langle 2j+1| \, \boldsymbol{V} \, |0\rangle \\
& - \langle 0| \, \boldsymbol{V} \left( |2i\rangle \langle 2i+1| - |2i+1\rangle \langle 2i| \right) \boldsymbol{W} \, |2j\rangle \langle 2j+1| \, \boldsymbol{W} \, |0\rangle \\
& - \langle 0| \, \boldsymbol{W} \left( |2i\rangle \langle 2i+1| - |2i+1\rangle \langle 2i| \right) \boldsymbol{W} \, |2j\rangle \langle 2j+1| \, \boldsymbol{V} \, |0\rangle \\
=& + \langle 0| \, \boldsymbol{W} \left( |2i\rangle \langle 2i+1| - |2i+1\rangle \langle 2i| \right) \boldsymbol{V} \, |2j\rangle \langle 2j+1| \, \boldsymbol{W} \, |0\rangle \\
& - \langle 0| \, \boldsymbol{V} \left( |2i\rangle \langle 2i+1| - |2i+1\rangle \langle 2i| \right) \boldsymbol{W} \, |2j\rangle \langle 2j+1| \, \boldsymbol{W} \, |0\rangle .
\end{aligned} \tag{232}$$

We can, in turn, combine these terms to obtain:

$$
\begin{aligned}
&+ \langle 0| \, \boldsymbol{W} \, |2j\rangle \langle 2j+1| \, \boldsymbol{V} \, (|2i\rangle \langle 2i+1| - |2i+1\rangle \langle 2i|) \, \boldsymbol{W} \, |0\rangle \\
&- \langle 0| \, \boldsymbol{W} \, |2j\rangle \langle 2j+1| \, \boldsymbol{W} \, (|2i\rangle \langle 2i+1| - |2i+1\rangle \langle 2i|) \, \boldsymbol{V} \, |0\rangle \\
&+ \langle 0| \, \boldsymbol{W} \, (|2i\rangle \langle 2i+1| - |2i+1\rangle \langle 2i|) \, \boldsymbol{V} \, |2j\rangle \langle 2j+1| \, \boldsymbol{W} \, |0\rangle \\
&- \langle 0| \, \boldsymbol{V} \, (|2i\rangle \langle 2i+1| - |2i+1\rangle \langle 2i|) \, \boldsymbol{W} \, |2j\rangle \langle 2j+1| \, \boldsymbol{W} \, |0\rangle \\
={}&+ \langle 0| \, \boldsymbol{W} \, |2j\rangle \langle 2j+1| \, \boldsymbol{V} \, (|2i\rangle \langle 2i+1| - |2i+1\rangle \langle 2i|) \, \boldsymbol{W} \, |0\rangle \\
&- \langle 0| \, \boldsymbol{W} \, |2j\rangle \langle 2j+1| \, \boldsymbol{W} \, (|2i\rangle \langle 2i+1| - |2i+1\rangle \langle 2i|) \, \boldsymbol{V} \, |0\rangle \\
&- \langle 0| \, \boldsymbol{W} \, |2j+1\rangle \langle 2j| \, \boldsymbol{V} \, (|2i\rangle \langle 2i+1| - |2i+1\rangle \langle 2i|) \, \boldsymbol{W} \, |0\rangle \\
&+ \langle 0| \, \boldsymbol{W} \, |2j+1\rangle \langle 2j| \, \boldsymbol{W} \, (|2i\rangle \langle 2i+1| - |2i+1\rangle \langle 2i|) \, \boldsymbol{V} \, |0\rangle \\
={}&+ \langle 0| \, \boldsymbol{W} \, (|2j\rangle \langle 2j+1| - |2j+1\rangle \langle 2j|) \, \boldsymbol{V} \, (|2i\rangle \langle 2i+1| - |2i+1\rangle \langle 2i|) \, \boldsymbol{W} \, |0\rangle \\
&- \langle 0| \, \boldsymbol{W} \, (|2j\rangle \langle 2j+1| - |2j+1\rangle \langle 2j|) \, \boldsymbol{W} \, (|2i\rangle \langle 2i+1| - |2i+1\rangle \langle 2i|) \, \boldsymbol{V} \, |0\rangle \,.
\end{aligned}
\tag{233}
$$

We can then combine terms with those linear in $\boldsymbol{W}$ to see that:

$$
\begin{aligned}
\hat{\ell}_{;i,j} ={}& \frac{2I\overline{o}}{rN^2} \langle 0| \, \boldsymbol{W} \, (|2i\rangle \langle 2i+1| - |2i+1\rangle \langle 2i|) \, \boldsymbol{V} \, (|2j\rangle \langle 2j+1| - |2j+1\rangle \langle 2j|) \, (\boldsymbol{W} + \boldsymbol{V}) \, |0\rangle \\
&- \frac{2I\overline{o}}{rN^2} \langle 0| \, \boldsymbol{V} \, (|2i\rangle \langle 2i+1| - |2i+1\rangle \langle 2i|) \, \boldsymbol{W} \, (|2j\rangle \langle 2j+1| - |2j+1\rangle \langle 2j|) \, (\boldsymbol{W} + \boldsymbol{V}) \, |0\rangle \,.
\end{aligned}
\tag{234}
$$

We now use two lemmas—proved in Appendix N—that will simplify the form of our answer by allowing us to justify the dropping of terms that are subleading w.h.p. In particular, we use Lemma 43 followed by Lemma 41 just as we did in Eqs. (194), (195), and (196). By the same logic, we have that we can take up to an order

$$
\alpha_5 = \mathrm{O}\left( \frac{\mathcal{N}^{\frac{1}{3}} (N-r)^{\frac{1}{3}} \log\left( \frac{p^2 N^{\frac{2}{3}}}{\mathcal{N}^{\frac{1}{3}} (N-r)^{\frac{1}{3}}} \right)}{N^{\frac{2}{3}}} \right) \leq \mathrm{O}\left( \frac{\mathcal{N}^{\frac{1}{3}} \sigma_o^{\frac{2}{3}} \log\left( \frac{p^2 N^{\frac{1}{3}} \overline{o}^{\frac{2}{3}}}{\mathcal{N}^{\frac{1}{3}} \sigma_o^{\frac{2}{3}}} \right)}{N^{\frac{1}{3}} \overline{o}^{\frac{2}{3}}} \right)
\tag{235}
$$

error in Ky Fan metric:

$$
\hat{\ell}_{;i,j} = -\frac{2I\overline{o}}{rN^2} \langle 0| \, \boldsymbol{V} \, (|2i\rangle \langle 2i+1| - |2i+1\rangle \langle 2i|) \, \tilde{\boldsymbol{W}} \, (|2j\rangle \langle 2j+1| - |2j+1\rangle \langle 2j|) \, \boldsymbol{W} \, |0\rangle \,,
\tag{236}
$$

where $\tilde{\boldsymbol{W}}$ is as $\boldsymbol{W}$, but with the first column of its Bartlett factor $\boldsymbol{L}_W$ removed.

At this point it is instructive to consider this joint distribution in terms of the marginal distributions of elements in the Bartlett decompositions of $\boldsymbol{W}$ and $\boldsymbol{V}$. Let $N_{W,i,j}$ be the i.i.d. standard normally distributed random variables over $\mathbb{R}$ in the off-diagonal elements of $\boldsymbol{L}_W$ (with the convention $i < j$), and similarly for $N_{V,i,j}$ and the Bartlett factor $\boldsymbol{L}_V$ of $\boldsymbol{V}$. Furthermore, let $\chi_{W,i}$ be the i.i.d. $\chi$-distributed random variables along the diagonal of $\boldsymbol{L}_W$, and similarly for $\chi_{V,i}$ and $\boldsymbol{L}_V$.

We first condition on $\hat{\ell} = z$. From Eq. (188) this is equivalent to conditioning on $\chi_{W,0} = \sqrt{\frac{r}{I\overline{o}}}z$. Conditioned on this event, the first derivative is distributed as:

$$
\hat{\ell}_{;i|z} = \frac{2I\overline{o}}{rN} \sqrt{\frac{r}{I\overline{o}}} z \chi_{V,0} \left( N_{W,0,2i+1} N_{V,0,2i} - N_{W,0,2i} N_{V,0,2i+1} \right).
\tag{237}
$$

Similarly, the second derivative conditioned on this event is distributed as:

$$
\begin{aligned}
\hat{\ell}_{;i,j|z} ={}& \frac{2I\overline{o}}{rN^2} \sqrt{\frac{r}{I\overline{o}}} z \chi_{V,0} \\
&\times \left( N_{V,0,2i+1} \langle 2i| - N_{V,0,2i} \langle 2i+1| \right) \tilde{\boldsymbol{W}} \left( |2j\rangle N_{W,0,2j+1} - |2j+1\rangle N_{W,0,2j} \right).
\end{aligned}
\tag{238}
$$

Note that $\tilde{\boldsymbol{W}}$ is independent from $N_{W,0,i}$ exactly due to the removal of the first column of the Bartlett factor of $\boldsymbol{W}$.

We will next simplify further using the fact that $\chi_{V,0}$ concentrates around its mean, in particular using the $\chi$-distribution tail bound (for $N - r > 0$, some universal constant $C > 0$, and bounded $t$):

$$
\mathbb{P}\left[ \frac{1}{(N-r)^{\frac{1}{4}}} \left| \chi - \sqrt{\beta(N-r)} \right| \geq t \right] \leq 2 \exp\left( -C\sqrt{N-r} \min\left( t, t^2 \right) \right)
\tag{239}
$$

which follows from Lemma 42, proved in Appendix N. Just as we accounted for $\alpha_5$, up to an order

$$\alpha_6 = O\left(\frac{\mathcal{N}^{\frac{1}{3}}r^{\frac{1}{6}}(N-r)^{\frac{1}{12}}\log\left(\frac{p^2 N^{\frac{2}{3}}}{\mathcal{N}^{\frac{1}{3}}r^{\frac{1}{6}}(N-r)^{\frac{1}{12}}}\right)}{N^{\frac{2}{3}}}\right) \leq O\left(\frac{\mathcal{N}^{\frac{1}{3}}\sigma_o^{\frac{1}{6}}\log\left(\frac{p^2 N^{\frac{5}{12}}\bar{o}^{\frac{1}{6}}}{\mathcal{N}^{\frac{1}{3}}\sigma_o^{\frac{1}{6}}}\right)}{N^{\frac{5}{12}}\bar{o}^{\frac{1}{6}}}\right) \tag{240}$$

error in Ky Fan metric,

$$\hat{\ell}_{;i|z} = \frac{2I\bar{o}}{N}\sqrt{\left(\frac{N}{r}-1\right)\frac{z}{I\bar{o}}}\left(N_{W,0,2i+1}N_{V,0,2i} - N_{W,0,2i}N_{V,0,2i+1}\right), \tag{241}$$

$$\hat{\ell}_{;i,j|z} = \frac{2I\bar{o}}{N^2}\sqrt{\left(\frac{N}{r}-1\right)\frac{z}{I\bar{o}}} \tag{242}$$
$$\times \left(N_{V,0,2i+1}\langle 2i| - N_{V,0,2i}\langle 2i+1|\right)\tilde{W}\left(|2j\rangle N_{W,0,2j+1} - |2j+1\rangle N_{W,0,2j}\right).$$

Using Lemma 25 we can simplify this further as:

$$\hat{\ell}_{;i|z} = \frac{2I\sigma_o}{N}\sqrt{\frac{z}{I\bar{o}}}\left(N_{W,0,2i+1}N_{V,0,2i} - N_{W,0,2i}N_{V,0,2i+1}\right), \tag{243}$$

$$\hat{\ell}_{;i,j|z} = \frac{2I\sigma_o}{N^2}\sqrt{\frac{z}{I\bar{o}}}\left(N_{V,0,2i+1}\langle 2i| - N_{V,0,2i}\langle 2i+1|\right)\tilde{W}\left(|2j\rangle N_{W,0,2j+1} - |2j+1\rangle N_{W,0,2j}\right). \tag{244}$$

We can simplify this even further by realizing that the joint distribution of $(N_{V,0,2i}, N_{V,0,2i+1})$ is invariant under orthogonal transformations, i.e., it is circularly symmetric. We can thus simultaneously transform $(N_{V,0,2i}, N_{V,0,2i+1})$ and conjugate $\tilde{W}$ by orthogonal operators such that we need only consider the joint distribution of:

$$\hat{\ell}_{;i|z} = \frac{2I\sigma_o}{N}\sqrt{\frac{z}{I\bar{o}}}\chi_i N_{V,0,2i+1}, \tag{245}$$

$$\hat{\ell}_{;i,j|z} = \frac{2I\sigma_o}{N^2}\sqrt{\frac{z}{I\bar{o}}}\chi_j\left(N_{V,0,2i}\langle 2i| - N_{V,0,2i+1}\langle 2i+1|\right)\tilde{W}|2j\rangle, \tag{246}$$

where here the $\chi_i$ are i.i.d. $\chi$-distributed random variables with 2 degrees of freedom given by:

$$\chi_i \equiv \sqrt{N_{W,0,2i}^2 + N_{W,0,2i+1}^2}. \tag{247}$$

This proves Theorem 17 for sectors with $\mathbb{F} = \mathbb{R}$, with accumulated error in Lévy–Prokhorov distance:

$$\pi' = \pi_1 + \alpha_3 + 2\alpha_4 + \alpha_6 = O\left(\frac{p\log\log\left(\mathcal{N}^{-t}\epsilon^{-1}\right)}{\log\left(\mathcal{N}^{-t}\epsilon^{-1}\right)} + \frac{p\log(t)}{\sqrt{t}} + \sqrt{\frac{\mathcal{N}\log(N)}{N^{1.001}}} + \frac{\mathcal{N}^{\frac{1}{3}}\log(N)}{N^{\frac{5}{12}}}\right). \tag{248}$$

We finally consider the second derivative conditioned on both $\hat{\ell} = z$ and all $\hat{\ell}_{;i|z} = 0$ (for $z > 0$). Note that the $\chi$ distribution density is zero at 0. In particular, this conditioning is equivalent to conditioning on $N_{V,0,2i+1} = 0$, yielding:

$$\hat{\ell}_{;i,j|z,\mathbf{0}} = \frac{2I\sigma_o}{N^2}\sqrt{\frac{z}{I\bar{o}}}N_{V,0,2i}\chi_j\langle 2i|\tilde{W}|2j\rangle. \tag{249}$$

As submatrices of Wishart matrices are also Wishart, we can more conveniently rewrite this by taking $\tilde{W}$ to be $p \times p$:

$$\hat{\ell}_{;i,j|z,\mathbf{0}} = \frac{2I\sigma_o}{N^2}\sqrt{\frac{z}{I\bar{o}}}\hat{N}_i\chi_j\langle i|\tilde{W}|j\rangle, \tag{250}$$

where the $\hat{N}_i$ are i.i.d. standard normal random variables. This proves Theorem 19 for sectors with $\mathbb{F} = \mathbb{R}$, with accumulated error in Lévy–Prokhorov distance:

$$\pi'' = \pi_1 + \alpha_3 + 2\alpha_4 + \alpha_5 + \alpha_6 = O\left(\frac{p^2\log\log\left(\mathcal{N}^{-t}\epsilon^{-1}\right)}{\log\left(\mathcal{N}^{-t}\epsilon^{-1}\right)} + \frac{p^2\log(t)}{\sqrt{t}} + \frac{\mathcal{N}^{\frac{1}{3}}\log(N)}{N^{\frac{1}{3}}}\right). \tag{251}$$

# E  QUANTUM ALGORITHM FOR DETERMINING ASYMPTOTIC TRAINABILITY

We elaborate here on a claim made in Sec. 6 that one can efficiently estimate $\mathrm{Tr}\left(\left(\boldsymbol{O}^{\alpha}\right)^2\right)$ and $\mathrm{Tr}\left(\left(\boldsymbol{\rho}^{\alpha}\right)^2\right)$ on a quantum computer and thus, by Definition 7, determine whether a given quantum neural network architecture is asymptotically trainable or not. We assume as input the Jordan algebra $\mathcal{A} \cong \bigoplus_{\alpha} \mathcal{A}_{\alpha}$ associated with the model, given as orthonormal bases $\{\boldsymbol{B}_{\alpha,i}\}_i$ spanning each $\mathcal{A}_{\alpha}$. Note that the corresponding traces $\mathrm{Tr}_{\alpha}\left(\left(\boldsymbol{O}^{\alpha}\right)^2\right)$ and $\mathrm{Tr}_{\alpha}\left(\left(\boldsymbol{\rho}^{\alpha}\right)^2\right)$ in the defining representation of $\mathcal{A}_{\alpha}$ can then also be evaluated using Eq. (11):

$$\mathrm{Tr}\left(\cdot\right) = I_{\alpha}\,\mathrm{Tr}_{\alpha}\left(\cdot\right); \tag{252}$$

in particular, $I_{\alpha}$ can be calculated from the ratio of traces of any $\boldsymbol{B}_{\alpha,i}$ in the defining representation of $\mathcal{H}^N\left(\mathbb{C}\right) \supseteq \mathcal{A}_{\alpha}$ and that of $\mathcal{A}_{\alpha}$.

We begin with $\mathrm{Tr}\left(\left(\boldsymbol{\rho}^{\alpha}\right)^2\right)$. It is immediate from the orthonormality of the $\boldsymbol{B}_{\alpha,i}$ that:

$$\mathrm{Tr}\left(\left(\boldsymbol{\rho}^{\alpha}\right)^2\right) = \sum_i \mathrm{Tr}\left(\boldsymbol{B}_{\alpha,i}\boldsymbol{\rho}\right)^2; \tag{253}$$

in particular, $\mathrm{Tr}\left(\left(\boldsymbol{\rho}^{\alpha}\right)^2\right)$ is calculable via, for instance, phase estimation (Nielsen & Chuang, 2010a) of the $\boldsymbol{B}_{\alpha,i}$.

This leaves only $\mathrm{Tr}\left(\left(\boldsymbol{O}^{\alpha}\right)^2\right)$. If $\boldsymbol{O}$ were an efficiently preparable, valid quantum state, one could immediately calculate $\mathrm{Tr}\left(\left(\boldsymbol{O}^{\alpha}\right)^2\right)$ in the same way as $\mathrm{Tr}\left(\left(\boldsymbol{\rho}^{\alpha}\right)^2\right)$. In the more general case where, for instance, $\boldsymbol{O}$ is given via its Pauli decomposition:

$$\boldsymbol{O} = \sum_{i=1}^S o_i \boldsymbol{P}_i, \tag{254}$$

one can proceed similarly via a block encoding using the linear combination of unitaries (LCU) subroutine (Childs & Wiebe, 2012).

To outline this procedure, we assume $\boldsymbol{O}$ is an observable on $n$ qubits. Let:

$$\boldsymbol{f}\left(\boldsymbol{O}\right) \equiv \sum_{i=1}^S \frac{o_i}{2} \boldsymbol{P}_i \otimes \left(\boldsymbol{I}_2 + \boldsymbol{Z}\right), \tag{255}$$

where $\boldsymbol{I}_N$ is the $N \times N$ identity matrix and $\boldsymbol{Z}$ the $2 \times 2$ single-qubit Pauli $Z$ matrix, and consider the $(n+1)$-qubit quantum state:

$$\boldsymbol{\omega}_0 \equiv \frac{\boldsymbol{I}_{2^n}}{2^n} \otimes |0\rangle\langle 0|. \tag{256}$$

We have by construction that:

$$\boldsymbol{O}^2 \otimes |0\rangle\langle 0| = \boldsymbol{f}\left(\boldsymbol{O}\right)\boldsymbol{\omega}_0\boldsymbol{f}\left(\boldsymbol{O}\right). \tag{257}$$

Thus,

$$\begin{aligned}
\mathrm{Tr}\left(\left(\boldsymbol{O}^{\alpha}\right)^2\right) &= \sum_i \mathrm{Tr}\left(\boldsymbol{B}_{\alpha,i}\boldsymbol{O}^2\boldsymbol{B}_{\alpha,i}\right) \\
&= \sum_i \mathrm{Tr}\left(\left(\boldsymbol{B}_{\alpha,i}\right)^2 \boldsymbol{f}\left(\boldsymbol{O}\right)\boldsymbol{\omega}_0\boldsymbol{f}\left(\boldsymbol{O}\right)\right).
\end{aligned} \tag{258}$$

Of course, $\boldsymbol{f}\left(\boldsymbol{O}\right)$ is not necessarily unitary, so naive application of $\boldsymbol{f}\left(\boldsymbol{O}\right)$ is impossible on a quantum computer. However, one can proceed using the standard LCU subroutine (Childs & Wiebe, 2012). Assuming $\boldsymbol{O}$ is normalized such that its root mean square eigenvalue is $\Theta\left(1\right)$,

$$\mathrm{Tr}\left(\boldsymbol{f}\left(\boldsymbol{O}\right)\boldsymbol{\omega}_0\boldsymbol{f}\left(\boldsymbol{O}\right)\right) = 2^{-n}\,\mathrm{Tr}\left(\boldsymbol{O}^2\right) = \Theta\left(1\right). \tag{259}$$

Following the protocol described by Theorem 1 of Chakraborty (2024), then, each $\mathrm{Tr}\left(\left(\boldsymbol{B}_{\alpha,i}\right)^2 \boldsymbol{f}\left(\boldsymbol{O}\right)\boldsymbol{\omega}_0\boldsymbol{f}\left(\boldsymbol{O}\right)\right)$ can be estimated to an additive error $\epsilon$ using one ancilla qubit and $\mathrm{O}\left(\frac{S^4}{\epsilon^2}\right)$ samples through an LCU block encoding.

## F    Open Questions

As our results are formal, there is room for violation of their conclusions in the settings where their assumptions do not hold. One example of this is the local minima distribution of quantum neural networks discussed in Appendix C.3. The results given there assume that the eigenvalue spectrum of the objective observable is concentrated as is typical of low-weight fermionic (Feng et al., 2019) and local spin Hamiltonians (Erdős & Schröder, 2014). However, the behavior of the local minima density differs when the spectrum is not concentrated, as is typical of observables drawn from the Gaussian unitary ensemble (GUE) and nonlocal spin systems. Observables from this latter class are known to be "quantumly easy" in the sense that phase estimation performed on the maximally mixed state efficiently prepares the ground state (Chen et al., 2024). Based on this intuition these nonlocal systems may also yield local minima distributions for quantum neural networks more amenable to variational optimization even at shallow depth. If this is true, this may give a natural class of optimization problems which are easy to solve given access to a quantum computer.

Furthermore, we nowhere discuss the impact of noise: we assume all operations implemented by the quantum neural network are unitary, not general quantum channels. It is already known that noise can affect the presence of barren plateaus (Wang et al., 2021) as well as local minima (Li & Hernandez, 2024; Mele et al., 2024) in variational loss landscapes. Heuristically the former can be understood as an additional channel mixing the variational ansatz over Hilbert space, and the latter as noise effectively limiting the number of parameters influencing the variational ansatz and thus always keeping the network in the underparameterized regime. We hope in the future to make this heuristic understanding rigorous.

## G    Reduction of Spin Factor Sectors to Real Symmetric Sectors

We elaborate here on the claim made in Appendix B.1 that the $\alpha$ where $\mathcal{A}_\alpha \cong \mathcal{L}_{N_\alpha}$—i.e., the spin factor sectors—are conceptually equivalent to the case $\mathcal{A}_\alpha \cong \mathcal{H}^{N_\alpha - 1}(\mathbb{R})$. To see this, consider a parameterization of $T^\alpha(\boldsymbol{\theta})$ in the spin factor case:

$$T(\boldsymbol{\theta}) = g_0^{\alpha\mathsf{T}} \prod_{i=1}^{p} g_i^\alpha \exp\left(\theta_i A_i^\alpha\right) g_i^{\alpha\mathsf{T}}, \tag{260}$$

where $g_i^\alpha \in \mathrm{SO}(N-1)$ and $A_i^\alpha \in \mathfrak{so}(N-1)$. As stated in Appendix B.1, $g_i^\alpha$ acts trivially on the first component in the defining representation. Because of this (assuming the loss is shifted by a constant to always be nonnegative), the associated $\ell^\alpha(\boldsymbol{\theta}; \boldsymbol{\rho})$ is just equal to the square root of an instance of the $\mathcal{H}^{N_\alpha - 1}(\mathbb{R})$ case with initial state:

$$\tilde{\boldsymbol{\rho}}^\alpha \equiv \boldsymbol{\rho}^{\alpha\mathsf{T}} \otimes \boldsymbol{\rho}^\alpha \tag{261}$$

and objective observable:

$$\tilde{\boldsymbol{O}}^\alpha \equiv \boldsymbol{O}^\alpha \otimes \boldsymbol{O}^{\alpha\mathsf{T}}. \tag{262}$$

## H    Reduction of Previously Studied Ansatz Classes to Jordan Algebra-Supported Ansatzes

We here show that Jordan algebra-supported ansatzes (JASAs) are reduced to by the algebraically defined classes of variational ansatzes previously introduced in the literature, with a primary focus on the Lie algebra-supported ansatzes (LASAs) (Fontana et al., 2024). A reduction in the other direction—i.e., given a Jordan algebra-defined ansatz, define an equivalent LASA—is not possible. As a simple example, consider the case when the objective observable $\boldsymbol{O}$ is a real symmetric matrix and the variational ansatz is PT symmetric (i.e., real) but has no other symmetries. This fits our framework with Jordan algebra $\mathcal{A} \cong \mathcal{H}^N(\mathbb{R})$ and automorphism group given by the action of conjugation by $\mathrm{SO}(N)$. However, $\mathfrak{so}(N)$ in the defining representation consists of antisymmetric matrices, so this is not a LASA. We also give a reduction from the variational matchgate circuits of Diaz et al. (2023) to our Jordan algebraic framework, and will later demonstrate that such a reduction to LASAs is not possible.

We first discuss how JASAs are reduced to by LASAs. The loss function of a LASA is of the form:

$$\ell\left(\boldsymbol{\theta}; \rho\right) = \tau\left(\rho, T\left(\boldsymbol{\theta}\right) O\right), \tag{263}$$

where $O$ belongs to some dynamical Lie algebra $\mathfrak{g}$, $T\left(\boldsymbol{\theta}\right)$ is the adjoint action of some element of the compact Lie group $G$ associated with $\mathfrak{g}$, and $\rho$ is arbitrary. As demonstrated in Fontana et al. (2024), only the projection of $\rho$ onto the $\mathfrak{g}$ contributes to the loss, so we consider when $\rho \in \mathfrak{g}$ WLOG.

$G$ is a compact Lie group. Thus, up to an Abelian sector that does not contribute to the loss (as $O \in \mathfrak{g}$), $G$ can be written as the direct sum of simple compact Lie groups. By the well-known classification of simple compact Lie groups these are either the classical $\mathfrak{so}\left(N\right)$, $\mathfrak{su}\left(N\right)$, $\mathfrak{sp}\left(N\right)$, or one of the exceptional Lie groups. The exceptional Lie groups are of fixed dimension—i.e., there is no sense of an asymptotic limit—so we consider here the classical cases only.

We begin with the case $\mathfrak{g} \cong \mathfrak{su}\left(N\right)$, with defining representation $\mathrm{d}\phi$ over $\mathbb{C}$. It is easy to see that $\mathrm{i}\,\mathrm{d}\phi\left(O\right)$ and $\mathrm{i}\,\mathrm{d}\phi\left(\rho\right)$ are in the defining representation of the Jordan algebra $\mathcal{H}\left(\mathbb{C}\right)$, with identical trace form. Such a simple component is thus equivalent to a JASA with simple subalgebra isomorphic to $\mathcal{H}^N\left(\mathbb{C}\right)$.

We now consider when $\mathfrak{g} \cong \mathfrak{su}\left(N\right)$, with defining representation $\mathrm{d}\phi$ over $\mathbb{H}$. As the loss is real-valued we have the expansion in this sector:

$$\begin{aligned}
\ell\left(\boldsymbol{\theta}; \rho\right) \propto \mathrm{Tr}&\left(\mathrm{Re}\left\{\mathrm{i}\,\mathrm{d}\phi\left(\rho\right)\right\}\mathrm{d}\phi\left(U\left(\boldsymbol{\theta}\right)\right)\mathrm{Re}\left\{\mathrm{i}\,\mathrm{d}\phi\left(O\right)\right\}\mathrm{d}\phi\left(U\left(\boldsymbol{\theta}\right)\right)^\dagger\right)\\
&+ \mathrm{Tr}\left(\mathrm{Re}\left\{\mathrm{j}\,\mathrm{d}\phi\left(\rho\right)\right\}\mathrm{d}\phi\left(U\left(\boldsymbol{\theta}\right)\right)\mathrm{Re}\left\{\mathrm{j}\,\mathrm{d}\phi\left(O\right)\right\}\mathrm{d}\phi\left(U\left(\boldsymbol{\theta}\right)\right)^\dagger\right)\\
&+ \mathrm{Tr}\left(\mathrm{Re}\left\{\mathrm{k}\,\mathrm{d}\phi\left(\rho\right)\right\}\mathrm{d}\phi\left(U\left(\boldsymbol{\theta}\right)\right)\mathrm{Re}\left\{\mathrm{k}\,\mathrm{d}\phi\left(O\right)\right\}\mathrm{d}\phi\left(U\left(\boldsymbol{\theta}\right)\right)^\dagger\right).
\end{aligned} \tag{264}$$

Each of these real parts is Hermitian and can be written as elements of the defining representation of $\mathcal{H}^N\left(\mathbb{H}\right)$, reducing this simple component to that of a JASA.

We finally consider when $\mathfrak{g} \cong \mathfrak{so}\left(N\right)$. There are more subtleties compared with the other classical Lie algebras as, in the defining representation over $\mathbb{R}$, there is no simple relation between Hermitian and anti-Hermitian operators. To make the following concrete, we here consider the representation of $\mathfrak{so}\left(N\right)$ defined by Majorana operators $\gamma_i$, with generators:

$$M_{ij} = \frac{1}{2}\left[\gamma_i, \gamma_j\right], \tag{265}$$

i.e., polynomials quadratic in Majorana operators. These are the generators of the spin algebra $\mathfrak{spin}$ associated with the vector space spanned by the $\gamma_i$ and thus is isomorphic to $\mathfrak{so}\left(N\right)$. Furthermore, the gates generated by the $M_{ij}$ are just matchgates. The loss function variance of variational matchgate circuits was studied in detail in Diaz et al. (2023). The authors there also consider when $O \notin \mathfrak{g}$, making this an example of a barren plateau result beyond the LASA formalism.

We now demonstrate how this setting fits into the Jordan algebra picture. Let $\mathcal{A}$ be the Jordan algebra of Majorana fermions, with Jordan multiplication $\circ$ given by half the anticommutator. As in the Lie algebra picture, the dimension of this algebra grows exponentially with $n$; similarly, the Jordan algebra generated by $k$-body ($2 < k < n$) Majorana terms has dimension growing exponentially with $n$, as does the generated algebra when acting on these $k$-body terms via conjugation by a matchgate. However, though these operators are not closed as an algebra under matchgate conjugation, they *are* closed as a vector space. Furthermore, the trace form of a $k$-body term $\psi$ and a $k'$-body term $\psi'$ is always zero when $k \neq k'$. This allows us to instead consider $O$ as a member of a different Jordan algebra $\mathcal{A}'$, with the same underlying vector space as $\mathcal{A}$ and new Jordan multiplication operation $\bullet$ defined as:

$$\psi \bullet \psi' = \begin{cases} 1 & \text{if } \psi = \psi', \\ \psi & \text{if } \psi' = 1, \\ \psi' & \text{if } \psi = 1, \\ 0 & \text{otherwise.} \end{cases} \tag{266}$$

Note that this new multiplication operation still satisfies the Jordan identity:

$$\begin{aligned}
\left(\psi \bullet \psi'\right)\left(\psi \bullet \psi\right) &= \psi \bullet \psi'\\
&= \psi \bullet \left(\psi' \bullet 1\right)\\
&= \psi \bullet \left(\psi' \bullet \left(\psi \bullet \psi\right)\right).
\end{aligned} \tag{267}$$

It also preserves the canonical trace form, as:

$$\mathrm{Tr}\left(L_\circ\left(\psi\circ\psi'\right)\right) = 2^n\delta_{\psi,\psi'} = \mathrm{Tr}\left(L_\bullet\left(\psi\bullet\psi'\right)\right). \tag{268}$$

Here, $L_\bullet$ is the linear transformation associated with the Jordan multiplication $\bullet$:

$$L_\bullet\left(u\right)v = u\bullet v, \tag{269}$$

and similarly for $L_\circ$. The loss function in this sector is thus a JASA under the Jordan algebra $\mathcal{A}'$, and thus both general matchgate ansatzes or the LASA setting with $\mathfrak{g}\cong\mathfrak{so}\left(N\right)$ reduce to JASAs.

Interestingly, one can show that the matchgate setting *cannot* generally be described by a LASA while maintaining the trace form, even when the Lie bracket is redefined. To see this, we consider the adjoint endomorphism

$$\mathrm{ad}_{[\cdot,\cdot]}\left(u\right)v = [u,v] \tag{270}$$

associated with the Lie bracket $[\cdot,\cdot]$. The canonical trace form for Lie algebras is the *Killing form*, given by:

$$K\left(u,v\right) = \mathrm{Tr}\left(\mathrm{ad}_{[\cdot,\cdot]}\left(u\right)\mathrm{ad}_{[\cdot,\cdot]}\left(v\right)\right). \tag{271}$$

Note that (as $\mathrm{ad}_{[\cdot,\cdot]}\left(u\right)$ is antisymmetric):

$$K\left(u,u\right) = -\left\|\mathrm{ad}_{[\cdot,\cdot]}\left(u\right)\right\|_{\mathrm{F}}^2. \tag{272}$$

For $u$ a Majorana monomial under the usual Lie algebra of Majoranas, this is just (minus) the number of terms anticommuting with the monomial. For $u$ a degree-1 monomial this will grow exponentially in the number of Majoranas $n$. If instead one considered defining a new Lie bracket $[\cdot,\cdot]'$ that is zero when mapping from $k$-degree Majorana monomials to those of different degree, $K\left(\cdot,\cdot\right)$ would always be zero on degree-1 Majorana monomials—in particular, the trace form is not preserved when modifying the Lie bracket.

## I   REDUCTION OF LOW-PURITY INPUTS TO MAXIMALLY MIXED INPUTS

We elaborate here on the claim made in Appendix B.2 that if the input state $\rho$ projected into a simple sector $\mathcal{A}_\alpha$ has sufficiently low purity, the loss contribution from this sector does not contribute in any meaningful way. This is true even when the loss is rescaled by some $\mathcal{N}\leq\mathrm{O}\left(N^{0.99}\right)$ as we consider elsewhere. Intuitively this is due to $\rho^\alpha$ being close to maximally mixed and thus the contribution to the loss in this algebraic sector is effectively constant. We formally state this claim as the following lemma.

**Lemma 29** (Low purity inputs are approximately trivial). *Let $\mathcal{J},\ell$ be as in Theorem 13. For a $\rho\in\mathcal{R}$ such that for some $\alpha$:*

$$\mathrm{Tr}\left(\left(\rho^\alpha\right)^2\right)\leq N_\alpha^{-0.999}, \tag{273}$$

*we have for sufficiently large $\epsilon^{-1},t$ that:*

$$\mathrm{Var}\left[\mathcal{N}\ell^\alpha\left(\boldsymbol{\theta};\boldsymbol{\rho}\right)\right]\leq\mathrm{O}\left(\frac{\mathcal{N}^2}{N_\alpha^{1.997}}\right)+\mathrm{O}\left(\pi\right)=\mathrm{o}\left(1\right) \tag{274}$$

*for all $\mathcal{N}\leq\mathrm{O}\left(N^{0.99}\right)$.*

*Proof.* Assume $\mathrm{Tr}\left(\rho^\alpha\right)=1$ WLOG by otherwise absorbing a factor of $\mathrm{Tr}\left(\rho^\alpha\right)$ into $O^\alpha$. Consider the decomposition into eigenspaces:

$$\rho^\alpha = \sigma^\alpha + \tau^\alpha, \tag{275}$$

where the nonzero eigenvalues of $\sigma^\alpha$ are the largest $\lfloor N_\alpha^{0.999}\rfloor$ eigenvalues of $\rho^\alpha$. Iterate this procedure

$$M_\alpha \equiv \left\lceil\frac{N_\alpha}{\lfloor N_\alpha^{0.999}\rfloor}\right\rceil \tag{276}$$

total times to yield the decomposition:

$$\rho^\alpha = \sigma^\alpha + \sum_{i=1}^{M_\alpha-1}\tau_i^\alpha. \tag{277}$$

The standard deviation of a sum of random variables is upper bounded by the sums of the standard deviations. Furthermore, from Eq. (277) we have that

$$\text{Tr}\left(\left(\boldsymbol{\rho}^\alpha\right)^2\right) \geq \text{Tr}\left(\left(\boldsymbol{\sigma}^\alpha\right)^2\right). \tag{278}$$

We can thus upper bound the standard deviation of the loss by applying Corollary 20 to the terms in Eq. (277):

$$\begin{aligned}
\text{Var}\left[\mathcal{N}\ell^\alpha\left(\boldsymbol{\theta}; \boldsymbol{\rho}\right)\right] &\leq \mathcal{N}^2 M_\alpha^2 \frac{\text{Tr}\left(\left(\boldsymbol{O}^\alpha\right)^2\right) \text{Tr}\left(\left(\boldsymbol{\sigma}^\alpha\right)^2\right)}{\dim_{\mathbb{R}}\left(\mathfrak{g}_\alpha\right)} + \text{O}\left(\pi\right) \\
&\leq \mathcal{N}^2 M_\alpha^2 \frac{\text{Tr}\left(\left(\boldsymbol{O}^\alpha\right)^2\right) \text{Tr}\left(\left(\boldsymbol{\rho}^\alpha\right)^2\right)}{\dim_{\mathbb{R}}\left(\mathfrak{g}_\alpha\right)} + \text{O}\left(\pi\right) \\
&\leq \text{O}\left(\frac{\mathcal{N}^2}{N^{1.997}}\right) + \text{O}\left(\pi\right),
\end{aligned} \tag{279}$$

yielding the final result. $\qquad\square$

## J $\epsilon$-APPROXIMATE 2-DESIGNS SUFFICE FOR THE $\mathcal{G}_i$

We here argue that one need only assume that the $\mathcal{G}_i$ form $\epsilon$-approximate 2-designs—not large-$t$ $t$-designs—if one takes the overall normalization $\mathcal{N}$ to be at most $\text{o}\left(N^{\frac{2}{3}}\right)$ and if $p$ grows sufficiently slowly with the $N_\alpha$. More concretely, instead of taking Assumption 15 we may take the following assumption:

**Assumption 30** (Scaling of parameter space with ansatz moments)**.** The number of trained parameters $p$ satisfies:[11]

$$p^2 \leq \text{o}\left(\min_\alpha\left(\frac{\log\left(\mathcal{N}^{-1} N_\alpha^{\frac{2}{3}}\right)}{\log\log\left(\mathcal{N}^{-1} N_\alpha^{\frac{2}{3}}\right)}\right)\right), \tag{280}$$

where the $\mathcal{G}_i^\alpha$ are i.i.d. $\epsilon$-approximate 2-designs over $\mathcal{T}_\alpha = \text{Aut}_1\left(\mathcal{A}_\alpha\right)$, where

$$\epsilon \leq \text{O}\left(N_\alpha^{-\frac{2}{3}}\right). \tag{281}$$

To achieve this we will rely on tools from random matrix theory to bound the error of various mixed cumulants of our expressions under such a change. In particular, we will use the existence of so-called *limit distributions of all orders*.

**Definition 31** (Limit distribution of all orders (Collins et al., 2007))**.** A sequence of sets of $N \times N$ random matrices $\{\boldsymbol{A}_i\}_{i=1}^d$ has a *limit distribution of all orders* if there exist functions $\alpha_{\boldsymbol{k}}$ such that the cumulants $\kappa_{\boldsymbol{k}}$ obey as $N \to \infty$:

$$\begin{aligned}
\lim_{N\to\infty} N^{\|\boldsymbol{k}\|_1 - 2} &\kappa_{\boldsymbol{k}}\left(\text{Tr}\left(p_1\left(\boldsymbol{A}_1, \ldots, \boldsymbol{A}_s\right)\right), \ldots, \text{Tr}\left(p_r\left(\boldsymbol{A}_1, \ldots, \boldsymbol{A}_s\right)\right)\right) \\
&= \alpha_{\boldsymbol{k}}\left(p_1\left(\boldsymbol{A}_1, \ldots, \boldsymbol{A}_s\right), \ldots, p_r\left(\boldsymbol{A}_1, \ldots, \boldsymbol{A}_s\right)\right)
\end{aligned} \tag{282}$$

for all fixed polynomials $p_i$. Here, $r \equiv \|\boldsymbol{k}\|_0$.

Simple classes of distributions which satisfy this property include constant matrices of bounded operator norm, Gaussian random matrices, Wishart random matrices, and Haar random unitaries. We here will be interested in a weaker notion of the existence of a limit distribution of all orders, requiring only that higher-order moments vanish sufficiently quickly and not necessarily that they have well-defined limits.

**Definition 32** (Weak limit distribution of all orders)**.** A sequence of sets of $N \times N$ random matrices $\{\boldsymbol{A}_i\}_{i=1}^d$ has a *weak limit distribution of all orders* if the cumulants $\kappa_{\boldsymbol{k}}$ obey as $N \to \infty$:

$$\kappa_{\boldsymbol{k}}\left(\text{Tr}\left(p_1\left(\boldsymbol{A}_1, \ldots, \boldsymbol{A}_s\right)\right), \ldots, \text{Tr}\left(p_r\left(\boldsymbol{A}_1, \ldots, \boldsymbol{A}_s\right)\right)\right) = \text{O}\left(N^{2 - \|\boldsymbol{k}\|_1}\right) \tag{283}$$

for all fixed polynomials $p_i$. Here, $r \equiv \|\boldsymbol{k}\|_0$.

---

[11]As in Assumption 15, this can be weakened to a bound on $p$ if one is only interested in the gradient.

Products of matrices with limit distributions of all orders over $\mathbb{F} = \mathbb{R}, \mathbb{C}, \mathbb{H}$ were considered in Mingo & Popa (2013); Collins et al. (2007); Redelmeier (2021) respectively, allowing us to prove the following lemma.

**Lemma 33** (Existence of weak limit distributions of all orders). *Let $L_i$, $i \in [d]$ be multilinear functions of the form:*

$$L_i = \mathrm{Tr}\left(M_i h O h^\dagger\right), \tag{284}$$

*where $h$ is Haar random and independent from the $M_i$. Assume all $M_i$ have weak limit distributions of all orders. For any cumulant $\kappa_{\boldsymbol{k}}\left(L_{i_1}, \ldots, L_{i_r}\right)$ with $\|\boldsymbol{k}\|_1 \geq 3$,*

$$\kappa_{\boldsymbol{k}}\left(L_{i_1}, \ldots, L_{i_r}\right) = \mathrm{O}\left(N^{2-\|\boldsymbol{k}\|_1}\right). \tag{285}$$

*Proof.* Following:

1. Lemma 5.12 of Redelmeier (2012) when $\mathbb{F} = \mathbb{R}$;

2. Remark 4.8 of Collins et al. (2007) when $\mathbb{F} = \mathbb{C}$;

3. and Corollary 3.2 of Redelmeier (2021) when $\mathbb{F} = \mathbb{H}$,

the product of a matrix from a unitarily-invariant ensemble with any other independent matrix with a weak limit distribution of all orders has a weak limit distribution of all orders. This implies that the $M_i h O h^\dagger$ have a weak limit distribution of all orders. The vanishing of higher-order cumulants as in Eq. (285) then follows from Definition 32. $\qquad\square$

We thus have the following lemma.

**Lemma 34** (Weak convergence to Haar random $g_i$). *Assume Assumption 30. Let $L_i$, $i \in [d]$ be uniformly bounded multilinear functions of the form:*

$$L_i = \mathrm{Tr}\left(\rho h O h^\dagger \prod_{j \in \mathcal{I}_i} g_j M_j g_j^\dagger\right), \tag{286}$$

*where $\mathcal{I}_i$ is some index set associated with $i$ of constant cardinality, $h$ is Haar random, $g_j \sim \mathcal{G}_j$, $\rho$ and the $M_i$ are fixed with bounded trace norm, and $d = \mathrm{o}\left(\frac{\log\left(\mathcal{N}^{-1}N^{\frac{2}{3}}\right)}{\log\log\left(\mathcal{N}^{-1}N^{\frac{2}{3}}\right)}\right)$. Let $\hat{L}_i$ be the same multilinear functions, where now the $g_j$ are Haar random. The joint distribution of $\mathcal{N}L_i$ differs from the joint distribution of $\mathcal{N}\hat{L}_i$ by an error at most*

$$\pi_2 = \mathrm{O}\left(\frac{d\log\left(\mathcal{N}^{-1}N^{\frac{2}{3}}\right)}{\left(\mathcal{N}^{-1}N^{\frac{2}{3}}\right)^{0.99}}\right) \tag{287}$$

*in Lévy–Prokhorov metric for any $\mathcal{N} \leq \mathrm{O}\left(N\right)$.*

*Proof.* The $g_j M_j g_j^\dagger$ have a limit distribution of all orders due to the $M_j$ being fixed. Thus, iteratively one can show the trace arguments of the $L_i$ have weak limit distributions of all orders by the results cited in the proof of Lemma 33. Incorporating the bounded trace norm of the $M_i$ and the normalization $\mathcal{N}$ gives an error for the cumulants of order $k > 2$:

$$\tilde{\epsilon} = \mathrm{O}\left(\mathcal{N}^k N^{1-k}\right) \leq \mathrm{O}\left(\left(\mathcal{N}N^{-\frac{2}{3}}\right)^k\right). \tag{288}$$

This also bounds the $k = 1, 2$ cases whenever

$$\epsilon \leq \mathrm{O}\left(N^{-\frac{2}{3}}\right), \tag{289}$$

which is true by Assumption 30. The result then follows from Corollary 40, where $\mu$ is subleading as in the proof of Lemma 26. $\qquad\square$

## K    EQUICONTINUITY OF PROBABILITY DENSITIES

In Appendix D we prove that the joint distribution of the loss function and its first two derivatives for a certain random class of variational quantum loss functions converges in distribution to a fairly simple expression involving Wishart-distributed random matrices. However, Corollaries 14 and 18 are stated in terms of the *probability densities* of the loss and first derivative, and weak convergence does not necessarily imply pointwise convergence in densities. However, it is known that this *is* the case when the sequence of densities are *equicontinuous* and bounded (Boos, 1985). Whether or not this is true depends on the details of the distributions $\mathcal{G}, \boldsymbol{H}$. As an example, we here show this is true for the loss when $\mathcal{G}, \boldsymbol{H}$ are Haar random, under reasonable assumptions (which we conjecture are not necessary) on the spectrum of the objective observable $\boldsymbol{O}$ projected into any simple component.

**Lemma 35** (Equicontinuity of loss density with Haar random ansatz). *Consider a sequence of $\boldsymbol{O}$ as $N \to \infty$ with eigenvalues:*

$$0 = o_1 \leq o_2 \leq \ldots \leq o_{N-1} \leq o_N = 1, \tag{290}$$

*with mean eigenvalue $\overline{o}$ and $\frac{\mathrm{Tr}(\boldsymbol{O})^2}{\mathrm{Tr}(\boldsymbol{O}^2)N} = \frac{r}{N} = \Theta\left(1\right)$. For $\boldsymbol{w} \in \mathbb{R}^{N-2}$, define:*

$$E_{\boldsymbol{w}} \equiv \sqrt{\tilde{\boldsymbol{o}} \cdot (\boldsymbol{w} \odot \boldsymbol{w})}, \tag{291}$$

$$R_{\boldsymbol{w}} \equiv \sqrt{\boldsymbol{w}^{\mathsf{T}} \cdot \boldsymbol{w}}, \tag{292}$$

*where $\tilde{\boldsymbol{o}}$ is the vector $(o_2, \ldots, o_{N-1})$ and $\odot$ denotes the Hadamard product. Let $D_x$ be the domain of $\boldsymbol{w}$ satisfying:*

$$E_{\boldsymbol{w}}^2 \leq \overline{o} + N^{-\frac{1}{2}}x \leq 1 + E_{\boldsymbol{w}}^2 - R_{\boldsymbol{w}}^2. \tag{293}$$

*Assume*

$$p_{\sqrt{N}\ell}(x) = \frac{1}{\sqrt{N\left(\overline{o} + N^{-\frac{1}{2}}x\right)}S_{N-1}} \int_{D_x} \mathrm{d}\boldsymbol{w} \left(1 - \left(\overline{o} + N^{-\frac{1}{2}}x\right)^{-1} E_{\boldsymbol{w}}^2\right)^{-\frac{1}{2}} \tag{294}$$

*is locally Lipschitz and bounded at any $x$ as $N \to \infty$, with Lipschitz constants independent from $N$. Then the density of $\sqrt{r}\ell$ is bounded by a constant independent of $N$ and is equicontinuous as a sequence in $N$.*

*Proof.* By assumption $r = \Theta\left(N\right)$ so we consider the density of $\sqrt{N}\ell$ WLOG. Let $\{|o_i\rangle\}_{i=1}^{N}$ be the eigenvectors of $\boldsymbol{O}$ associated with the eigenvalues $o_i$. Let:

$$|u\rangle = \sum_{i=1}^{N} u_i |o_i\rangle, \tag{295}$$

and let:

$$S(\boldsymbol{u}) = \sum_{i=2}^{N} o_i |u_i|^2. \tag{296}$$

Let $p_{\sqrt{N}\ell}(x)$ be the density of $\sqrt{N}\ell = \sqrt{N}\overline{o} + x$. We have that:

$$p_{\sqrt{N}\ell}(x) = S_{\beta N-1}^{-1} \int_{\|\boldsymbol{u}\|_2=1} \mathrm{d}\boldsymbol{u}\, \delta\left(\sqrt{N}\left(S(\boldsymbol{u}) - \overline{o}\right) - x\right). \tag{297}$$

Here, $S_{\beta N-1}$ is the surface area of the $(\beta N - 1)$-sphere:

$$S_{\beta N-1} = \frac{2\pi^{\frac{\beta N}{2}}}{\Gamma\left(\frac{\beta N}{2}\right)}. \tag{298}$$

We can integrate out the phases of the $u_i$ and note that the delta function has no dependence on $u_1$ since $o_1 = 0$. Therefore,

$$p_{\sqrt{N}\ell}(x) = S_{N-1}^{-1} 2^N \int_{\substack{\|\tilde{\boldsymbol{u}}\|_2 \leq 1, \\ \tilde{\boldsymbol{u}} \geq \boldsymbol{0}}} \mathrm{d}\tilde{\boldsymbol{u}}\, \delta\left(\sqrt{N}\left(S(\tilde{\boldsymbol{u}}) - \overline{o}\right) - x\right). \tag{299}$$

Here, $\tilde{\boldsymbol{u}} \in \mathbb{R}_{\geq 0}^{N-1}$ has indices in $2, \ldots, N$ such that $\tilde{u}_i = |u_i|$. We will similarly use $\bar{\boldsymbol{o}}$ to denote $\boldsymbol{o}$ with the $o_1 = 0$ component projected out.

We now proceed to show equicontinuity and uniform boundedness of $p_f(x)$ by showing that the derivative of $p_f(x)$ is bounded everywhere by some $N$-independent constant. By symmetry we restrict to the components of $v_N \geq 0$, introducing an overall factor of $2^\beta$. We also introduce new coordinates $w_2, \ldots, w_N$, where $w_i = \tilde{u}_i$ for $2 \leq i \leq N-1$ and $w_N = S(\tilde{\boldsymbol{u}})$. The constraint $\tilde{u}_N^2 \geq 0$ becomes in these coordinates:

$$\sum_{i=2}^{N-1} o_i w_i^2 \leq w_N. \tag{300}$$

Only the final row of the resulting Jacobian is nontrivial and is equal to:

$$\boldsymbol{J_N} = 2\tilde{\boldsymbol{o}}^\mathsf{T} \odot \tilde{\boldsymbol{u}}^\mathsf{T}, \tag{301}$$

where $\odot$ denotes the Hadamard product. This gives a Jacobian determinant of (recalling that $o_N = 1$):

$$\det(\boldsymbol{J}) = 2\tilde{u}_N = 2\sqrt{w_N - \sum_{i=2}^{N-1} o_i w_i^2}. \tag{302}$$

Putting everything together yields a density:

$$p_{\sqrt{N}\ell}(x) = S_{N-1}^{-1} 2^{N-2} \int_D \mathrm{d}\boldsymbol{w} \, \frac{\delta\left(\sqrt{N}\left(w_N - \bar{o}\right) - x\right)}{\sqrt{w_N - \sum_{i=2}^{N-1} o_i w_i^2}}, \tag{303}$$

where $D$ is the domain of positive $w_i$ obeying the normalization constraint of Eq. (300) as well as:

$$w_N + \sum_{i=2}^{N-1} (1 - o_i) w_i^2 \leq 1. \tag{304}$$

Defining:

$$E_{\tilde{\boldsymbol{w}}}^2 = \sum_{i=2}^{N-1} o_i \tilde{w}_i^2, \tag{305}$$

$$R_{\tilde{\boldsymbol{w}}}^2 = \sum_{i=2}^{N-1} \tilde{w}_i^2, \tag{306}$$

and rescaling $w_N$ we can rewrite this as:

$$p_{\sqrt{N}\ell}(x) = \frac{1}{\sqrt{N} S_{N-1}} \int_{B^{N-2}} \mathrm{d}\tilde{\boldsymbol{w}} \int_{\sqrt{N} E_{\tilde{\boldsymbol{w}}}^2}^{\sqrt{N}\left(E_{\tilde{\boldsymbol{w}}}^2 + 1 - R_{\tilde{\boldsymbol{w}}}^2\right)} \mathrm{d}w_N \, \frac{\delta\left(w_N - \sqrt{N}\bar{o} - x\right)}{\sqrt{N^{-\frac{1}{2}} w_N - E_{\tilde{\boldsymbol{w}}}^2}}, \tag{307}$$

where $B^{N-2}$ is the unit $(N-2)$-ball. Integrating out the delta function then yields:

$$p_{\sqrt{N}\ell}(x) = \frac{1}{\sqrt{N\left(\bar{o} + N^{-\frac{1}{2}}x\right)} S_{N-1}} \int_{D_x} \mathrm{d}\tilde{\boldsymbol{w}} \left(1 - \left(\bar{o} + N^{-\frac{1}{2}}x\right)^{-1} E_{\tilde{\boldsymbol{w}}}^2\right)^{-\frac{1}{2}}, \tag{308}$$

where $D_x$ is the domain of $\tilde{\boldsymbol{w}}$ satisfying:

$$E_{\tilde{\boldsymbol{w}}}^2 \leq \bar{o} + N^{-\frac{1}{2}}x \leq 1 + E_{\tilde{\boldsymbol{w}}}^2 - R_{\tilde{\boldsymbol{w}}}^2. \tag{309}$$

By assumption this is locally Lipschitz and bounded as $N \to \infty$ with Lipschitz constants independent from $N$, and thus is equicontinuous and bounded. $\qquad \square$

We now discuss the main assumption of Lemma 35 in more detail, particularly the local Lipschitz continuity of Eq. (294). Note that the defining equations of $D_x$ imply that all $w_i^2 \leq 1$, i.e., the domain of integration is always of $\mathrm{O}\,(1)$ volume. This holds true even if some $o_i$ vanish with $N$; it is apparent that these terms do not contribute to the integrand at leading order $\boldsymbol{O}$, and only the $\Theta\,(1)$ eigenvalues contribute. Because of this, we will perform an example calculation of the Lipschitz constants of Eq. (294) when all $o_i = \bar{o} = \frac{1}{2}$ for $i \neq 1, N$, though due to this observation similar results hold for all reasonable spectrums.

For such $\boldsymbol{O}$,

$$E_{\tilde{\boldsymbol{w}}}^2 = \frac{1}{2} R_{\tilde{\boldsymbol{w}}}^2, \tag{310}$$

so

$$
\begin{aligned}
p_{\sqrt{N}\ell}\,(x) &= \frac{\sqrt{2}}{\sqrt{N\left(1 + 2N^{-\frac{1}{2}}x\right)}S_{N-1}} \int_{D_x} \mathrm{d}\tilde{\boldsymbol{w}} \left(1 - \left(1 + 2N^{-\frac{1}{2}}x\right)^{-1} R_{\tilde{\boldsymbol{w}}}^2\right)^{-\frac{1}{2}} \\
&= \frac{\sqrt{2}S_{N-3}}{\sqrt{N\left(1 + 2N^{-\frac{1}{2}}x\right)}S_{N-1}} \int_0^{\sqrt{1 - 2N^{-\frac{1}{2}}|x|}} \mathrm{d}r\, r^{N-3} \left(1 - \left(1 + 2N^{-\frac{1}{2}}x\right)^{-1} r^2\right)^{-\frac{1}{2}} \\
&= \frac{S_{N-3}}{\sqrt{2N\left(1 + 2N^{-\frac{1}{2}}x\right)}S_{N-1}} \int_0^{1 - 2N^{-\frac{1}{2}}|x|} \mathrm{d}r\, r^{\frac{N}{2}-2} \left(1 - \left(1 + 2N^{-\frac{1}{2}}x\right)^{-1} r\right)^{-\frac{1}{2}} \\
&= \frac{\left(1 - 2N^{-\frac{1}{2}}|x|\right)^{\frac{N}{2}-1} S_{N-3}}{\sqrt{2N\left(1 + 2N^{-\frac{1}{2}}x\right)}S_{N-1}} \int_0^1 \mathrm{d}r\, r^{\frac{N}{2}-2} \left(1 - \frac{1 - 2N^{-\frac{1}{2}}|x|}{1 + 2N^{-\frac{1}{2}}x} r\right)^{-\frac{1}{2}} \\
&= \frac{\left(1 - 2N^{-\frac{1}{2}}|x|\right)^{\frac{N}{2}-1} \Gamma\left(\frac{N}{2}\right)}{\pi\sqrt{2N\left(1 + 2N^{-\frac{1}{2}}x\right)}\,\Gamma\left(\frac{N}{2}-1\right)} \int_0^1 \mathrm{d}r\, r^{\frac{N}{2}-2} \left(1 - \frac{1 - 2N^{-\frac{1}{2}}|x|}{1 + 2N^{-\frac{1}{2}}x} r\right)^{-\frac{1}{2}}.
\end{aligned}
\tag{311}
$$

This integral is just the integral definition of the hypergeometric function $_2\mathrm{F}_1$, yielding:

$$p_{\sqrt{N}\ell}\,(x) = \frac{\left(1 - 2N^{-\frac{1}{2}}|x|\right)^{\frac{N}{2}-1}}{\pi\sqrt{2N\left(1 + 2N^{-\frac{1}{2}}x\right)}} \,_2\mathrm{F}_1\left(\frac{1}{2}, \frac{N}{2}-1; \frac{N}{2}; \frac{1 - 2N^{-\frac{1}{2}}|x|}{1 + 2N^{-\frac{1}{2}}x}\right). \tag{312}$$

We now consider the large-$N$ asymptotics of the derivative of this expression. By Gauss's hypergeometric theorem, for $z = 1 + \mathrm{O}\left(N^{-\frac{1}{2}}\right)$,

$$_2\mathrm{F}_1\left(\frac{1}{2}, \frac{N}{2}-1; \frac{N}{2}; z\right) = \left(1 + \mathrm{O}\left(N^{-\frac{1}{2}}\right)\right) \frac{\Gamma\left(\frac{N}{2}\right)\Gamma\left(\frac{1}{2}\right)}{\Gamma\left(\frac{N-1}{2}\right)\Gamma\left(1\right)} = \mathrm{O}\left(\sqrt{N}\right). \tag{313}$$

Similarly, we have by the derivative rule for the hypergeometric function that:

$$_2\mathrm{F}_1{}'\left(\frac{1}{2}, \frac{N}{2}-1; \frac{N}{2}; z\right) = \frac{1}{2}\left(1 - 2N^{-1}\right) {}_2\mathrm{F}_1\left(\frac{3}{2}, \frac{N}{2}; \frac{N}{2}+1; z\right) = \mathrm{O}\left(N^{\frac{3}{2}}\right). \tag{314}$$

Taking into account the $N^{-\frac{1}{2}}$ factors scaling $x$ thus yields (where defined):

$$p'_{\sqrt{N}\ell}\,(x) = \mathrm{O}\,(1)\,. \tag{315}$$

In particular, $p_{\sqrt{N}\ell}\,(x)$ is locally Lipschitz with Lipschitz constants independent from $N$. These $\boldsymbol{O}$ thus satisfy the assumptions of Lemma 35, as claimed.

## L    BOUNDING LARGE DEVIATIONS OF THE HESSIAN DETERMINANT

In calculating the density of local minima in Appendix C.3 we require analyzing the asymptotics of:

$$\mathbb{E}\left[\det\left(\boldsymbol{D}_z\right)\mathbf{1}\left\{\boldsymbol{D}_z \succeq \mathbf{0}\right\}\right], \tag{316}$$

where $\boldsymbol{D}_z$ is as in Eq. (124), i.e., it is a Wishart-distributed random matrix normalized by its degrees of freedom parameter. As the determinant is exponentially sensitive to its argument, exponentially unlikely deviations in the convergence of the spectrum of $\boldsymbol{D}_z$ to its asymptotic value can, in principle, cause this expectation to not converge to the determinant of the (almost sure) asymptotic spectral measure of $\boldsymbol{D}_z$. The same is true of the minimum eigenvalue of $\boldsymbol{D}_z$. We here show that these deviations do not asymptotically contribute to Eq. (316). We will here be brief as this is not the main focus of our work, and instead refer the interested reader to Anschuetz (2022) for a more detailed account of very similar arguments. In the following we consider only the case when $\gamma \leq 1$ as otherwise $\boldsymbol{D}_z$ is never full-rank and Eq. (316) is identically zero.

We begin with the empirical spectral measure of $\boldsymbol{D}_z$. The empirical spectral measures of $\boldsymbol{W}$ satisfies a large deviations principle at a speed $p^2$ with good rate function maximized by the Marčenko–Pastur distribution $\mathrm{d}\mu_{\mathrm{MP}}^{\gamma}\left(\lambda\right)$ (Hiai & Petz, 1998). As the determinant is only sensitive at a speed $p$, i.e.,

$$\det\left(\boldsymbol{D}_z\right) = \exp\left(p\int\mathrm{d}\mu_{\mathrm{MP}}^{\gamma}\left(\lambda\right)\ln\left(\lambda\right)\right), \tag{317}$$

by Varadhan's lemma (Dembo & Zeitouni, 2010) the expected determinant converges to the determinant of our asymptotic expression for $\boldsymbol{D}_z$.

We now consider the smallest eigenvalue of $\boldsymbol{D}_z$. For fluctuations below its asymptotic value this satisfies a large deviations principle with good rate function at a speed $p$, where this rate function is maximized at the infimum of the support of the Marčenko–Pastur distribution; for fluctuations above its asymptotic value it satisfies a large deviations principle at a speed $p^2$ (Katzav & Pérez Castillo, 2010). By Varadhan's lemma, then, only fluctuations below the asymptotic value potentially contribute, but as these fluctuations only decrease the value of the argument of Eq. (316) they do not contribute asymptotically (Dembo & Zeitouni, 2010). These together imply that Eq. (316) converges as claimed in Appendix C.3.

## M    LEMMAS ON CONVERGENCE IN DISTRIBUTION

We here prove a few helper lemmas that will allow us to obtain quantitative error bounds on the rate of convergence of two sequences of distributions given bounds on the differences of their moments or cumulants. We will achieve this by bounding the *Lévy–Prokhorov distance* between the two distributions, which metricizes weak convergence. We first restate a result of Berkes & Philipp (1979) which bounds this distance, with a slight modification due to our use of the infinity norm rather than the Euclidean norm (see Appendix A.5).

**Theorem 36** (Slightly modified version of Lemma 2.2, Berkes & Philipp (1979))**.** *Let $p\left(\boldsymbol{x}\right), q\left(\boldsymbol{x}\right)$ be two distributions on $\mathbb{R}^d$ with characteristic functions $\phi\left(\boldsymbol{u}\right), \gamma\left(\boldsymbol{u}\right)$, respectively. Assume there exists a $C$ such that $\int_{\|\boldsymbol{x}\|_\infty \geq C} p\left(\boldsymbol{x}\right) \leq \mu$. There exists a universal constant $K$ such that for all $T \geq \max\left(2C, Kd\right)$, the Lévy–Prokhorov distance between $p$ and $q$ is bounded by:*

$$\pi\left(p, q\right) \leq \left(\frac{T}{\pi}\right)^d \int_{\|\boldsymbol{u}\|_\infty \leq T} \left|\phi\left(\boldsymbol{u}\right) - \gamma\left(\boldsymbol{u}\right)\right|\mathrm{d}\boldsymbol{u} + \frac{16\ln\left(T\right)}{T} + \mu. \tag{318}$$

Motivated by this, we prove a general bound on the error of the characteristic functions given errors in the moments.

**Lemma 37** (Bound on characteristic functions from moments)**.** *Let $p\left(\boldsymbol{x}\right), q\left(\boldsymbol{x}\right)$ be two probability densities on $\mathbb{R}^d$ with characteristic functions $\phi\left(\boldsymbol{u}\right), \gamma\left(\boldsymbol{u}\right)$. Assume each moment of order $k \leq t$ of $p$ differs from that of $q$ by an additive error at most $\epsilon > 0$, and assume all moments of order $k > t$ are bounded by $\left(C\sqrt{k}\right)^k$ for some $k$-independent $C \geq 0$. Then, for all $\boldsymbol{u}$ such that $\|\boldsymbol{u}\|_1 \leq \frac{\sqrt{t+1}}{2eC}$,*

$$\left|\phi\left(\boldsymbol{u}\right) - \gamma\left(\boldsymbol{u}\right)\right| \leq \epsilon\exp\left(\|\boldsymbol{u}\|_1\right) + \exp_2\left(1 - t\right). \tag{319}$$

*Proof.* We have by the definition of the characteristic function and the triangle inequality that:

$$|\phi(\boldsymbol{u}) - \gamma(\boldsymbol{u})| \leq \epsilon \sum_{\boldsymbol{k} \neq \boldsymbol{0} \in \mathbb{N}_0^d} \prod_{j=1}^d \frac{\left|(iu_j)^{k_j}\right|}{k_j!} + 2 \sum_{\|\boldsymbol{k}\|_1 \geq t+1} C^{\|\boldsymbol{k}\|_1} \|\boldsymbol{k}\|_1^{\frac{\|\boldsymbol{k}\|_1}{2}} \prod_{j=1}^d \frac{\left|(iu_j)^{k_j}\right|}{k_j!}. \quad (320)$$

The first term has the upper bound:

$$\epsilon \sum_{\boldsymbol{k} \neq \boldsymbol{0} \in \mathbb{N}_0^d} \prod_{j=1}^d \frac{\left|(iu_j)^{k_j}\right|}{k_j!} \leq \epsilon \exp(\|\boldsymbol{u}\|_1). \quad (321)$$

For the second term, we have the upper bound (for $\|\boldsymbol{u}\|_1 < \frac{\sqrt{t+1}}{eC}$):

$$\sum_{\|\boldsymbol{k}\|_1 \geq t+1} C^{\|\boldsymbol{k}\|_1} \|\boldsymbol{k}\|_1^{\frac{\|\boldsymbol{k}\|_1}{2}} \prod_{j=1}^d \frac{|u_j|^{k_j}}{k_j!} \leq \sum_{k \geq t+1} \frac{\left\|\sqrt{e}C\boldsymbol{u}\right\|_1^k}{\sqrt{k!}}$$

$$\leq \sum_{k \geq t+1} \left\|\frac{eC\boldsymbol{u}}{\sqrt{t+1}}\right\|_1^k \quad (322)$$

$$= \frac{\left(\frac{eC}{\sqrt{t+1}}\|\boldsymbol{u}\|_1\right)^{t+1}}{1 - \frac{eC}{\sqrt{t+1}}\|\boldsymbol{u}\|_1}.$$

When $\|\boldsymbol{u}\|_1 \leq \frac{\sqrt{t+1}}{2eC}$ this gives the bound:

$$2 \sum_{\|\boldsymbol{k}\|_1 \geq t+1} C^{\|\boldsymbol{k}\|_1} \|\boldsymbol{k}\|_1^{\frac{\|\boldsymbol{k}\|_1}{2}} \prod_{j=1}^d \frac{\left|(iu_j)^{k_j}\right|}{k_j!} \leq 4 \left(\frac{eC}{\sqrt{t+1}}\|\boldsymbol{u}\|_1\right)^{t+1} \leq \exp_2(1-t). \quad (323)$$

Combining these bounds yields the final result. $\qquad \square$

This characteristic function bound combined with Theorem 36 gives us the following corollary.

**Corollary 38** (Bound on Lévy–Prokhorov metric from moments). *Let $p_N(\boldsymbol{x}), q_N(\boldsymbol{x})$ be two sequences of distributions on $\mathbb{R}^d$ with characteristic functions $\phi_N(\boldsymbol{u}), \gamma_N(\boldsymbol{u})$, respectively. Assume each moment of order $k \leq t$ of $p_N$ differs from that of $q_N$ by an ($N$-dependent) additive error at most $\epsilon > 0$, and assume all moments of order $k > t$ (for $t$ $N$-dependent) are bounded by $\left(C\sqrt{k}\right)^k$ for some $(k, N)$-independent constant $C$. Assume as well that $\int_{\|\boldsymbol{x}\|_\infty \geq \frac{1}{2} \min\left(\frac{0.99}{d}\ln(\epsilon^{-1}), \frac{\sqrt{t+1}}{2eCd}\right)} p(\boldsymbol{x}) \leq \mu$ for some ($N$-dependent) $\mu$. The Lévy–Prokhorov distance between $p_N, q_N$ is bounded by:*

$$\pi(p_N, q_N) = O\left(\frac{d\log\log(\epsilon^{-1})}{\log(\epsilon^{-1})} + \frac{d\log(t)}{\sqrt{t}} + \mu\right). \quad (324)$$

*Proof.* Let $T = \min\left(\frac{0.99}{d}\ln(\epsilon^{-1}), \frac{\sqrt{t+1}}{2eCd}\right)$. From Theorem 36,

$$\pi(p_N, q_N) \leq \tilde{O}\left(T^{2d}\right) \sup_{\|\boldsymbol{u}\|_\infty \leq T} |\phi_N(\boldsymbol{u}) - \gamma_N(\boldsymbol{u})| + O\left(\frac{d\log(T)}{T}\right) + O(\mu). \quad (325)$$

From Lemma 37,

$$\sup_{\|\boldsymbol{u}\|_\infty \leq T} |\phi_N(\boldsymbol{u}) - \gamma_N(\boldsymbol{u})| \leq \epsilon^{0.01} + \exp_2(1-t). \quad (326)$$

The convergence rate given in Eq. (324) is trivial when $d = \Omega\left(\min\left(\frac{\log(\epsilon^{-1})}{\log\log(\epsilon^{-1})}, \frac{\sqrt{t}}{\log(t)}\right)\right)$ as $\epsilon^{-1}, t, N \to \infty$. When instead $d = o\left(\min\left(\frac{\log(\epsilon^{-1})}{\log\log(\epsilon^{-1})}, \frac{\sqrt{t}}{\log(t)}\right)\right)$,

$$T^{2d} = \exp\left(o\left(\min\left(\log(\epsilon^{-1}), \sqrt{t}\right)\right)\right). \quad (327)$$

The second term in Eq. (325) thus dominates in this setting, implying the final bound. $\qquad \square$

We also prove a bound on the error of characteristic functions given bounds on the cumulants.

**Lemma 39** (Bound on characteristic functions from cumulants). *Let $p(\boldsymbol{x}), q(\boldsymbol{x})$ be two probability densities on $\mathbb{R}^d$ with characteristic functions $\phi(\boldsymbol{u}), \gamma(\boldsymbol{u})$. Assume each cumulant of order $k$ of $p$ differs from that of $q$ by an additive error of at most $\epsilon^k > 0$. Then*

$$|\phi(\boldsymbol{u}) - \gamma(\boldsymbol{u})| \leq \max\left(\exp\left(\exp\left(\epsilon\|\boldsymbol{u}\|_1\right) - 1\right) - 1, 1 - \exp\left(1 - \exp\left(\epsilon\|\boldsymbol{u}\|_1\right)\right)\right). \tag{328}$$

*Proof.* We have that:

$$\begin{aligned}
|\phi(\boldsymbol{u}) - \gamma(\boldsymbol{u})| &\leq |\phi(\boldsymbol{u})| \left|1 - \exp\left(\ln(\gamma(\boldsymbol{u})) - \ln(\phi(\boldsymbol{u}))\right)\right| \\
&= \left|1 - \exp\left(\ln(\gamma(\boldsymbol{u})) - \ln(\phi(\boldsymbol{u}))\right)\right|.
\end{aligned} \tag{329}$$

By the definition of the cumulant generating function,

$$|\ln(\gamma(\boldsymbol{u})) - \ln(\phi(\boldsymbol{u}))| \leq \sum_{\|\boldsymbol{k}\|_1 > 0} \prod_{j=1}^{d} \frac{\left|(i\epsilon u_j)^{k_j}\right|}{k_j!} = \exp\left(\epsilon\|\boldsymbol{u}\|_1\right) - 1. \tag{330}$$

Taking into account the absolute values yields the final result. $\qquad\square$

This characteristic function bound combined with Theorem 36 gives us the following corollary.

**Corollary 40** (Bound on Lévy–Prokhorov metric from cumulants). *Let $p_N(\boldsymbol{x}), q_N(\boldsymbol{x})$ be two sequences of distributions on $\mathbb{R}^d$ with characteristic functions $\phi_N(\boldsymbol{u}), \gamma_N(\boldsymbol{u})$, respectively. Assume each cumulant of order $k$ of $p_N$ differs from that of $q_N$ by an additive error of at most $\epsilon^k > 0$. Assume as well that $\int_{\|\boldsymbol{x}\|_\infty \geq \frac{1}{2d}\epsilon^{-0.99}} p(\boldsymbol{x}) \leq \mu$ for some ($N$-dependent) $\mu$. Finally, assume that $d = \mathrm{o}\left(\frac{\log(\epsilon^{-1})}{\log\log(\epsilon^{-1})}\right)$ as $N \to \infty$. The Lévy–Prokhorov distance between $p_N, q_N$ is bounded by:*

$$\pi(p_N, q_N) = \mathrm{O}\left(\frac{d\log\left(\epsilon^{-1}\right)}{\epsilon^{-0.99}} + \mu\right). \tag{331}$$

*Proof.* Let $T = \frac{1}{d}\epsilon^{-0.99}$. From Theorem 36,

$$\pi(p_N, q_N) \leq \tilde{\mathrm{O}}\left(T^{2d}\right) \sup_{\|\boldsymbol{u}\|_\infty \leq T} |\phi_N(\boldsymbol{u}) - \gamma_N(\boldsymbol{u})| + \mathrm{O}\left(\frac{d\log(T)}{T}\right) + \mathrm{O}(\mu). \tag{332}$$

From Lemma 39,

$$\sup_{\|\boldsymbol{u}\|_\infty \leq T} |\phi_N(\boldsymbol{u}) - \gamma_N(\boldsymbol{u})| \leq \Theta\left(\epsilon^{0.01}\right). \tag{333}$$

Similarly,

$$T^{2d} = \exp\left(\mathrm{o}\left(\log\left(\epsilon^{-1}\right)\right)\right). \tag{334}$$

The second term in Eq. (332) thus dominates, implying the given convergence rate. $\qquad\square$

## N   TAIL BOUND LEMMAS

We here prove some helper lemmas giving tail bounds for the marginal entries of $\beta$-Wishart random variables. We will particularly leverage the Bartlett decomposition of $\beta$-Wishart matrices (see Appendix A.4). In the following we will use $|\cdot\rangle$ to denote vectors in the vector space on which the defining representation of $\mathcal{H}^{p+1}(\mathbb{F})$ acts, with basis $|0\rangle \cup \{|i\rangle\}_{i=1}^{p}$.

We first prove that $\beta$-Wishart matrices are robust to certain low-rank perturbations of their Bartlett decomposition. In particular, we show that removing a single column from the Bartlett factor $\boldsymbol{L}$ of a $\beta$-Wishart matrix with many degrees of freedom—effectively reducing the degrees of freedom parameter of the matrix by one—has little effect on the joint distribution of entries of the matrix. Intuitively, this allows us to argue in Appendix D.3 that such a Wishart matrix is approximately independent from these entries.

**Lemma 41** (Robustness of $\beta$-Wishart Bartlett decompositions). *Let $W$ be a $(p+1) \times (p+1)$ $\beta$-Wishart matrix with $r$ degrees of freedom and identity scale matrix. Let $LL^\dagger$ be the Bartlett decomposition of $W$. Let $\tilde{L}$ be $L$ with the first column removed, and define $\tilde{W} \equiv \tilde{L}\tilde{L}^\dagger$. For every $t \geq 0$,*

$$\mathbb{P}\left[r^{-\frac{1}{2}} \max_{i,j \in [1,...,p]} \left|\langle i|\, W\, |j\rangle - \langle i|\, \tilde{W}\, |j\rangle\right| \geq t^2 + \sqrt{2}\beta^{-\frac{1}{2}}r^{-\frac{1}{4}}t + r^{-\frac{1}{2}}\right] \leq p \exp\left(-\frac{\sqrt{r}}{2}t^2\right). \tag{335}$$

*In particular, the argument converges to zero under the Ky Fan metric as $r \to \infty$ at a rate $\Omega\left(\frac{\sqrt{r}}{\log(pr)}\right)$.*

*Proof.* Let $N_{0,i}$ be the $(i,0)$ entry of $L$. Note that the $N_{0,i}$ are i.i.d. The lemma statement is then equivalent to showing that

$$\mathbb{P}\left[\max_{i,j \in [1,...,p]} \left|N_{0,i}N_{0,j}^*\right|^2 \geq \left(\sqrt{r}t^2 + \sqrt{2}\beta^{-\frac{1}{2}}r^{\frac{1}{4}}t + 1\right)^2\right] \leq p \exp\left(-\frac{\sqrt{r}}{2}t^2\right). \tag{336}$$

This in turn is implied by:

$$\mathbb{P}\left[\beta \max_{i \in [1,...,p]} \left|N_{0,i}\right|^2 \geq \beta\sqrt{r}t^2 + \sqrt{2\beta}r^{\frac{1}{4}}t + \beta\right] \leq p \exp\left(-\frac{\sqrt{r}}{2}t^2\right). \tag{337}$$

$\beta\left|N_{0,i}\right|^2$ is $\chi^2$-distributed with $\beta$ degrees of freedom. The union bound along with standard $\chi^2$ tail bounds (see, e.g., Lemma 1 of Laurent & Massart (2000)) thus immediately imply Eq. (337) and therefore also the final result. $\square$

We now prove a tail bound for a $\chi$-distributed random variable with $D$ degrees of freedom.

**Lemma 42** (Tail bounds for $\chi$-distributed random variables). *Let $\chi$ be $\chi$-distributed with $D$ degrees of freedom. For every $0 \leq t \leq \frac{3}{8}D^{\frac{1}{4}}$,*

$$\mathbb{P}\left[D^{-\frac{1}{4}}\left|\chi - \sqrt{D}\right| \geq \frac{4}{3}t + D^{-\frac{1}{4}}t^2\right] \leq 2\exp\left(-\sqrt{D}t^2\right). \tag{338}$$

*In particular, the argument converges to zero under the Ky Fan metric as $D \to \infty$ at a rate $\Omega\left(\frac{\sqrt[4]{D}}{\sqrt{\log(D)}}\right)$.*

*Proof.* We have from standard $\chi^2$ tail bounds (see, e.g., Lemma 1 of Laurent & Massart (2000)) that:

$$\mathbb{P}\left[\chi^2 - D \geq 2D^{\frac{3}{4}}t + 2\sqrt{D}t^2\right] \leq \exp\left(-\sqrt{D}t^2\right), \tag{339}$$

$$\mathbb{P}\left[\chi^2 - D \leq -2D^{\frac{3}{4}}t\right] \leq \exp\left(-\sqrt{D}t^2\right). \tag{340}$$

This implies that:

$$\mathbb{P}\left[\chi \geq \sqrt{D + 2D^{\frac{3}{4}}t + 2\sqrt{D}t^2}\right] \leq \exp\left(-\sqrt{D}t^2\right) \tag{341}$$

and (for $t \leq \frac{1}{2}D^{\frac{1}{4}}$)

$$\mathbb{P}\left[\chi \leq \sqrt{D - 2D^{\frac{3}{4}}t}\right] \leq \exp\left(-\sqrt{D}t^2\right). \tag{342}$$

Using the general inequality (for $x \geq -1$):

$$\sqrt{1+x} \leq 1 + \frac{x}{2}, \tag{343}$$

we have that:

$$\sqrt{D + 2D^{\frac{3}{4}}t + 2\sqrt{D}t^2} - \sqrt{D} \leq D^{\frac{1}{4}}t + t^2 \tag{344}$$

such that

$$\mathbb{P}\left[\chi - \sqrt{D} \geq D^{\frac{1}{4}}t + t^2\right] \leq \exp\left(-\sqrt{D}t^2\right). \tag{345}$$

Similarly, using the general inequality (for $0 \leq x \leq \frac{3}{4}$):

$$1 - \frac{2x}{3} \leq \sqrt{1-x}, \tag{346}$$

we have under the given assumptions that:

$$\mathbb{P}\left[\sqrt{D} - \chi \geq \frac{4}{3}D^{\frac{1}{4}}t\right] \leq \exp\left(-\sqrt{D}t^2\right). \tag{347}$$

Noting that both $D^{\frac{1}{4}}t + t^2$ and $\frac{4}{3}D^{\frac{1}{4}}t$ are upper bounded by $\frac{4}{3}D^{\frac{1}{4}}t + t^2$ then implies the final result. $\qquad\square$

We end with a tail bound for the off-diagonal entries of a $\beta$-Wishart matrix.

**Lemma 43** (Off-diagonal $\beta$-Wishart tail bounds)**.** *Let $W$ be a $(p+1) \times (p+1)$ $\beta$-Wishart matrix with $r$ degrees of freedom and identity scale matrix. There exist universal constants $C, K > 0$ that depends only on $\beta$ such that, for every $t \geq 0$,*

$$\mathbb{P}\left[r^{-\frac{1}{2}} \max_{i \in [1,\ldots,p]} |\langle 0| \, W \, |i\rangle| \geq 1 + t\right] \leq Kp\exp\left(-C\min\left(t, t^2\right)\right). \tag{348}$$

*Proof.* Let $LL^\dagger$ be the Bartlett decomposition of $W$. Let $\chi_0$ be the $(0,0)$ entry of $L$ and $N_{0,i}$ the $(i,0)$ entry. The lemma statement is then equivalent to showing that

$$\mathbb{P}\left[\chi_0^2 \max_{i \in [1,\ldots,p]} |N_{0,i}|^2 \geq r\left(1+t\right)^2\right] \leq Kp\exp\left(-C\min\left(t, t^2\right)\right). \tag{349}$$

Note that:

$$\mathbb{P}\left[\chi_0^2 \max_{i \in [1,\ldots,p]} |N_{0,i}|^2 \geq r\left(1+t\right)^2\right] \leq \mathbb{P}\left[\max\left(\frac{\chi_0^2}{r}, \max_{i \in [1,\ldots,p]} |N_{0,i}|^2\right)^2 \geq (1+t)^2\right]$$

$$= \mathbb{P}\left[\max\left(\frac{\chi_0^2}{r}, \max_{i \in [1,\ldots,p]} |N_{0,i}|^2\right) \geq 1 + t\right], \tag{350}$$

so by the union bound we need only check whether

$$\max\left(\mathbb{P}\left[\chi_0^2 \geq r\left(1+t\right)\right], \mathbb{P}\left[|N_{0,i}|^2 \geq 1 + t\right]\right) \leq \frac{K}{2}\exp\left(-C\min\left(t, t^2\right)\right) \tag{351}$$

to prove the final result. As $\beta |N_{0,i}|^2$ is $\chi^2$-distributed with $\beta$ degrees of freedom the final result immediately follows from standard $\chi^2$ tail bounds. $\qquad\square$

