# OpenReview forum: "A Unified Theory of Quantum Neural Network Loss Landscapes"
_ICLR.cc/2025/Conference — ICLR 2025 Poster_

### Official Review · Reviewer_ZJd3 · 2024-10-31

**Soundness:** 4
**Presentation:** 3
**Contribution:** 3
**Rating:** 8
**Confidence:** 4

**Summary:**

This work establishes a unified theoretical framework for characterizing the trainability behavior of quantum neural networks. It integrates previous findings on barren plateaus and local minima phenomena by introducing a novel class of random processes.

**Strengths:**

- This paper establishes a unified framework for quantum nerual networks.

- It shows the necessary and sufficient conditions for QNNs to approach the Gaussian process limit, offering comprehensive insights into the trainability of these networks.

- While Jordan algebra is relatively unfamiliar in this domain, the authors have made efforts to clarify its application and relevance to the audience.

Overall, the results presented in this paper are impressive, and I believe they make a valuable contribution to the field of quantum machine learning.

**Weaknesses:**

- The paper should clearly show its contributions by more explicitly comparing what it offers beyond previous works. For example, it would be beneficial to clarify whether the results offer improvements in handling shallow circuits compared to previous studies in local minima.

**Questions:**

1. Is there an algorithm capable of automatically determining the trainability of any given QNNs?

2. Could you explain how to calculate $O^\alpha$? Are there any efficient algorithms for evaluating these calculations? It would be better if you can provide an example calculation or pseudocode for computing $O^\alpha$.

3. Previous work in [Anschuetz & Kiani,2022] analyzes local minima phenomena in local and deep, brickwall circuits. Does the current study address these issues? Furthermore, is it applicable to shallow circuits as well?

4. Does this work cover all previous results concerning BP and local minima?  If possible, it would be better to provide a more comprehensive comparison table or discussion section that explicitly shows how their framework subsumes or extends specific prior results on barren plateaus and local minima.

5. Could you clarify the rationale behind using Jordan algebra in this study?

---

> ### Author Response · Authors · 2024-11-16
>
> We thank the reviewer for their kind remarks on the novelty and contribution of our results. We now address the five questions of the reviewer, which also addresses the weakness of the paper specified by the reviewer.
>
> 1. By our Definition 7, one need only measure the asymptotic scaling of the $\operatorname{Tr}\left(\left(\mathbf{O}^\alpha\right)^2\right)$ and the $\operatorname{Tr}\left(\left(\mathbf{\rho}^\alpha\right)^2\right)$ to determine the asymptotic trainability of a class of quantum neural networks. The latter of these can be explicitly measured on a quantum device through the measurement of basis elements of $\mathcal{A}$; the former, by doing the same on what is called a "block encoding" of $\mathbf{O}$. We now state this following Definition 7. We also provide in the new Appendix E explicit error bounds for this procedure.
>
> 2. A quantum protocol for calculating $\operatorname{Tr}\left(\left(\mathbf{O}^\alpha\right)^2\right)$ is now stated following Definition 7. We have also added the new Appendix E, which gives a step-by-step, detailed outline for an implementation of the protocol using the standard linear combination of unitaries subroutine for block encoding ([Quantum Info. Comput. 12, 901](https://doi.org/10.26421/QIC12.11-12-1)), and shows that this implementation of the protocol is efficient.
>
> 3. [Anschuetz & Kiani (2022)](https://doi.org/10.1038/s41467-022-35364-5) describe local minima in shallow (not deep), local, brickwork circuits. Their argument essentially is a light cone argument: shallow, local ansatzes with local observables explore a subspace of the full Hilbert space, and brickwork circuits are underparameterized with respect to this local Hilbert space dimension. This is also true for sums of local observables, which the authors approximate by many independent light cones. The results of [Anschuetz & Kiani (2022)](https://doi.org/10.1038/s41467-022-35364-5)---and the equivalent results for other classes of local shallow circuits---are therefore implied by our Corollary 6 by taking the Jordan algebra associated with the network to be a direct sum of algebras associated with each light cone. This is now stated in the new Table 1 (added in response to the reviewer's fourth question), clarified in a footnote in Sec. 1.2, and discussed in Sec. 2.1. This amendment also addresses the weakness of the paper specified by the reviewer.
>
> 4. We thank the reviewer for this suggestion. This was the initial idea behind Figure 1 in the initial submission, but we have now replaced it with a table (Table 1) which explicitly references some of the papers our framework generalizes. This table also summarizes what limits of our corollaries encompass these previous results. This amendment also addresses the weakness of the paper specified by the reviewer.
>
> 5. Jordan algebras arise naturally when considering quantum neural networks as operations acting on quantum states, which are Hermitian matrices. As Hermitian matrices over a field span one of the classical Jordan algebras (not, for instance, one of the classical Lie algebras), quantum networks can be considered as automorphisms acting on Jordan algebras. As a benefit, this unified description allows us to consider what were previously thought of as "special cases" of algebraically constrained quantum neural networks on the same footing. For instance, quantum networks composed of matchgates are not describable in a Lie algebraic formalism (see, e.g., [arXiv:2310.11505](https://doi.org/10.48550/arXiv.2310.11505)), but we show in what is now Appendix H that they are describable in a Jordan algebraic formalism. The new Table 1 now also explicitly states that the Lie algebraic setting is a limit of our Jordan algebraic setting.

---

> > ### Comment · Reviewer_ZJd3 · 2024-11-20
> >
> > Thank you for your detailed response. I have increased my score from 6 to 8. Also, I have a few more questions:
> >
> > 1. Can you provide a concrete example to showcase the trainability evaluation? For instance, if the QNN is a brickwork circuit of depth L composed of 2-qubit fully parametrized blocks (e.g., the Cartan decomposition), how can one use the results in this paper to derive the trainability of the QNN? Namely, what are the expressions of the cost variance and the overparametrization ratio in this case?
> >
> > 2. In the pursuit of practical applications for QNNs, several key properties must be balanced, including expressivity, trainability, noise robustness, and resistance to classical simulation. Based on your findings, do you have any insights on how to design QNNs that achieve this balance?

---

> ### Author Response · Authors · 2024-11-22
>
> We thank the reviewer for their raising of their score. We now answer the reviewer's follow-up questions.
>
> ## Response to Question 1
>
> We assume a 1D brickwork geometry with $n$ qubit input $\mathbf{\rho}$, and assume the objective observable $\mathbf{O}$ only acts locally for simplicity (otherwise, one should perform the block decomposition into light cones as mentioned in our initial response to the reviewer). We also only concretely consider the case when $L\leq\operatorname{O}\left(\log\left(n\right)\right)$ (the "shallow setting") as this is the setting the reviewer previously was commenting on, though calculations in the deep setting proceed similarly. We use $\mathcal{H}_L$ to denote the Hilbert space corresponding the to light cone of $\mathbf{O}$.
>
> The number of parameters $p_L$ of the network in the light cone of $\mathbf{O}$ is $p_L\sim L$; similarly, the Hilbert space dimension of the light cone is $N\_L\equiv\dim\left(\mathcal{H}\_L\right)\sim 2^L$ (as there are $n_L\sim L$ qubits in the light cone of $\mathbf{O}$). Assuming $\mathbf{O}$ is given as a Pauli decomposition,
> \begin{equation}
>     r=\frac{\operatorname{Tr}\left(\mathbf{O}\right)^2}{\operatorname{Tr}\left(\mathbf{O}^2\right)}
> \end{equation}
> can be efficiently evaluated classically; typically (i.e., outside of fine-tuned $\mathbf{O}$) this will be $r\sim N\_L$. Similarly,
> \begin{equation}
>     \mathbf{\rho}\_L\equiv\operatorname{Tr}\_{\mathcal{H}\_L^\complement}\left(\mathbf{\rho}\right)
> \end{equation}
> is the state in the light cone of $\mathbf{O}$, and $\operatorname{Tr}\left(\mathbf{\rho}\_L^2\right)$ is just the purity of $\mathbf{\rho}\_L$ which can be estimated via the standard swap test. This gives by our Definition 7:
> \begin{equation}
>     \operatorname{Var}\_{\mathbf{\theta}}\left[\ell\left(\mathbf{\rho};\mathbf{\theta}\right)\right]=\frac{\operatorname{Tr}\left(\mathbf{O}^2\right)\operatorname{Tr}\left(\mathbf{\rho}\_L^2\right)}{\dim_{\mathbb{R}}\left(\operatorname{SU}\left(N_L\right)\right)}\sim\frac{N_L\operatorname{Tr}\left(\mathbf{\rho}_L^2\right)}{N_L^2}\sim\frac{\operatorname{Tr}\left(\mathbf{\rho}_L^2\right)}{\operatorname{\Omega}\left(\operatorname{poly}\left(n\right)\right)}
> \end{equation}
> and
> \begin{equation}
>     p_L\sim L\ll 2^L\sim N_L.
> \end{equation}
> Thus, by our Definition 7, we expect---for $\mathbf{\rho}$ an area law state such that $\operatorname{Tr}\left(\mathbf{\rho}_L^2\right)$ does not vanish exponentially with $n$---the model to have no barren plateaus, but to have poor local minima. In this special setting, the former was proved in [Nat. Commun. 12, 1791](https://doi.org/10.1038/s41467-021-21728-w) and [PRX Quantum 2, 040316](https://doi.org/10.1103/PRXQuantum.2.040316), and the latter in [Nat. Commun. 13, 7760](https://doi.org/10.1038/s41467-022-35364-5).
>
> ## Response to Question 2
>
> As discussed in our Sec. 6, we do not believe there are any generic, randomly initialized QNNs which achieve this balance during training. Our results indicate that barren plateaus and poor local minima in randomly initialized QNNs can only be simultaneously avoided if $\dim\left(\mathcal{A}\right)\sim\operatorname{poly}\left(n\right)$ (where $\mathcal{A}$ is the Jordan algebra associated with the QNN). However, this suggests the existence of an efficient classical simulation algorithm, as discussed in our reply to Reviewer 3K5s.
>
> As discussed in our Sec. 6, though our results are pessimistic, this still leaves settings with a quantum-classical separation in machine learning (indeed, many separations have been proved unconditionally!). These include:
>
> 1. QNNs which have no quantum advantage during training but still having an advantage during inference, as is done in, e.g., [PRX Quantum 4, 020338](https://doi.org/10.1103/PRXQuantum.4.020338) and [arXiv:2402.08606](https://doi.org/10.48550/arXiv.2402.08606).
>
> 2. QNNs which are warm started, as is done in [arXiv:2404.10044](https://doi.org/10.48550/arXiv.2404.10044).
>
> 3. Quantum models trained via specialized algorithms (i.e., not uniformly-initialized gradient descent), as is done in [Nat. Phys., 17, 1013](https://doi.org/10.1038/s41567-021-01287-z) and [STOC 2024, pp. 1343--1351](https://doi.org/10.1145/3618260.3649722).

---

> > ### Comment · Reviewer_ZJd3 · 2024-11-23
> >
> > Thank for your detailed response.
> >
> > Regarding question 1, I would like to seek further clarification on the following points:
> > 1. What is the specific role of Jordan algebra in this setting? Additionally, is it correct to assume that for any circuit composed of fully parametrized blocks, the "alpha" index can simply be dropped? This interpretation seems consistent with your response, but I would appreciate further clarification.
> > 2. Should there be only $n_L = L$ qubits in the light cone of $O$?
> > 3. For a 1D brickwork circuit of depth L with each block composed of R_zz and R_x, how can one use the results in this paper to derive the expressions of the cost variance and the overparametrization ratio in this case?

---

> > > ### Author Response · Authors · 2024-11-24
> > >
> > > 1. If there is no algebraic structure---like in the example given---then correct, the alpha index can be dropped. In particular, in the specific example given, the Jordan algebraic structure essentially plays no role. If there were multiple light cones (e.g., if the objective observable were a sum of local terms), then this can be modeled by a direct sum of Jordan algebras corresponding to Hermitian matrices contained in each light cone.
> > > 2. Yes; this and also a couple of other typos in the reply have now been fixed. The conclusions are identical.
> > > 3. Any 1D brickwork architecture that "scrambles" over a uniformly random initialization---more specifically, any architecture which forms an $\epsilon$-approximate $t$-design---gives an identical calculation to that given in my reply. See Assumption 12 of the manuscript for a formal statement of this assumption.

---

> > > > ### Comment · Reviewer_ZJd3 · 2024-11-25
> > > >
> > > > 1. It is known that the 1D brickwork circuit composed of nearest-neighbor Rzz gates and Rx gate is not scrambled. So Question 3 in the last comment should be replied seriously.
> > > >
> > > > 2. In fact, all the above questions are intended to make the author give a solid example to convince the readers that the Jordan algebra tool here could be useful and give calculable results beyond previous literature, but unfortunately, there seems to be no such clear example so far. We sincerely hope that the author could clarify this point, which is of great importance for the impact of the paper.

---

> ### Author Response · Authors · 2024-11-25
>
> We address the reviewer's concerns jointly:
>
> Yes, 1D, nearest-neighbor Rzz and Rx gates are a subset of "matchgates"; this is isomorphic to $\operatorname{SO}\left(N\right)$ as a Lie group. So while these gates do not scramble over the full space of unitaries (i.e., $\operatorname{SU}\left(N\right)$), they scramble over $\operatorname{SO}\left(N\right)\subset\operatorname{SU}\left(N\right)$ (see, for instance, [arXiv:2310.11505](https://doi.org/10.48550/arXiv.2310.11505)). Our theorems thus apply exactly as in the $\operatorname{SU}\left(N\right)$ case, with the only difference being instead of taking $\beta=2$ one takes $\beta=1$ in our formulas.
>
> This is a setting where the Jordan algebaic description of the network is more natural than, say, a Lie algebraic description. For instance, if the objective observable $\mathbf{O}$ is a real symmetric matrix, it does _not_ belong to the algebra $\mathfrak{so}\left(N\right)$---this precludes any Lie algebraic description (indeed, [arXiv:2310.11505](https://doi.org/10.48550/arXiv.2310.11505)---which studies matchgate networks---includes "beyond the dynamical Lie algebra" in the title). However, $\mathbf{O}$ _does_ belong to the Jordan algebra of real symmetric matrices $\mathcal{H}^N\left(\mathbb{R}\right)$, and matchgates are isomorphic to the automorphism group of this space. Our results thus also hold in this setting, where the associated Jordan algebra $\mathcal{A}\cong\mathcal{H}^N\left(\mathbb{R}\right)$. The Jordan algebraic structure of this network is what allows us to calculate results beyond the existing literature; for instance, for the first time, we calculate the local minima distribution of networks composed of these operations in our Corollary 6. This calculation heavily relies on the Jordan algebraic structure of the network, as it leverages the Wishart process description and real Wishart matrices are not elements of the Lie algebra $\mathfrak{so}\left(N\right)$ but _are_ elements of the Jordan algebra $\mathcal{H}^N\left(\mathbb{R}\right)$.

---

### Official Review · Reviewer_3K5s · 2024-11-02

**Soundness:** 2
**Presentation:** 1
**Contribution:** 4
**Rating:** 3
**Confidence:** 3

**Summary:**

It is known that (classical) neural networks typically behave like Gaussian processes in the infinite-width limit under reasonable regularity assumptions. In this work, this result is extended to Quantum Neural Networks which behave like a Wishart process instead. This result is then used to study training, similar to Neural Tangent Kernels in the classical case, and the loss landscape thereof in particular with respect to barren plateus.

While the authors should be commended on their opus magnum and an exciting set of ideas / contributions from theoretical physics to machine learning, I share some concerns regarding technical soundness, presentation and topical fit to this publication venue.

**Strengths:**

1) Several new ideas are introduced from theoretical physics / quantum mechanics and connected with neural networks

2) The claimed result is a major generalization of major theorems in "classical" neural networks such as the infinite-width limit of neural networks (Gaussian processes) or Neural Tangent Kernel theory. While several results are derived from this for quantum NNs, there is potential that these ideas can also lead to new results for classical NNs, e.g. via the correspondence principle.

**Weaknesses:**

1) Technical: The paper lacks basic definitions and introduction of notation, or precision in doing so, that are crucial to follow the train of thought. This is detrimental as several parts of the notation are non-standard, or at least non-standard within the subsections and audiences of this publication venue. For example, in Theorem 1, a notation is introduced as "....denotes the projection into a (Jordan sub-algebra) A_α ....", first mention in the introduction around eq 2. Nowhere in the main body is it defined what this precisely means. QM people are left with a vague idea, but not more. Instead, the reader must dig deep into the appendix. It was until around eq 178 in the appendix until I got a better idea, by implication only. Further examples of lack of definition are around eq 5, 7 and 13, or mathematical objects \mathcal{G} and \mathcal{H} in Line 291. An example for unclear notation is also in eq 21 in Theorem 2. While non-standard notation is perfectly fine, it doesn't help much if things are kept too informal, even when things "are easy to see", and it should defined and introduced properly. This list is non-exhaustive.

2) Presentation: When agreeing to review, I was expecting a 10-page paper in line with the ICLR submission guidelines. While this work technically fulfills these requirements, one can learn and understand very little without working through the 45 page appendix. More honestly, this is a 59-page paper that has been last minute broken into a 10 page teaser with several big claims, many with unclear or vague rationale, followed by the actual paper.

Minor:
I know this publication venue typically expects numerical experiments, of which there is a lack of any in this paper. From a theoretical physics perspective, this is fine though.

**Questions:**

what is the potential of classical limits within this framework?

---

> ### Author Response · Authors · 2024-11-16
>
> We thank the reviewer for their kind remarks on the contributions of this work. We apologize that many of the technical details of the work are placed in appendices; this is due to the number of results we show, as well as the lengths of the required proofs and formal definitions. We have updated the manuscript in an effort to make the main text more readable and intuitive, primarily by including many mathematical definitions in the main text that were previously only given in the appendices. A nonexhaustive list of expressions which are now explicitly defined in the main text include: the density of the gamma distribution; the convolution; the projection of an algebra element into a subalgebra; and the semisimple decomposition of an algebra. We have also fixed many typos that were present in the initial submission, which accounted for many mistakenly undefined quantities.
>
> We now address the reviewer's concerns regarding the manuscript.
>
> 1. We thank the reviewer for pointing out these undefined quantities throughout the paper---many of these were typos, mistakenly unremoved from a previous version. These (and others where found) have now been defined in the main text. We have also removed definitions and cumbersome notation which were unused, which we believe clarifies many statements which were confusing in the original submission. This includes those in the statements of Theorems 2 and 3 that the reviewer mentioned.
>
> 2. We again apologize that many of the technical details of the work are in the appendices. In the updated manuscript, we are more comprehensive in our introduction of Jordan algebras in Sec. 2.2 in an effort to make Sec. 3 easier to follow without diving into the appendices. We also make an effort in Secs. 4 and 5 to sketch the techniques we use in the proofs of our results, such that the proof strategies can be understood without requiring all of the technical details supplied in the appendices.
>
> Finally, we comment on the reviewer's question on the potential of classical limits within this framework; indeed, a corollary of our result is that each trainable quantum neural network (QNN) has an equivalent, efficient, quantum-inspired classical neural network. As discussed in Sec. 3, every quantum neural network architecture has associated with it a Jordan algebra $\mathcal{A}\cong\bigoplus_\alpha\mathcal{A}\_\alpha$ generated by $\mathbf{U}\_{\mathbf{\theta}}^\dagger\mathbf{O}\mathbf{U}\_{\mathbf{\theta}}$ under the anticommutator; this leads to the general algebraic form of a quantum loss function given in Eq. (16). Note that the argument of the trace in this equation involves matrices in the $N_\alpha\times N_\alpha$ defining representations of the $\mathcal{A}\_\alpha$. If the $N_\alpha$ grow only polynomially with the system size, then, this immediately gives a quantum-inspired classical neural network: given a Hermitian, $N_\alpha\times N_\alpha$-dimensional encoding of the input, one conjugates it with a parameterized unitary matrix, and takes the Frobenius inner product with an $N_\alpha\times N_\alpha$ Hermitian matrix to calculate the loss.
>
> This general "dequantization" procedure for algebraically-constrained quantum networks has been previously noted in the literature, see, e.g., [Quantum 7, 1189](https://doi.org/10.22331/q-2023-11-28-1189) and [arxiv:2308.01432](https://doi.org/10.48550/arXiv.2308.01432). As efficient trainability on a quantum computer _also_ requires the $N_\alpha$ to grow only polynomially with the system size by our Definition 7, we state in Sec. 6: "...it seems unlikely that there exists any computational quantum advantage during the training of QNNs...due to efficient classical simulation algorithms for quantum systems algebraically constrained to explore a low-dimensional subspace..." We now more explicitly state in Sec. 6 that this is an implication of our Definition 7. We also now make a similar statement toward the end of Sec. 1.1.

---

### Official Review · Reviewer_CGjm · 2024-11-02

**Soundness:** 2
**Presentation:** 3
**Contribution:** 3
**Rating:** 8
**Confidence:** 2

**Summary:**

The authors present a theory study of quantum neural networks showing, in contrast to classical neural network studies, that QNNs obey a so-called Wishart process. In doing so, they derive the existence of barren plateaus and show when a QNN behaves like a Gaussian process.

**Strengths:**

The paper is written very well and tackles a valid problem. The theoretical analysis appears to be well performed and the authors have done a lot to highlight the connection to prior results.

**Weaknesses:**

The main weakness I see is in the discussion around limits. It isn't always clear in what limit the authors are discussing the behaviour of the networks. It is also reasonable that a classical neural network's neural tangent kernel matrix has eigenvalues behaving a Wishart distribution and evolving during training. Further, a covariance matrix used to describe a GPR will likely also look very Wishart. While the math and results were presented in the paper, it was never completely clear to me how this all aligned and how I should think about the WIshart process.

**Questions:**

* Where does entanglement fall into these results? One criticism of quantum ML is that entanglement appears not to provide much a benefit in real-world problems. Is there an element of the theory presented here that can touch on this problem?

* What exactly should follow a Wishart distribution in the QNN framework? In a neural network, it is the weights, whereas the NTK will likely look Wishart and the Fisher matrix the same. Purely looking at the math, the NTK and Fisher matrix of a QNN should do the same.

---

> ### Author Response · Authors · 2024-11-16
>
> We thank the reviewer for their kind remarks on the organization of the paper and the quality of the technical analysis. We acknowledge that the precise asymptotic limit we are taking was confusingly worded in the theorem statements; we are now more clear, and directly state in both the main text and theorem statements that the limit we are taking is as the dimension of the algebra $\mathcal{A}$ associated with the quantum neural network (QNN) goes to $\infty$ (i.e., the "wide" limit of many qubits).
>
> We now address the two questions of the reviewer.
>
> ## Response to Question 1
>
> Entanglement arises in a subtle way, which we now make more clear. Eqs. (4) and (6) are functions of the _generalized entanglement_ ([Phys. Rev. Lett. **92**, 107902](https://doi.org/10.1103/PhysRevLett.92.107902)) of the input $\mathbf{\rho}$ and objective observable $\mathbf{O}$, respectively, with respect to the algebraic structure of the Jordan algebra $\mathcal{A}=\bigoplus_\alpha \mathcal{A}_\alpha $ associated with the network. To describe the essence of this property, we first recall that "normal" entanglement is defined as the inability for a state $\mathbf{\rho}$ belonging to a tensor product Hilbert space $\mathcal{H}_A\otimes\mathcal{H}_B$ to be decomposed into a tensor product of states itself. This can be probed by, for instance, taking the partial trace of the state over a subsystem $\mathbf{\rho}_A\equiv\operatorname{Tr}_B\left(\mathbf{\rho}\right)$ and considering the resulting purity $\operatorname{Tr}\left(\mathbf{\rho}_A^2\right)$; $\mathbf{\rho}$ is entangled across the cut between Hilbert spaces $\mathcal{H}_A$ and $\mathcal{H}_B$ if and only if $\operatorname{Tr}\left(\mathbf{\rho}_A^2\right)<1$, where the smaller $\operatorname{Tr}\left(\mathbf{\rho}_A^2\right)$ is the more entangled the state is.
>
> Similarly, a state $\mathbf{\rho}$ belonging to a direct sum of algebras $\bigoplus_\alpha \mathcal{A}\_\alpha$ exhibits _generalized entanglement_ if $\operatorname{Tr}\left(\left(\mathbf{\rho}^\alpha\right)^2\right)<1$, where $\mathbf{\rho}^\alpha$ is the projection of $\mathbf{\rho}$ into the subalgebra $\mathcal{A}_\alpha$ (see Eq. (12) of the submission for a formal definition). This quantity governs the trainability of the network as described in Definition 7, in the sense that data sets composed of highly entangled states yield barren plateaus and poor local minima in the loss landscape of the QNN. In the latest version of the manuscript, we now state this explicitly at the end of Sec. 5.1 and Appendix C.1.
>
> ### Entanglement's Impact on Expressivity
>
> What has been described is how entanglement influences the trainability of a given quantum neural network architecture, which is the focus of this work; the reviewer's comment also asks about the potential for an expressivity advantage using entanglement. We note that entanglement has been used to rigorously derive settings in which there is a provable advantage in quantum machine learning over classical machine learning models in learning time series. Typically this is studied through the lens of "quantum contextuality," which is a byproduct of quantum entanglement. We cite two of these works in Sec. 6: [PRX Quantum **4**, 020338](https://doi.org/10.1103/PRXQuantum.4.020338) and [arXiv:2402.08606](https://doi.org/10.48550/arXiv.2402.08606). [arXiv:2410.03094](https://doi.org/10.48550/arXiv.2410.03094) is very recent (released after this work was submitted) but also proves a similar result. See also [arXiv:1403.3351](https://doi.org/10.48550/arXiv.1403.3351) for how contextuality can be used to more efficiently model natural language than noncontextual models.
>
> ## Response to Question 2
>
> In an unfortunate overloading of terms, in the quantum information literature the "quantum neural tangent kernel (QNTK) limit" refers to the existence of a Gaussian process limit for the loss function---we agree with the reviewer that the neural tangent kernel is often Wishart-distributed in the classical setting. What we show is that, for QNNs initialized uniformly at random, the _loss function itself_ is not Gaussian-distributed but instead is Wishart-distributed. This leads to the different phenomenological behavior between quantum and classical neural networks, as discussed in Sec. 1.1. Indeed, the covariance of the resulting process in the QNN setting is _not_ Wishart-distributed; see the expression we give for the covariance in Eq. (25). We now clarify throughout the manuscript that we show that the loss function---not the associated kernel matrix---of QNNs is what converges to a Wishart process, and have replaced most uses of the term "QNTK limit" with "Gaussian process limit" to avoid confusion.

---

> > ### Comment · Reviewer_CGjm · 2024-11-28
> > **Response to authors**
> >
> > Thank you for the extensive response and addressing the questions raised. I think the theoretical insight, while perhaps not too actionable, is nonetheless of interest to the community.

---

### Official Review · Reviewer_TpUa · 2024-11-03

**Soundness:** 4
**Presentation:** 3
**Contribution:** 2
**Rating:** 8
**Confidence:** 3

**Summary:**

The article defines a quantum neural network (QNN) as a chain of unitary operations applied to an initial quantum state. The loss function is the expectation value of an operator the resulting state with respect to an operator. This definition is common and general. The authors then assume random initialization and show that the loss converges to what they term a Wishart process in the limit of infinitely wide networks. This is in contrast to classical neural nets, where the limit is a Gaussian process (GP).   Having established the Wishart process framework, they use it to formulate several results within it:

(1) While the possibility of QNN convergence to GPs had already been studied in prior cited work, the authors formulate conditions within this framework for the convergence.

(2) Furthermore, they extend and generalize two results on trainability of neural networks from cited articles:
- Conditions for barren plateaus, i.e. vanishing gradients.
- The density of local minima is concentrated near the global minimum iff the network is over-parameterized.

**Strengths:**

- The article is very well structured and seems technically of excellent quality. (However, I did not check or verify the proofs in the Appendix in detail)
- The overall question on the structure of quantum neural networks and their potential advantages is relevant and the new findings advance the field.
The new Wishart process framework introduced in this article seems useful, as demonstrated by the results derived mentioned in the summary.

**Weaknesses:**

- The results are not groundbreaking new but generalizations of previous results and formulated in a more general language and setting.
- The results are a bit underwhelming, essentially "it seems unlikely that there exists any computational quantum advantage during the training of QNNs", which already seemed unlikely before. But this is expected, and this evidence is also valuable, so this is not a strong weakness.
- The main text is hard to understand without the long Appendix. However, this seems inevitable given the page limit and the subject.

**Questions:**

Can you give more explanation and background on N (equation 25) in the main text?

---

> ### Author Response · Authors · 2024-11-16
>
> We thank the reviewer for their kind remarks on the organization, technical quality, and contributions of the work. We agree with the reviewer that it is unfortunate that these results do not point to any new parameter regimes where one might hope for a generic, large, computational advantage during the training of quantum neural networks. Instead, the value of our work is that the nonexistence of such a setting is _proved_ for networks which do not follow the conditions of Definition 7; we specify exactly the parameter regimes where quantum neural networks (QNNs) are untrainable. In the latest version of the manuscript we also provide a protocol for empirically checking whether or not a given QNN architecture is asymptotically trainable, discussed just after Definition 7.
>
> We now address the question on $\mathcal{N}$. Here, we are studying when a class of randomly initialized QNNs with loss function $\ell\left(\mathbf{\rho}\right)$ are such that this loss function asymptotically approaches a Gaussian process. In typical settings, the variance of $\ell\left(\mathbf{\rho}\right)$ vanishes asymptotically; we thus consider normalizing by some $\mathcal{N}\geq 1$ such that $\mathcal{N}\ell\left(\mathbf{\rho}\right)$ has variance nonvanishing asymptotically. However, if there exists no $\mathcal{N}$ such that:
>
> 1. this variance is nonvanishing asymptotically, and
>
> 2. the higher-order cumulants vanish asymptotically,
>
> then $\mathcal{N}\ell\left(\mathbf{\rho}\right)$ cannot form a Gaussian process. What Eq. (25) of the original submission described was when this second condition is satisfied.
>
> We acknowledge that the associated section (Sec. 5.2) was written in a confusing manner. We have now rewritten this section. We now emphasize that we are demonstrating when there exists an $\mathcal{N}$ satisfying these two conditions, and explicitly state that (what is now) Eq. (24) describes when the second condition is possible.

---

> > ### Comment · Reviewer_TpUa · 2024-11-25
> >
> > Thank you for addressing my question around Section 5.2 and the notation introducing N. I keep the rating of 8.

---

### Author Response · Authors · 2024-11-16
**Response to All Reviewers**

We thank all of the reviewers for their suggestions and questions, as well as for their highlighting of the contribution and technical quality of the work. The reviewers' main concern regarding the initial submission was in its clarity and presentation, which we have made an effort to rectify in the newly posted, revised version.

Specifically, in the newly submitted version of the manuscript, we now:

1. More specifically define terms and operations we use throughout the main text, which previously were left ambiguous if one did not search for the precise definitions in the appendices. The main text of the latest version is much more self-contained and readable as a result.

2. Fixed typos and stray sentences, which we thank Reviewer 3K5s for finding many of.

3. More clearly state how our results generalize and unify previous results in quantum neural network landscape theory; in particular, we thank Reviewer ZJd3 for the suggestion of a table which clarifies precisely how our results generalize previous results, which we have now included as Table 1.

4. Rewrote subsections in a more clear way when they led to ambiguity and questions, including those posed by the reviewers.

These changes are highlighted in the "Revisions" feature in OpenReview. We more specifically address the comments and concerns of each reviewer in a reply to each.

---

> ### Comment · Reviewer_3K5s · 2024-11-24
> **Comments brushed off**
>
> In a paper that is all about formal constructions, the comment is brushed off as "Fixed typos and stray sentences" and reader did not "search for the precise definitions in the appendices". Unfortunately, the "typos" are mostly found within the formal construction at the center of attention of this work and hamper the "precision" thereof. Also, it is the authors' job to present the work without having readers "search for definitions in the appendix". If that is not possible, one may question whether this is the right venue for this work.

---

> > ### Author Response · Authors · 2024-11-24
> >
> > We agree. Sorry, the comment "search for the precise definitions in the appendices" was meant to emphasize that our original submission was not sufficiently clear and that it was a clear weakness of the work, as no reader should have to sift through appendices for important definitions. It was not our intention to shift blame to the reader.
> >
> > We hope the revised submission has addressed these concerns on clarity and that the main text is now more self-contained.

---

### Meta-Review · Area_Chair_mzs8 · 2024-12-19

**Metareview:**

The current paper proves that the loss and its first two derivatives of quantum neural networks asymptotically form Wishart processes. The reviewers acknowledged that it is a solid contribution extending classical results with deep theoretical insights. The main criticisms arise from the readability of the main text, based on which the authors made a positive effort in the rebuttal phase. There is still notable place for improvements in the self-containedness after re-shuffling the contents and the overall clarity, e.g., on the concept of "ansatz" and definition 7 in the conclusion section.

In the final version, please consider addressing all reviewer's comments especially the criticisms raised by reviewer 3K5s, and make sure the main definitions and theorems are clearly presented and self-contained.

**Additional Comments On Reviewer Discussion:**

All reviewers acknowledged the soundness of the contribution. The main criticism is regarding the notations and clarity.

The reviewer 3K5s, only one on the rejection side with the others giving a high score of 8, acknowledged that the clarify is improved compared to the initial submission. Their remaining concerns is on that the reshuffling of contents between main text and appendix leads to incoherent presentation.

The paper is recommended to be accepted based on its theoretical value. It could be a spotlight paper if the contents are well organized and clearly presented, which is not the case here.

---

> ### Public Comment · ~Eric_Ricardo_Anschuetz1 · 2025-02-06
>
> We once again thank the reviewers and Area Chair for their reviews and comments. We have uploaded the camera-ready version of the manuscript, with clarifications on the definition of "ansatz" near Eq. (7) and the definition of "Hilbert space" in the text of Definition 7.

---

### Decision · Program_Chairs · 2025-01-22

Accept (Poster)